ARTICLES

# Lymph node homeostasis and adaptation to immune challenge resolved by fibroblast network mechanics

Harry L. Horsnell [1], Robert J. Tetley[2], Henry De Belly[3], Spyridon Makris[1], Lindsey J. Millward[1], Agnesska C. Benjamin[1], Lucas A. Heeringa[1], Charlotte M. de Winde[1], Ewa K. Paluch[3], Yanlan Mao[2,4] and Sophie E. Acton [1]✉

**Emergent physical properties of tissues are not readily understood by reductionist studies of their constituent cells. Here, we show molecular signals controlling cellular, physical, and structural properties and collectively determine tissue mechanics of lymph nodes, an immunologically relevant adult tissue. Lymph nodes paradoxically maintain robust tissue architecture in homeostasis yet are continually poised for extensive expansion upon immune challenge. We find that in murine models of immune challenge, cytoskeletal mechanics of a cellular meshwork of fibroblasts determine tissue tension independently of extracellular matrix scaffolds. We determine that C-type lectin-like receptor 2 (CLEC-2)–podoplanin signaling regulates the cell surface mechanics of fibroblasts, providing a mechanically sensitive pathway to regulate lymph node remodeling. Perturbation of fibroblast mechanics through genetic deletion of podoplanin attenuates T cell activation. We find that increased tissue tension through the fibroblastic stromal meshwork is required to trigger the initiation of fibroblast proliferation and restore homeostatic cellular ratios and tissue structure through lymph node expansion.**

Unlike developmental systems of progressive tissue growth and maturation[1], the homeostatic state of adult tissues is robust, maintaining form and function[2]. Secondary lymphoid organs are uniquely able to dramatically change size in response to immune challenge, adapting to the increased space requirements of infiltrating and proliferating lymphocytes while remaining structurally and functionally intact[3–5]. As a mechanical system, lymph nodes (LNs) continually resist and buffer forces exerted by lymphocytes entering and leaving the tissue[6], and managing diurnal fluctuations in cell trafficking[7]. Tissue size is determined by lymphocyte numbers, highlighted by the small organ size in genetic models blocking lymphocyte development (Rag1 knockout (KO))[8], and the ability of the tissue to expand two- to tenfold in size to accommodate lymphocyte proliferation through adaptive immunity[3–5,9]. Here, we ask whether mechanical forces determine the kinetics of LN tissue remodeling through immune challenge.

LNs function at the interface of immunity and fluid homeostasis, constructing a physical three-dimensional cellular meshwork linking fluid flow, immune surveillance, and adaptive immunity[10]. The most populous stromal cell component are FRCs, which span the whole tissue generating an interconnected cellular network with small-world properties[11], forming robust clustered nodes with short path lengths and surrounding bundles of extracellular matrix fibers[12]. It is widely assumed that extracellular matrix scaffolds are the predominant force-bearing structures determining tissue mechanics[1,2]. However, the relative contributions of the cellular structures versus the underlying extracellular network to tissue mechanics have not been addressed in a highly cellularized system undergoing such extensive expansion[2]. As LNs expand in response to immune challenge, the tissue becomes more deformable[3]. During LN expansion, the FRC network maintains connectivity through the elongation and increased spacing between FRCs, increasing mesh size of the network[3]. It is also known that CLEC-2+ dendritic cells are required to prime the stromal architecture for tissue expansion and affects LN deformability[3,4], but the downstream impacts on the mechanical properties of the cellular network and extracellular matrix scaffolds driving the adaptation in tissue mechanics are unknown. During tissue expansion, FRCs reduce their adhesion to the underlying extracellular matrix bundles, and these matrix scaffolds become fragmented[9]. This makes LNs an ideal model system to address the relative mechanical contributions of cellular and material structures to emergent tissue mechanics.

Using molecular cell biology approaches in combination with quantitative biophysics, we asked how LN tissue mechanics is controlled and what impact tissue mechanics has on adaptive immune responses. We provide evidence that tissue tension is offset and sensed by the fibroblastic reticular network through actomyosin structures. FRCs require PDPN expression to regulate network tension through acute LN expansion, and signaling downstream of PDPN determines FRC membrane tension and regulates their response to external forces. We find that tension through the FRC network gates fibroblast proliferation and that the failure of FRCs to sense increasing forces inhibits T cell activation and proliferation. These results show that adaptive immune responses are regulated through mechanical cues sensed by the stromal architecture and identify PDPN as a key mechanical sensor in fibroblasts.

[1]Stromal Immunology Group, MRC Laboratory for Molecular Cell Biology, University College London, London, UK. [2]Tissue Mechanics Group, MRC Laboratory for Molecular Cell Biology, University College London, London, UK. [3]Physiological Laboratory, Department of Physiology, Development and Neuroscience, University of Cambridge, Cambridge, UK. [4]Institute for the Physics of Living Systems, University College London, London, UK. ✉e-mail: s.acton@ucl.ac.uk

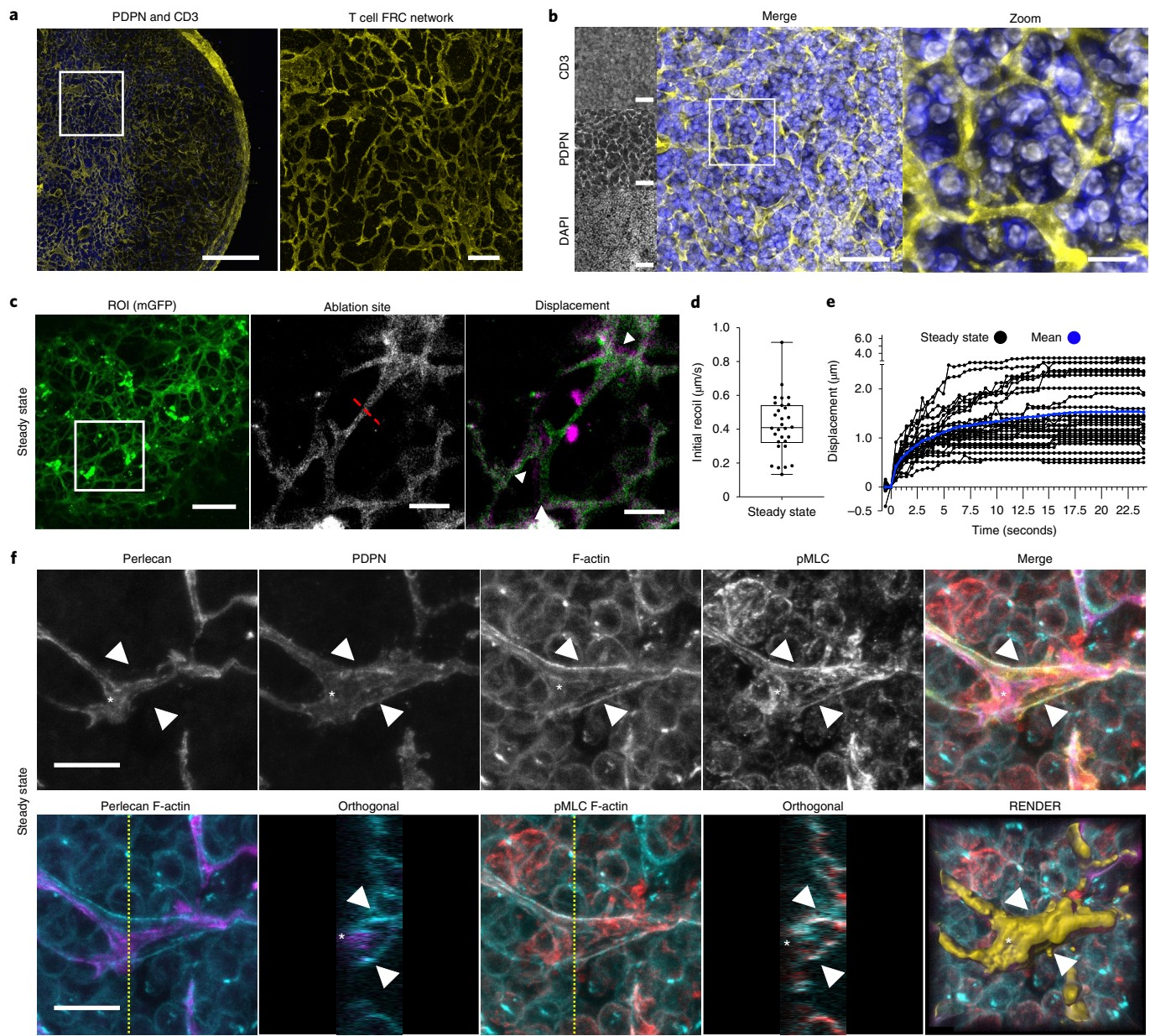

**Fig. 1 | The FRC network is under mechanical tension in steady-state LNs. a**, LN tile scan (left), paracortex (right) maximum z-projection of PDPN (FRCs, yellow), and CD3 (T cells, blue). Scale bars, 500 μm (left) and 25 μm (right). **b**, FRC network structure and T cell compaction in vibratome slices of PDPN (FRCs), CD3 (T cells), and DAPI (nuclei). Scale bars, 50 μm (left) and 10 μm (right). **c**, PDGFRα⁺mGFP⁺ (FRCs) ablation region of interest (ROI) (white box) (left) and cut site (red dotted line) (middle). Scale bars, 50 μm. Right: Recoil displacement (arrowheads) with pre- (green) and postcut (magenta) overlay (right). Scale bar, 10 μm. **d**, Initial recoil velocity (μm s⁻¹). Each point represents an ablation. Box plot indicates median, interquartile range, and minimum/maximum. **e**, Individual recoil curves (black) compared to the mean (blue). **d,e**, n = 30 individual ablations over 10 LNs. **f**, Paracortical steady-state FRCs of perlecan (matrix; purple), PDPN (FRC; yellow), phalloidin (F-actin; blue), and pMLC (red). Asterisk and arrowheads indicate F-actin cables. Orthogonal views (yellow dotted line, yz axis, 10 μm depth). Representative image of three independent experiments. Scale bars, 10 μm.

## Results

**The FRC network is under tension in steady-state LNs.** Because fibroblasts are contractile, force-generating cells[13], we hypothesized that the FRC network determines LN tissue mechanics. The FRC network spans the whole LN and specifically supports CD3⁺ T cell function in the paracortex, providing trafficking routes from high endothelial venules to B cell follicles[14–17] (Fig. 1a,b). We asked how the FRC network and associated extracellular matrix participate in tissue mechanics in the immunological steady state. We used a fibroblast-specific membrane-EGFP mouse model (*Pdgfra*-mGFP-CreERT2) (Fig. 1c and Extended Data Fig. 1) to visualize

the fine cellular connections through the FRC network (Fig. 1c). Following laser ablation (Fig. 1c,d, Extended Data Fig. 1e, and Supplementary Video 1), we measured a mean recoil of 0.42 μm s⁻¹ in the FRC network of the paracortex, formally demonstrating that the reticular network is under mechanical tension (Fig. 1c,d). The severed network recoiled in all directions adjacent to the ablation site, suggesting that mechanical forces are buffered throughout the network. To investigate how cellular-scale stromal components contributed to tissue mechanics, we examined the cytoskeletal and extracellular matrix structures in the reticular network. In steady state, FRCs adhere to and enwrap the bundled collagen of the underlying

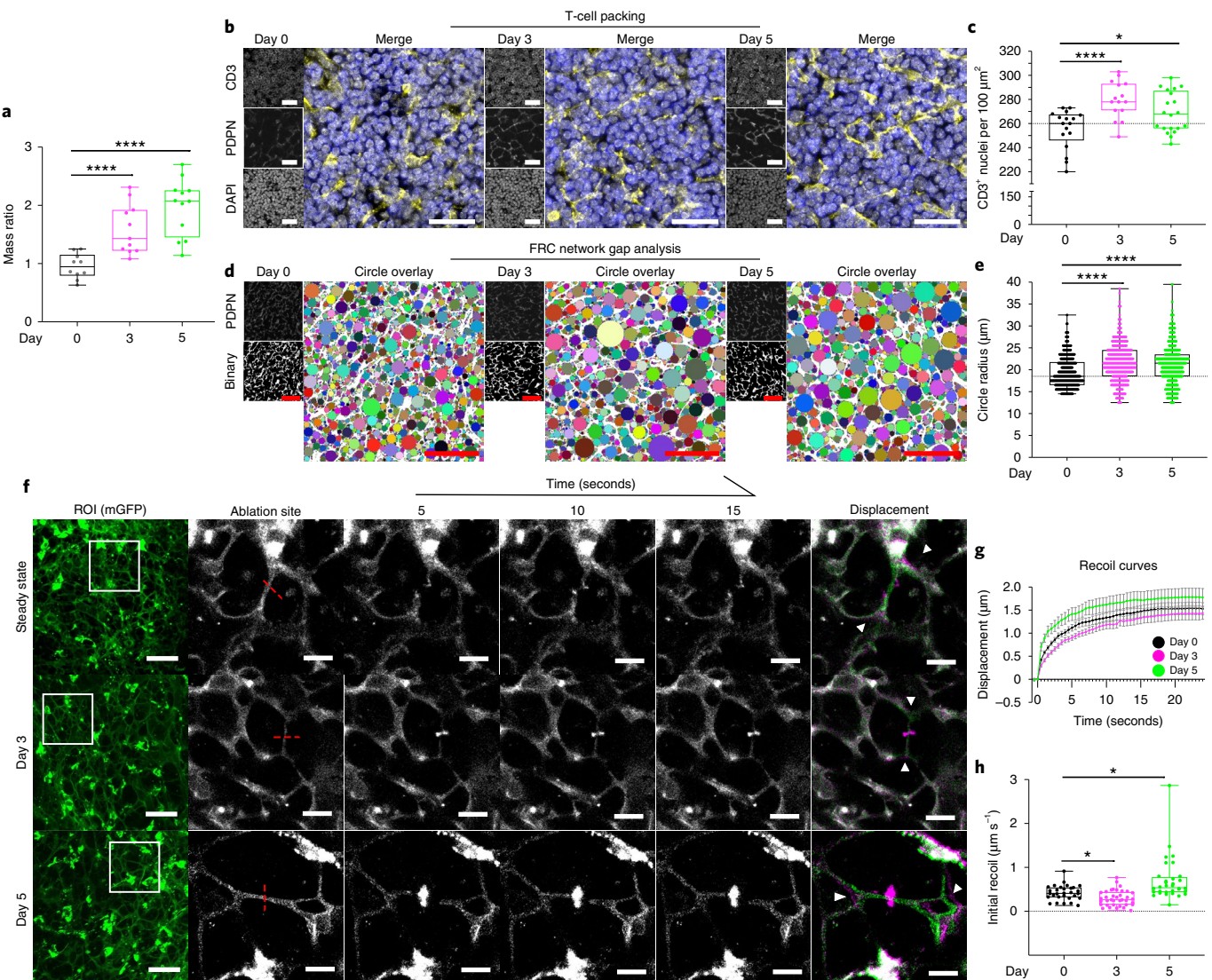

**Fig. 2 | FRCs adapt to changing external forces throughout immune challenge. a**, LN mass change after IFA/OVA immunization. One-way analysis of variance (ANOVA) with Tukey's multiple comparisons, ****$P = 1.00 \times 10^{-6}$. Each point represents one LN at day 0 ($n = 10$), day 3 ($n = 11$), and day 5 ($n = 12$). **b**, T cell packing in the LN during inflammation for PDPN (FRCs, yellow), CD3 (T cells, blue), and DAPI (nuclei, gray). Scale bars, 25 μm. **c**, Quantification of CD3+ T cell nuclei per 100 μm². Box plot indicates median, interquartile range, and minimum/maximum. Dotted line represents median of day 0. One-way ANOVA with Tukey's multiple comparisons, *$P = 0.0362$, ****$P = 0.0003$. $n$ indicates individual image ROI on day 0 ($n = 17$), day 3 ($n = 15$), and day 5 ($n = 20$) from three independent LNs. **d**, FRC gap analysis during inflammation. PDPN (FRCs), binary, and circle overlay. Scale bars, 50 μm. **e**, Quantification of circle radius. Box plot indicates median, interquartile range, and minimum/maximum. Dotted line represents median of day 0. One-way ANOVA with Tukey's multiple comparisons, ****$P = 2.00 \times 10^{-6}$. Each point represents a circle radius. $n$ indicates individual image ROI on day 0 ($n = 12$), day 3 ($n = 15$), and day 5 ($n = 15$) from three independent LNs. **f**, Laser ablation of the FRC network throughout inflammation. PDGFRα+mGFP+ (FRCs) ablation ROI (white box) and cut site (red dotted line). Scale bars, 50 μm. Recoil displacement (arrowheads) with pre- (green) and postcut (magenta) overlay. Scale bars, 10 μm. **g**, Recoil curves of network displacement (μm) (mean ± standard error of the mean (s.e.m.)). **h**, Initial recoil velocity (μm s⁻¹) after IFA/OVA immunization. Box plot indicates median, interquartile range, and minimum/maximum. Kruskal–Wallis test with Dunnett's test, *$P < 0.05$. **g,h**, $n$ indicates an ablation on day 0 ($n = 30$), day 3 ($n = 37$), and day 5 ($n = 32$) for three independent experiments.

conduit[18]. Therefore, recoil following laser ablation is a combined mechanical measurement of the cell and the extracellular matrix structures (Fig. 1c–f). Maximum z-projections and orthogonal views of FRCs show F-actin cables aligned proximal to the conduit and beneath the T cell-facing FRC plasma membrane (Fig. 1f and Supplementary Video 2). These F-actin structures colocalized with phosphorylated myosin regulatory light chain (pMLC2)[19] (Fig. 1f), indicating that FRCs generate contractility and strain in steady state[20].

**FRCs adapt to external forces throughout immune challenge.** We asked how the FRC network reacts to forces exerted by lymphocytes

recruited to and proliferating in the tissue following immune challenge. Immunization with incomplete Freund's adjuvant (IFA) and ovalbumin (OVA) causes LNs to expand two- to threefold over the first 5 days (Fig. 2a and Extended Data Fig. 1f). As a proxy measurement of external forces exerted, we quantified T cell packing and the spacing of network fibers through acute LN expansion. We found that after immunization, the density of CD3+ cells within tissue increased just 3 days after immunization, and we quantified 10–20% more T cells per 100 μm² (Fig. 2b,c). This increase in T cell packing density was maintained at day 5 (Fig. 2b,c), suggesting that the FRC network would experience higher external forces following

immune challenge. We next quantified the spacing of the FRC network fibers using our gap analysis algorithm[3]. Consistent with previous observations, we find that the spaces between network fibers were increased as the FRC network stretches (Fig. 2d,e) and that FRC network integrity and connectivity remains intact.

If the FRC network balances mechanical forces in LNs, then we would expect tension in the FRC network to change throughout LN expansion. Surprisingly, at day 3 after immunization, initial recoil velocity decreased by 29% to 0.30 μm s⁻¹ (Fig. 2f–h and Supplementary Videos 3 and 4) despite a 1.5-fold increase in tissue mass (Fig. 2a) and increased packing of T cells (Fig. 2b,c). In contrast, at day 5 after immunization, mean initial recoil velocity was 60% higher than in the steady state at 0.71 μm s⁻¹ (Fig. 2f–h and Supplementary Videos 3 and 5). Therefore, FRC network tension did not correlate with either T cell packing density or network stretching, indicating that tissue tension is not solely determined by external forces and that cell-intrinsic factors must also impact FRC network mechanics.

**Actomyosin contractility sets FRC network tension.** We next sought to determine the cellular determinants of FRC network tension. Laser ablation methods cannot discriminate between cellular and extracellular matrix components[21,22], as the FRC network is tightly associated with the underlying conduit[18,23,24]. We therefore examined the cytoskeletal structures in FRCs within the tissue context and their relationship to the underlying matrix through acute immune challenge. We compared cytoskeletal and matrix structures of FRCs at day 3 (Fig. 3a, lower tension) versus day 5 (Fig. 3b, higher tension). The majority (75–85%) of F-actin structures were located proximal to the basement membrane (asterisks, Fig. 3a–c, and Supplementary Video 6). However, 5 days after immunization, contractile F-actin cables structures span FRC cell bodies in the absence of underlying matrix bundles (arrowhead) (Fig. 3b–d and Supplementary Video 7). We have previously reported that during acute LN expansion, the conduit network becomes fragmented[9], whereas the cellular network remains connected[3,4,11]. We reported that polarized microtubule networks via LL5β control FRC-matrix deposition during early LN expansion[9]. Laminins, collagens, and basement membrane components become fragmented as LNs expand causing disruption of conduit flow[9]. We compared the conduit matrix using second harmonic signals at day 3 and day 5 after immune challenge (Extended Data Fig. 2) and found that disruption in the conduit occurred at day 5, correlating with increased tension. We measured increased fibril thickness in areas of disruption at day 5 (Extended Data Fig. 2) and that fibers appeared straighter in draining LNs, consistent with fibers being pulled taught before breaking. To determine whether FRCs formed connections in the absence of underlying

matrix, we genetically labeled individual FRCs via sporadic Cre-recombinase activation in *Pdgfra*-mGFP-CreERT2 mice to express the R26R-Confetti[25] conditional allele (*Pdgfra*^iR26R-Confetti^). We observed individual FRCs spanning sections of disrupted conduit (Fig. 3d), suggesting that the matrix structure may not contribute to the increased network tension at day 5 (Fig. 2f–h) and that the intact cellular network balances tissue tension. Contractility and tension through actin filaments is regulated through interactions with myosin[26,27]. We quantified myosin light chain 2 phosphorylation (T18, S19), within PDPN⁺ structures, to determine contractile cables in FRCs stained positive for pMLC (Fig. 3e). However, at day 3, pMLC⁺ F-actin cables were reduced and in some areas absent (Fig. 3e). The pattern of reduced actomyosin contractility in FRCs at day 3 mirrored the reduced tissue tension measured at the same time point (Fig. 2f–h). We therefore hypothesized that the actomyosin cytoskeleton is a major contributor to FRC network tension. We tested this pharmacologically using the ROCK inhibitor Y27632 to inhibit phosphorylation of myosin light chain. Pre-treatment of LNs with ROCK inhibitor reduced tissue tension to basal levels in steady state and following immunization (Fig. 3f,g and Extended Data Fig. 3a–d). Following ROCK inhibitor treatment (Fig. 3h), FRCs remained connected and F-actin cables were visible (arrowheads, Fig. 3h,i), but the proportion of contractile pMLC⁺ F-actin cables in both steady-state and immunized LNs was reduced (Fig. 3j), confirming the correlation between actomyosin contractility and FRC network tension.

These data show that mechanical forces are generated by increased packing of lymphocytes in the FRC meshwork and resisted by actomyosin through the FRC network (Fig. 2). However, there is not a simple linear relationship between the increases in external forces caused by T cell packing and tissue tension measured through the FRC network. Tissue tension varies throughout responses to immunogenic challenge, and we find that increased tissue tension at day 5 occurs independently of extracellular matrix integrity. We have previously reported that 5 days after immunization, the whole, intact LN is more deformable[3], yet we now measure increased tension through the fibroblastic reticular network increases (Fig. 2f–h). Because the elasticity of gels is known to scale as the inverse of mesh size[28], the combination of the fragmentation of the matrix (Fig. 3c,d) and increasing mesh size (Fig. 2d,e) of the FRC network may explain increased LN deformability through acute expansion.

**CLEC-2/PDPN signaling controls intrinsic FRC mechanics.** Because the FRC network did not resist increased packing of lymphocytes at day 3 (Fig. 2b,c), actomyosin contractility alone cannot explain how the FRC network remains connected through LN expansion. We sought to understand the cellular scale mechanical

**Fig. 3 | Actomyosin contractility sets FRC network tension in response to tissue expansion. a,b,** Actomyosin structure in FRCs day 3 (a) and day 5 (b) after IFA/OVA immunization for perlecan (matrix, magenta), PDPN (FRC, yellow), phalloidin (F-actin, cyan), and pMLC (red). Asterisks and arrowheads indicate F-actin cables with or without perlecan alignment. Orthogonal views (yellow dotted line, *yz* axis, 10 μm depth). Representative image of three independent experiments. Scale bars, 10 μm. **c,** Percentage of Fibers. Stacked bar plots with mean ± standard deviation (s.d.) error bars. **d,** Confetti-labeled FRCs (yellow) spanning space without (white asterisk) ECM (magenta). Scale bar, 5 μm. **e,** Percentage of Fibers. Box plot indicates median, interquartile range, and minimum/maximum. One-way ANOVA with Tukey's multiple comparisons, **$P < 0.01$. **c,e,** *n* indicates image ROI on day 0 (*n* = 14), day 3 (*n* = 13), and day 5 (*n* = 12) from three mice. **f,** Initial recoil velocity (μm s⁻¹) after IFA/OVA immunization, ± ROCK inhibition (Y27632). Resistive myosin forces (blue text indicates the ratio between PBS/Y27). Box plot indicates median, interquartile range, and minimum/maximum. Two-way ANOVA with Tukey's multiple comparisons, ****$P < 0.0001$, **$P = 0.001973$. **g,** Recoil curves of network displacement (μm) (mean ± s.e.m.). **f,g,** *n* indicates an ablation on day 0 (*n* = 54), day 0 + Y27 (*n* = 43), day 3 (*n* = 18), day 3 + Y27 (*n* = 20), day 5 (*n* = 19), and day 5 + Y27 (*n* = 15) over three independent experiments. **h,** Actomyosin structure in FRCs day 5 after IFA/OVA immunization with or without ROCK inhibition (Y27632). Perlecan (matrix, magenta), PDPN (FRC, yellow), phalloidin (F-actin, cyan), and pMLC (red). Asterisks and arrowheads indicate F-actin cables with or without perlecan alignment. Representative image of three independent experiments. Scale bars, 10 μm. **i,** Percentage of Fibers. Stacked bar plots with mean ± s.d. error bars. Two-way ANOVA with Tukey's multiple comparisons, ****$P = 0.000577$, *$P = 0.03824$. **j,** Percentage of Fibers. Box plot indicates median, interquartile range, and minimum/maximum. One-way ANOVA with Tukey's multiple comparisons, ****$P < 0.0001$. **i,j,** *n* indicates image ROI on day 0 (*n* = 12), day 0 + Y27 (*n* = 12), day 5 (*n* = 18), and day 5 + Y27 (*n* = 18) from three mice. NS, not significant.

adaptation of FRCs to explain how network tension is decreased while T cell packing forces increase. To increase mesh size without increasing FRC number[3,4], FRCs must elongate to maintain network integrity[3,11]. We therefore investigated mechanisms controlling the cell surface mechanics of FRCs. Effective membrane tension (hereafter membrane tension) is determined by the in-plane tension of the lipid bilayer and the strength of membrane-to-cortex attachments[29,30] and can regulate cell changes in morphology and

cell fate[29–31]. Contact between antigen-presenting dendritic cells and FRCs through CLEC-2/PDPN, peaking at day 3, is known to regulate LN deformability and tissue expansion[3,4]. Dendritic cells lacking CLEC-2 were unable to induce LN expansion, and injection of recombinant CLEC-2-Fc protein was sufficient to rescue these defects in LN remodeling[3]. Therefore, we asked whether CLEC-2/PDPN binding regulates FRC cell surface mechanics to explain the reduced tissue tension we measure at day 3. We used

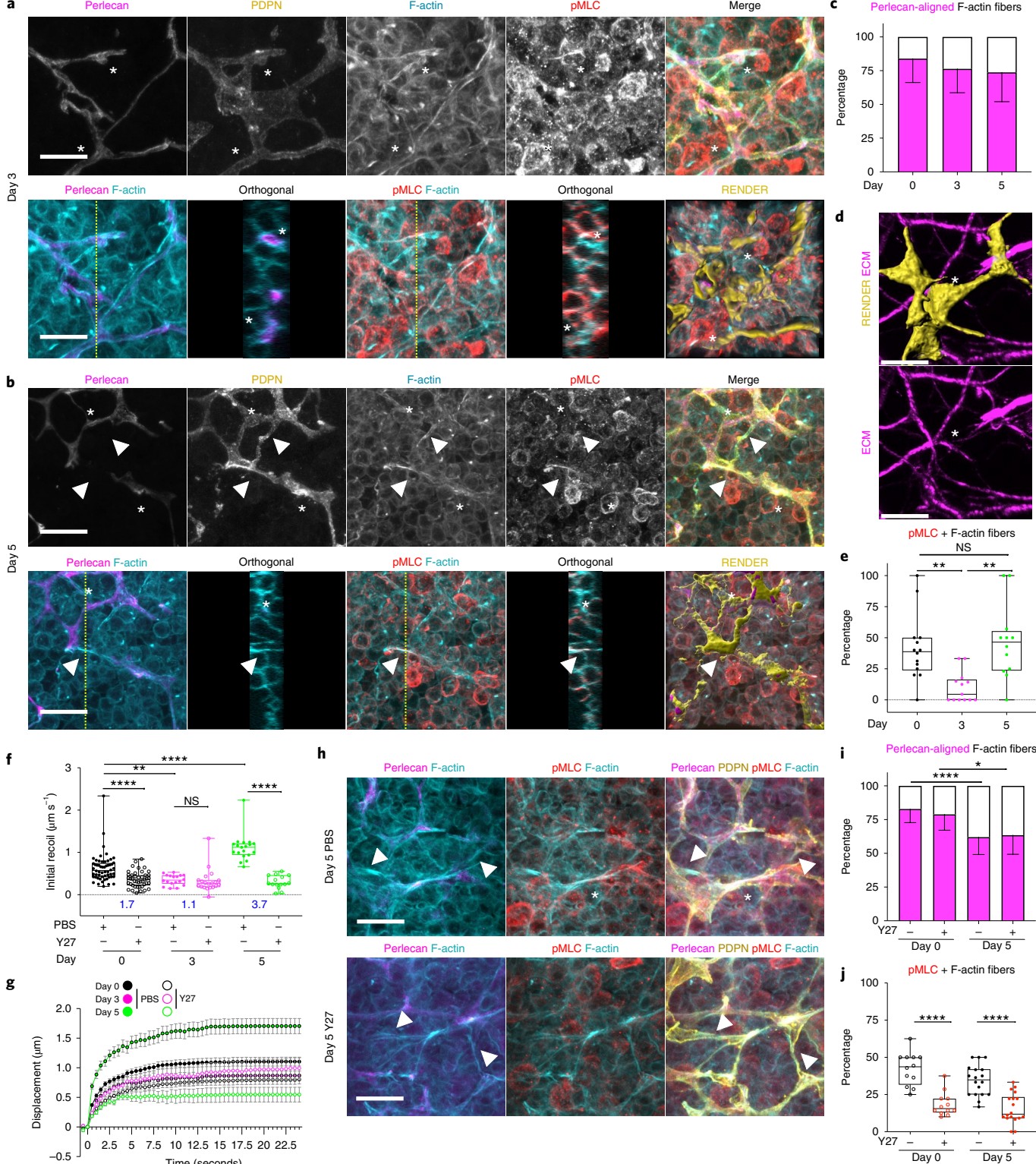

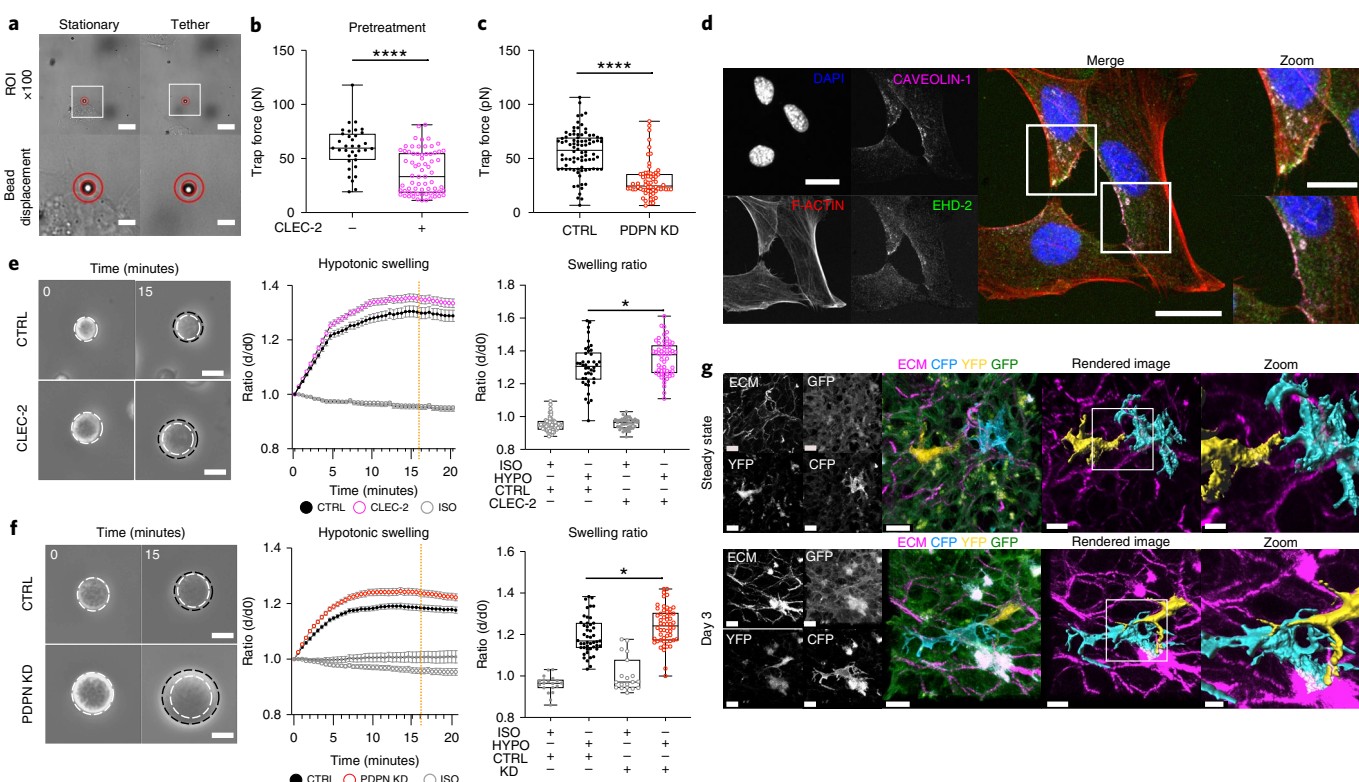

**Fig. 4 | CLEC-2/PDPN signaling controls intrinsic FRC mechanics for morphological adaption. a**, Optical tweezers generate membrane tethers of FRCs; scale bar (10 μm, top panels). Stationery (bottom left) and tether (bottom right) bead displacement (red circles). Scale bar, 2 μm (bottom panels). **b**, Trap force (pN μm⁻¹) of FRCs pretreated with CLEC-2. Box plot indicates median, interquartile range, and minimum/maximum. Mann–Whitney test (two tailed), ****$P = 1.00 \times 10^{-6}$. $n$ indicates cell for CLEC-2⁻ ($n = 35$) and CLEC-2⁺ ($n = 69$) over five independent experiments. **c**, Trap force (pN μm⁻¹) of PDPN short hairpin RNA (shRNA) knockdown (KD) FRC. Box plot indicates median, interquartile range, and minimum/maximum. Mann–Whitney test (two tailed), ****$P = 1.00 \times 10^{-6}$. $n$ indicates cell for control ($n = 86$), PDPN KD ($n = 64$) over five independent experiments. **d**, Caveolae structures in FRCs for DAPI (blue), F-actin (red), Caveolin-1 (magenta), and EHD2 (green). Representative images of two independent experiments. Scale bars, 20 μm (zoom-ins (white box), 10 μm). **e,f**, Swelling of PDPN CTRL FRCs with or without pretreatment CLEC-2 (e) and PDPN shRNA KD FRCs (f). Initial size (left, white circle) after swelling (black circle). Scale bars, 25 μm. Change in diameter ratio (middle, mean ± s.e.m.). Diameter ratio (right) of control and CLEC-2-treated cells (e) or PDPN KD cells (f) at 15 min after swelling (orange dotted line). Box plot indicates median, interquartile range, and minimum/maximum. **e**, One-way ANOVA with Tukey's multiple comparisons, *$P = 0.0277$. $n$ indicates cell for CLEC-2⁻ isotonic control (ISO) ($n = 48$), CLEC-2⁻ hypotonic solution (HYPO) ($n = 56$), CLEC-2⁺ ISO ($n = 55$), and CLEC-2⁺ HYPO ($n = 42$) cells analyzed over five independent experiments. **f**, One-way ANOVA with Tukey's multiple comparisons, *$P = 0.0212$. $n$ indicates cell for control ISO ($n = 19$), control HYPO ($n = 47$), PDPN KD ISO ($n = 19$), and PDPN KD HYPO ($n = 54$) cells analyzed over five independent experiments. **g**, Individually labeled FRCs in vivo. Second harmonic (ECM, magenta), CFP (cyan), YFP (yellow), GFP (green). Representative images of three LNs over two independent experiments. Scale bars, 10 μm (zoomed-in region (white box), 5 μm).

optical tweezers (Fig. 4a, Extended Data Fig. 4g, and Supplementary Video 8) and found that specific engagement of recombinant CLEC-2 to PDPN reduced membrane tension (Fig. 4a,b) and down-regulated phosphorylation of ezrin, radixin, and moesin family proteins (pERM) (Extended Data Fig. 4f), which tether cortical actin to the plasma membrane. Signaling specificity was confirmed by exogenous expression PDPN mutants that cannot bind CLEC-2 (T34A) and cannot signal through the cytoplasmic tail (S167A-S171A) (Extended Data Fig. 4a–d). Neither mutant responded to CLEC-2 (Extended Data Fig. 4a–d) indicating that CLEC-2 regulated membrane tension specifically through binding PDPN and altering PDPN-dependent signaling. As PDPN interacts with CD44[32], which is also known to bind ezrin[33], we tested the relative contributions of CD44 and PDPN to membrane tension in unstimulated FRCs. We found that knockout of CD44 did not affect FRC membrane tension, suggesting that PDPN is the key driver FRC surface mechanics (Fig. 4c and Extended Data Fig. 4e).

FRCs must elongate contact with their network neighbors as they spread and expand the stromal architecture[11,34]. FRC elongation could be achieved by increasing the surface area to cell volume ratio through exocytotic pathways or unfolding membrane reservoirs[35,36]. We found that FRCs contain active EHD2⁺ caveolae structures[37] that could contribute to tension-sensitive plasma membrane reservoirs (Fig. 4d). We tested, using osmotic shock (Extended data Fig. 5a,b), whether regulation of membrane tension through the CLEC-2/PDPN signaling axis impacted how rapidly FRC uses existing membrane reservoirs (Fig. 4e,f). We found that CLEC-2 engagement (Fig. 4e, Extended Data Fig. 5c, and Supplementary Videos 9 and 11) or knockdown of PDPN (PDPN KD) (Fig. 4f and Supplementary Video 10) permitted more rapid cell expansion in hypotonic conditions but did not alter total membrane availability (Extended Data Fig. 5d and Supplementary Video 12). In vivo, we observed differentially labeled FRCs in *Pdgfra*^iR26R-Confetti mice extending cell–cell contacts at day 3 (Fig. 4g), suggesting morphological adaptations for cell extensions. We therefore conclude that CLEC-2/PDPN binding plays a dual role in FRC network remodeling, changing cell surface mechanics of FRCs and reducing actomyosin contractility[3,4] (Fig. 3f), which together permit LN expansion while maintaining FRC network integrity.

**FRCs are mechanically sensitive in vitro.** Existing studies report that FRC proliferation lags behind lymphocytes[3–5], but it is not known how FRC proliferation is triggered or spatially regulated[5]. Five days after IFA/OVA immunization, the number of proliferating FRCs (Ki67[+]) doubled compared to steady state (Fig. 5a,b). Ki67[+] FRCs were sporadically located throughout the paracortex, and we observed no specific proliferative niche surrounding blood vessels or beneath the capsule (Fig. 5a). This finding suggested that entry into the cell cycle is not spatially regulated or limited to an FRC subpopulation. Five days after immunization, pressure from T cells (Fig. 2a–c) increased FRC network tension (Fig. 2f–h), which is resisted by actomyosin contractility (Fig. 3f). We hypothesized that increased mechanical tension may gate FRC proliferation. Because cell responses in tissues are driven by a complex combination of chemical and mechanical cues, we measured FRC proliferation in response to mechanical stimuli alone in vitro using a reductionist system of polyacrylamide gels ranging in stiffness[38] (Fig. 5c). Using automated tracking of FRC nuclei, we found that FRCs proliferated more rapidly on stiffer substrates (Fig. 5c,d), indicating that FRC proliferation is mechanically sensitive. Inhibition of ROCK using Y27632 (Fig. 5e), which blocked increased tissue tension in response to immune challenge in vivo, also blocked the mechanically determined changes to FRC proliferation in vitro (Fig. 5e). Knockout of PDPN expression also attenuated the impact of increased stiffness on FRC proliferation (Fig. 5f). These data show that even in the absence of cell–cell communication with immune cells and soluble chemical cues in the tissue, FRC proliferation is mechanically sensitive and stimulated by stiffer external environments (Fig. 5c,d), sensed by PDPN signaling and actomyosin contractility (Fig. 5e,f).

We have previously reported that PDPN is upregulated in FRCs following immune challenge[32] by CLEC-2 binding[32]. Additionally, PDPN expression in immortalized FRC cell lines, cultured on rigid plastic, is higher than in primary cells[3], but the regulation of PDPN expression remains poorly understood. Because we have measured increased tissue tension during acute LN expansion, which correlates with increasing PDPN expression, we investigated whether PDPN expression might be mechanically regulated. We measured cell surface expression of PDPN on primary FRCs (CD140$\alpha^+$ CD31[−]) using flow cytometry after cell culture on rigid plastic and found that inhibiting actomyosin using Y27632 reduced PDPN expression by approximately 30% after 3 days of culture and more (75%) after 6 days (Fig. 5g,h). These data indicate not only that PDPN is a mechanical sensor in FRCs in vitro but also that PDPN expression is impacted by mechanical forces.

**PDPN deletion blocks FRC mechanical adaptation in vivo.** We next sought to test the role of PDPN in LN mechanics in vivo. Mice with full knockout of PDPN fail to develop LNs and die shortly after birth due to circulatory defects, so PDPN function has not been tested specifically in FRCs[39]. We generated *Pdgfra*-mGFP-CreERT2 × *Pdpn*[fl/fl][40] mice (*Pdgfra*[mGFPΔPDPN]) to conditionally delete PDPN in FRCs (Fig. 6a). FRCs are constitutively labeled with membrane-targeted GFP (Extended Data Fig. 1), which in control mice (*Pdgfra*-mGFP-CreERT2) also express PDPN (Fig. 6b). Reduced PDPN expression was quantified by flow cytometry in

FRCs in steady-state LNs of *Pdgfra*[mGFPΔPDPN] mice (Fig. 6b,c). We saw no gross alterations to FRC network structure 7 days following tamoxifen treatment (Fig. 6b).

We next immunized control and *Pdgfra*[mGFPΔPDPN] mice (Extended Data Fig. 6) and found that mice lacking PDPN[+] FRCs exhibited attenuated acute LN expansion (Fig. 6d), phenocopying CD11c[ΔCLEC2] mice[3]. We examined the FRC network structure in *Pdgfra*[mGFPΔPDPN] reactive LNs. The constitutive expression of membrane-targeted EGFP enabled us to identify FRCs independently of PDPN expression to directly compare PDPN[+] and PDPN[−] FRCs side by side in situ (Fig. 6e). We observed F-actin cables along the underlying conduits, adjacent to the basement membrane (perlecan), in both PDPN[+] (asterisk) and PDPN[−] (arrowhead) FRCs (Fig. 6f), and we observed no clear difference in pMLC staining of F-actin cables in *Pdgfra*[mGFPΔPDPN] LNs in steady state (Fig. 6f). We next tested the role of PDPN in LN mechanics through immune challenge using laser ablation. In contrast to our experiments using ROCK inhibitor (Fig. 2), deletion of PDPN did not alter FRC network tension in the steady state. However, *Pdgfra*[mGFPΔPDPN] LNs did not exhibit lower network tension at day 3 or increased tension at day 5 (Fig. 6g–i and Supplementary Videos 13 and 14), indicating that PDPN is required for the FRC network to adapt to mechanical forces through immune challenge. We conclude from these data that PDPN functions as a mechanical sensor for remodeling of the FRC network in vivo.

**FRC mechanics impacts immune outcomes in vivo.** Because LN expansion is an integral part of adaptive immunity[10,24,41], we asked how LN tissue mechanics impacted immune outcomes. LN expansion was significantly attenuated in *Pdgfra*[mGFPΔPDPN] following immunization with CFA/OVA measured by both tissue mass (Fig. 7a) and cellularity (Fig. 7b). PDPN deletion in FRCs resulted in both fewer stromal and immune cell populations 5 days after immunization (Fig. 7c). Both B cell and T cell populations were reduced (Fig. 7d,e), but the ratio of T cells to B cells was not affected (Fig. 7f). Because FRCs primarily support T cell populations in the paracortex[42], we quantified the T cell subsets (Fig. 7g and Extended Data Fig. 7). We found that naive CD4[+] and naive CD8[+] cells were similarly increased in LNs 5 days after immunization in both control and *Pdgfra*[mGFPΔPDPN] LNs, suggesting that recruitment and trapping of naive lymphocytes (CD62L[+], CD44[−]) from the circulation was unaffected by PDPN deletion in FRCs (Fig. 7h–j). Upregulation of CD25 was not significantly affected (Extended data Fig. 8a), suggesting that antigen presentation was not inhibited. However, approximately 50% fewer CD4[+] and CD8[+] cells expressed CD44 following immunization in *Pdgfra*[mGFPΔPDPN] LNs, and effector memory T cells (CD62L[−], CD44[+]) were specifically constrained (Fig. 7h–j). These data suggest that FRCs in *Pdgfra*[mGFPΔPDPN] LNs lack the ability to adapt and stretch to accommodate increasing T cell numbers and as a result constrain effector T cell populations.

Because deletion of PDPN in FRCs attenuated LN expansion, we also examined the impact of PDPN-dependent mechanical perturbation on stromal populations. We found that blood endothelial cells proliferated similarly in control and *Pdgfra*[mGFPΔPDPN] LNs, suggesting that angiogenesis in the LN is not dependent on mechanical signals in FRCs (Fig. 7k). However, the number of

**Fig. 5 | FRCs are mechanically sensitive in vitro. a**, LN tile scan of PDPN[+] (yellow) and Ki67[+] (magenta) cells. Scale bars, 500 μm (zoom-in T cell area (white box), 50 μm). Arrowheads mark PDPN[+] Ki67[+] FRCs. **b**, Flow cytometric analysis of Ki67[+] percentage of FRCs after IFA/OVA immunization. Two-way ANOVA with Dunnett's test, ****$P = 1.00 \times 10^{-6}$. *n* indicates LNs on day 0 (*n* = 5), day 3 (*n* = 5), and day 5 (*n* = 4). **c**, FRC GFP nuclei are tracked over 72 h on different polyacrylamide gel stiffnesses. Scale bar, 50 μm. **d–f**, Average number of divisions per cell in 72 h and over time on different substrate rigidities. **d**, Control FRCs. **e**, FRCs treated with ROCK inhibition (Y27632). **f**, PDPN KO FRCs. Box plot indicates median, interquartile range, and minimum/maximum. One-way ANOVA with Tukey's multiple comparisons, *$P = 0.0298$. *n* indicates image ROI for 2 kPa (*n* = 8), 12 kPa (*n* ≥ 4), 30 kPa (*n* ≥ 4), and glass (*n* ≥ 5) over three independent experiments. Division curves compare glass and 2 kPa (mean ± s.e.m.). **g**, Flow gating and PDPN histograms geometric mean fluorescent intensity (gMFI) of isolated primary FRCs after ROCK inhibition (Y27632). **h**, Fold change in gMFI of PDPN on FRCs following ROCK inhibition (Y27632). Mann–Whitney test (two tailed), ****$P < 0.0001$. *n* = 3 LNs.

lymphatic endothelial cells was reduced (Fig. 7l). When we examined FRCs, gating on either GFP⁺ or alternatively CD140α⁺ CD31⁻ stroma, we measured approximately 50% fewer FRCs 5 days after

immunization in *Pdgfra^mGFPΔPDPN* LNs (Fig. 7m). The reduction in FRCs could be a cause or a consequence of the attenuated LN expansion (Fig. 7a,b). However, as FRC proliferation is mechanically

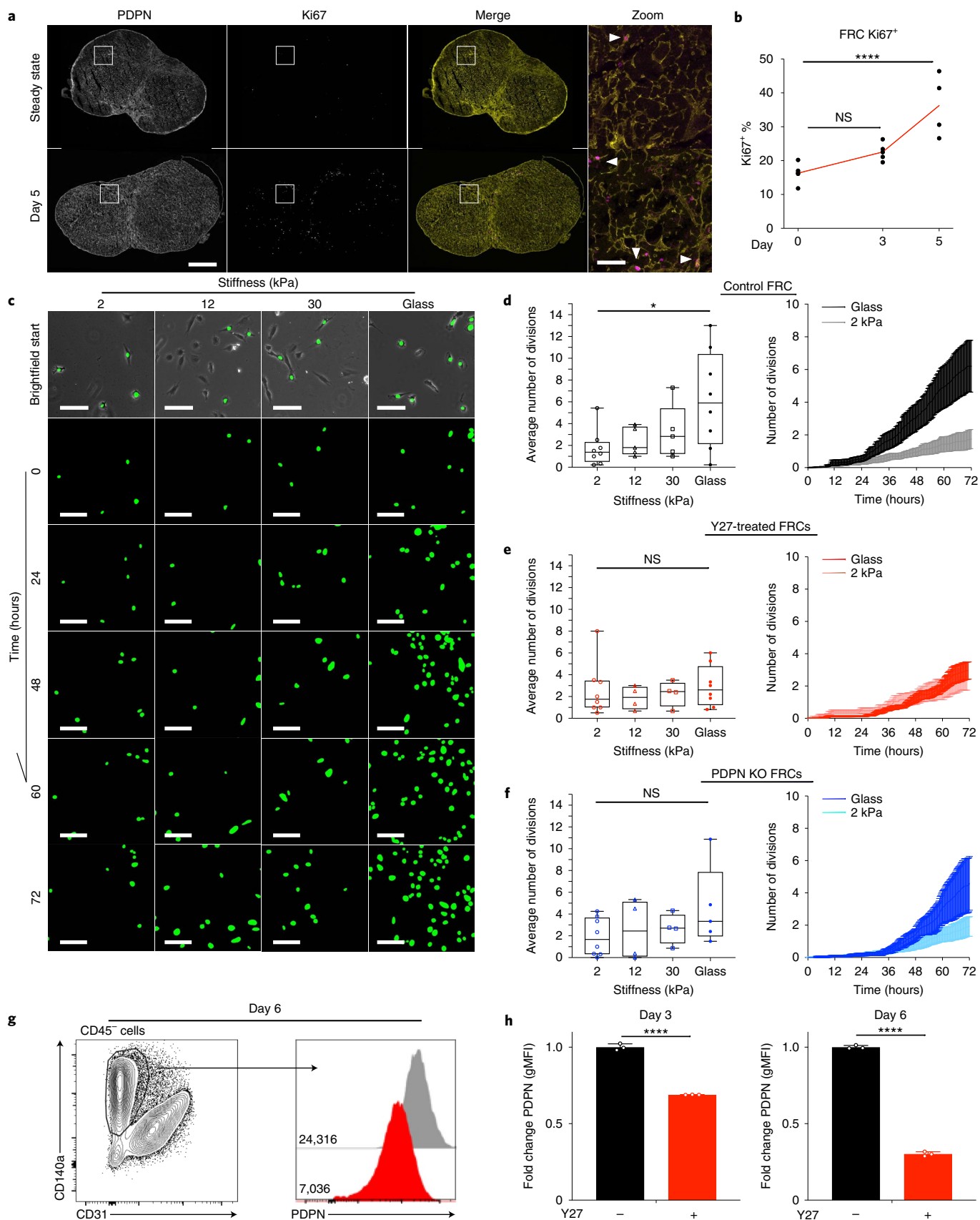

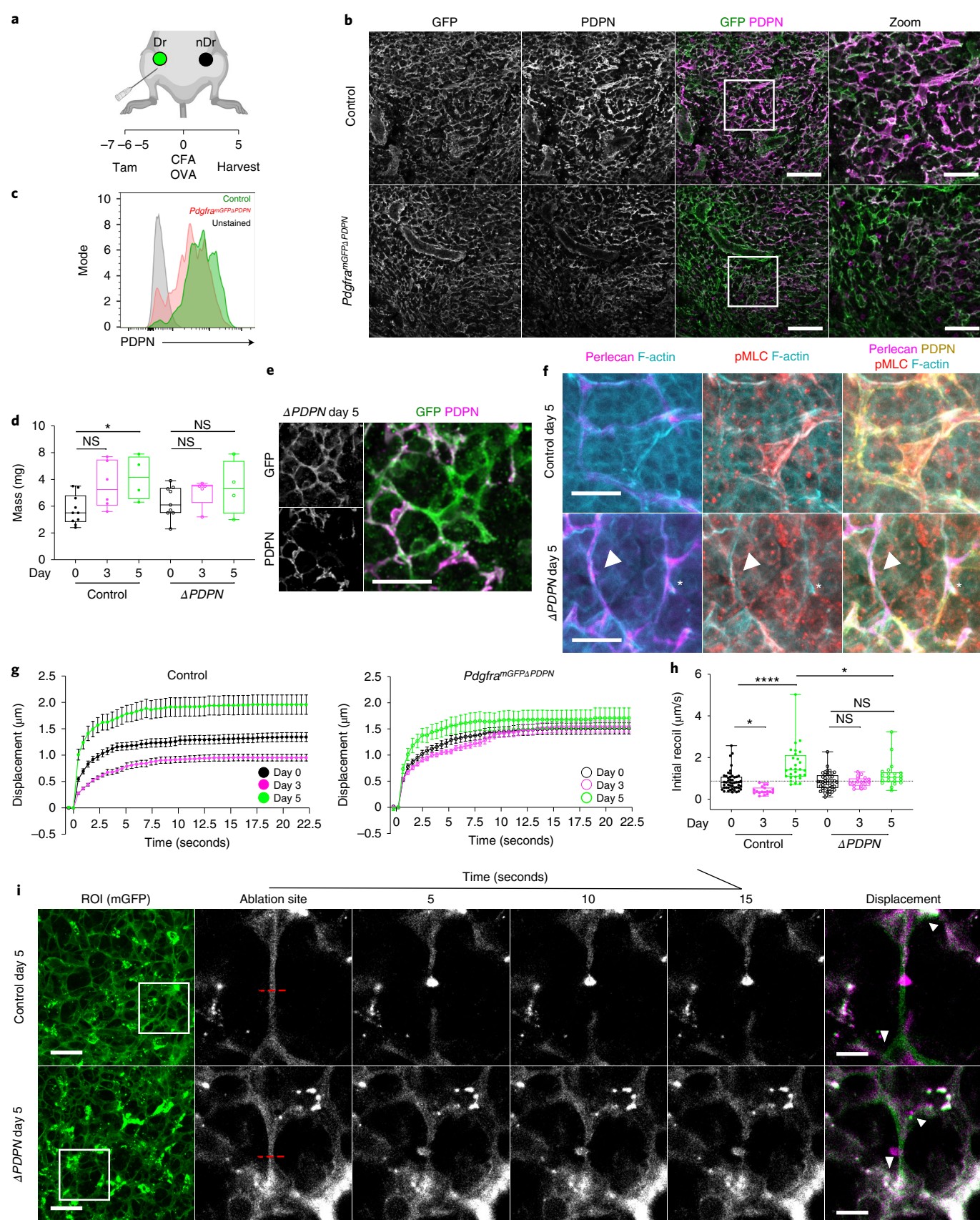

sensitive in vitro (Fig. 5), we hypothesize that reduced FRC network tension in *Pdgfra^{mGFPΔPDPN}* mice (Fig. 6g,h) may inhibit FRC proliferation in vivo.

**Mechanical tension gates FRC proliferation in vivo.** We have shown that PDPN is a mechanical sensor in FRCs impacting cell-intrinsic mechanics and morphological adaptation in response

**Fig. 6 | Deletion of PDPN in vivo attenuates mechanical adaptation of the FRC network in response to immune challenge. a**, Tamoxifen and immunization strategy for *Pdgfra*[mGFPΔPDPN] mice. Draining LNs (Dr) and non-draining LNs (nDr). **b**, LN paracortex maximum z-projection. GFP (*Pdgfra*[mGFP], green), PDPN (FRCs, magenta). Scale bars, 50 μm (zoom, 30 μm). **c**, Representative histograms of surface protein expression for PDPN in control and *Pdgfra*[mGFPΔPDPN] steady-state LNs. **d**, LN mass after CFA/OVA immunization. Box plot indicates median, interquartile range, and minimum/maximum. Two-way ANOVA with Dunnett's test, *$P = 0.030$. *n* indicates LNs on day 0 ($n = 10$), day 0[ΔPDPN] ($n = 9$), day 3 ($n = 6$), day 3[ΔPDPN] ($n = 5$), day 5 ($n = 4$), and day 5[ΔPDPN] ($n = 4$). **e**, Representative image of GFP[+] (green), PDPN[−] (magenta) FRC cell body in day 5 after immunization in *Pdgfra*[mGFPΔPDPN] LNs from two independent experiments. Scale bar, 25 μm. **f**, Representative images of actomyosin and ECM structures within control and *Pdgfra*[mGFPΔPDPN] FRCs 5 days after immunization. Arrowheads and asterisks mark PDPN[−] and PDPN[+] FRCs respectively; perlecan (magenta), PDPN (yellow), F-actin (cyan), and pMLC (red). Scale bars, 10 μm. **g**, Recoil curves of network displacement (μm) (mean ± SEM) for control and *Pdgfra*[mGFPΔPDPN] mice. **h**, Initial recoil velocity (μm s[−1]) after CFA/OVA immunization in control and *Pdgfra*[mGFPΔPDPN] mice. Box plot indicates median, interquartile range, and minimum/maximum. Two-way ANOVA with Sidak's multiple comparisons, *$P < 0.05$, ****$P = 1.00E^{−6}$. Each point represents an ablation. **i**, Laser ablation of the FRC network throughout inflammation in control and *Pdgfra*[mGFPΔPDPN] mice. PDGFRα[+]mGFP[+] (FRCs) ablation ROI (white box) and cut site (red dotted line). Scale bars, 50 μm. Recoil displacement (arrowheads) with pre- (green) and postcut (magenta) overlay. Scale bars, 10 μm. **g–i**, *n* indicates ablation at day 0 ($n = 48$), day 0[ΔPDPN] ($n = 44$), day 3 ($n = 18$), day 3[ΔPDPN] ($n = 18$), day 5 ($n = 28$), and day 5[ΔPDPN] ($n = 21$) over three independent experiments.

to immune challenge (Figs. 4 and 5). However, the CLEC-2/PDPN signaling axis regulates an array of immunologically relevant pathways in FRCs in addition to cell mechanics, dendritic cell migration[43], ECM deposition[9], and high endothelial venule function[16], and we cannot definitively conclude that the impact on immune outcome we measure in *Pdgfra*[mGFPΔPDPN] LNs results solely from mechanical perturbation. Therefore, to examine the specific impact of FRC network mechanics on adaptive immune function, we manipulated tissue tension through pharmacological inhibition of actomyosin contractility. We reduced tissue tension in vivo through pharmacological inhibition of ROCK[19] in wild-type (C57BL/6 J) mice (Fig. 8a). We found that in contrast to *Pdgfra*[mGFPΔPDPN] mice, increases in tissue mass and cellularity were unaffected by ROCK inhibition (Fig. 8b). Total immune cell numbers (CD45[+]) were unaffected, but there was a trend toward a reduction in stromal populations at day 5 after immunization (Fig. 8c). B cell and T cell numbers were similar between PBS-treated controls and Y27632-treated samples, and T cell activation was also unaffected (Fig. 8d,e and Extended Data Fig. 9a,b). We examined the expression of Ki67 in T cells within the tissue context and confirmed that the initiation of T cell proliferation was not affected by actomyosin inhibition (Extended Data Fig. 8b,c). However, we found that inhibition of actomyosin selectively constrained T cell proliferation, changing the T cell to B cell ratio in reactive LNs (Fig. 8f). Proliferation of endothelial cells was unaffected by actomyosin inhibition, whereas the FRCs were significantly reduced 5 days after immunization (Fig. 8g and Extended Data Fig. 9c,d). This leads us to conclude that the fibroblastic stromal architecture is reactive to the physical space requirements of lymphocytes. Indeed, previous studies identified a robust ratio between fibroblastic stroma and T cells, maintained as the tissue expands[5]. A mechanical cue for stromal cell proliferation would maintain the steady-state ratio of fibroblastic stroma to lymphocytes independently of the kinetics or scale of the immune reaction, ensuring a supportive immune microenvironment for lymphocyte populations[42]. We examined the relationship between FRC network tension and FRC proliferation in steady-state and reactive LNs and how these were altered by mechanical pathways (Fig. 8h). We found that higher network tension coincides with higher FRC number and that this association can be disrupted by either inhibition of actomyosin contractility (Fig. 3) or deletion of PDPN from FRCs (Fig. 6). Together, our data demonstrate that the fibroblastic structure of the LN is the active mechanical component during tissue expansion. Using the dynamic cellular network rather than the more rigid ECM to respond to changing lymphocytes numbers in the tissue is an elegant mechanical system that can proportionately respond to lymphocyte requirements.

## Discussion
Other studies have shown that tissue scale properties emerge from cellular scale mechanics in the transition from developing to adult tissues[44]. We have directly addressed this concept in immunologically relevant adult mammalian tissue during homeostasis and immune challenge. We show that the fibroblastic reticular cellular network deploys molecular signals controlling cellular mechanical properties to collectively determine tissue scale mechanics of LNs.

We have quantified tissue tension through the FRC network. Because by definition forces must be balanced in steady state[2], these data gave us an indirect measurement of the forces exerted by lymphocytes. Upon immune challenge, LNs expand to accommodate increasing numbers of lymphocytes, first trapped from the circulation and then proliferating in response to antigen-specific activation. We quantified increased T cell packing and FRC network spacing during LN expansion, but unexpectedly, these increases did not correlate with increased tension through the FRC network. The uncoupling of tissue size with tissue tension at day 3 after immunization provided evidence that FRC network tension is not solely determined by the external forces. Rather, FRCs can actively and intrinsically adapt their cellular-scale mechanics, and we show that the LN becomes mechanically permissive to expansion through PDPN signaling.

**Fig. 7 | Loss of FRC mechanical sensitivity constrains effector T cell populations in vivo. a**, LN mass. **b**, Cellularity. Box plots indicate median, interquartile range, and minimum/maximum. One-way ANOVA with Tukey's multiple comparisons, ****$P = 1.00 \times 10^{−6}$, ***$P = 0.000567$, **$P = 0.002087$. *n* indicates LNs at day 0 ($n = 12$), day 0[ΔPDPN] ($n = 12$), day 5 ($n \geq 6$), and day 5[ΔPDPN] ($n = 8$) over two independent experiments. **c–e**, CD45[+], CD45[−], CD19[+] and CD3[+] cell numbers. Box plots indicate median, interquartile range, and minimum/maximum. One-way ANOVA with Tukey's multiple comparisons, ****$P = 1.00 \times 10^{−6}$, ***$P < 0.001$, **$P = 0.00989$, *$P = 0.027535$. **f**, Ratio of CD19[+] and CD3[+] cell numbers. Box plots indicate median, interquartile range, and minimum/maximum. One-way ANOVA with Tukey's multiple comparisons, ****$P = 1.00 \times 10^{−6}$. Each point represents one LN. **g**, Flow cytometric gating. Representative dot plots and percentages of CD45[+], CD45[−], CD19[+], CD3[+] and CD3[+] CD4[+] and CD3[+] CD8[+] subpopulations. **h**, Representative flow cytometric gating comparing control and *Pdgfra*[mGFPΔPDPN] CD3[+] CD4[+], CD3[+] CD8[+] subpopulations. **i,j**, CD3[+] CD4[+] and CD3[+] CD8[+] subpopulation cell numbers. Box plots indicate median, interquartile range, and minimum/maximum. One-way ANOVA with Tukey's multiple comparisons, ****$P = 1.00 \times 10^{−6}$, **$P = 0.0058$, *$P < 0.05$. Each point represents one LN. **k–m**, Stromal cell numbers for blood endothelial cells (BECs) (k), lymphatic endothelial cells (LECs) (l), and GFP[+] and CD140α[+] FRCs (m). Box plots indicate median, interquartile range, and minimum/maximum. One-way ANOVA with Tukey's multiple comparisons, ****$P = 1.00 \times 10^{−6}$, **$P = 0.0043$, *$P < 0.05$.

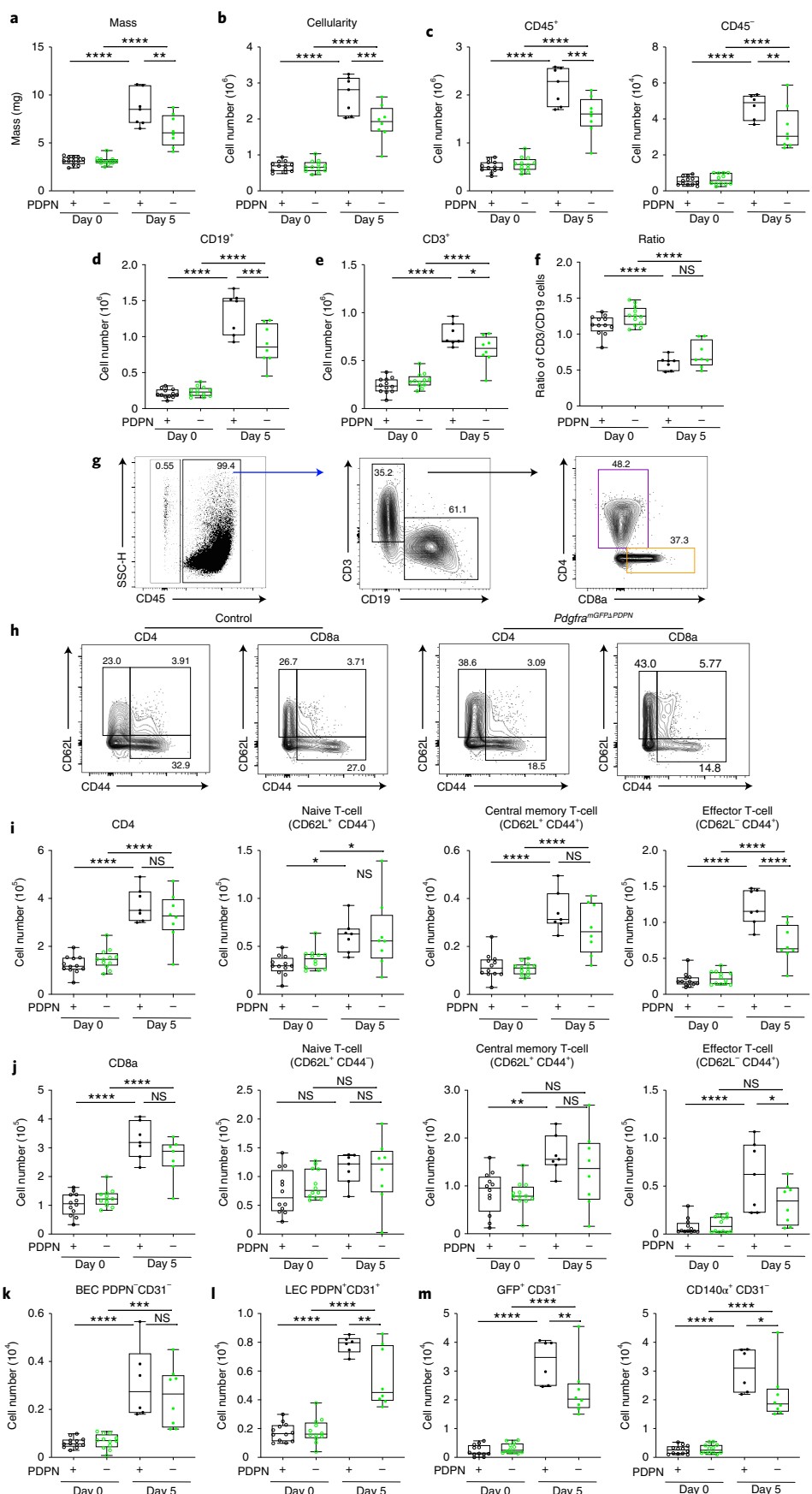

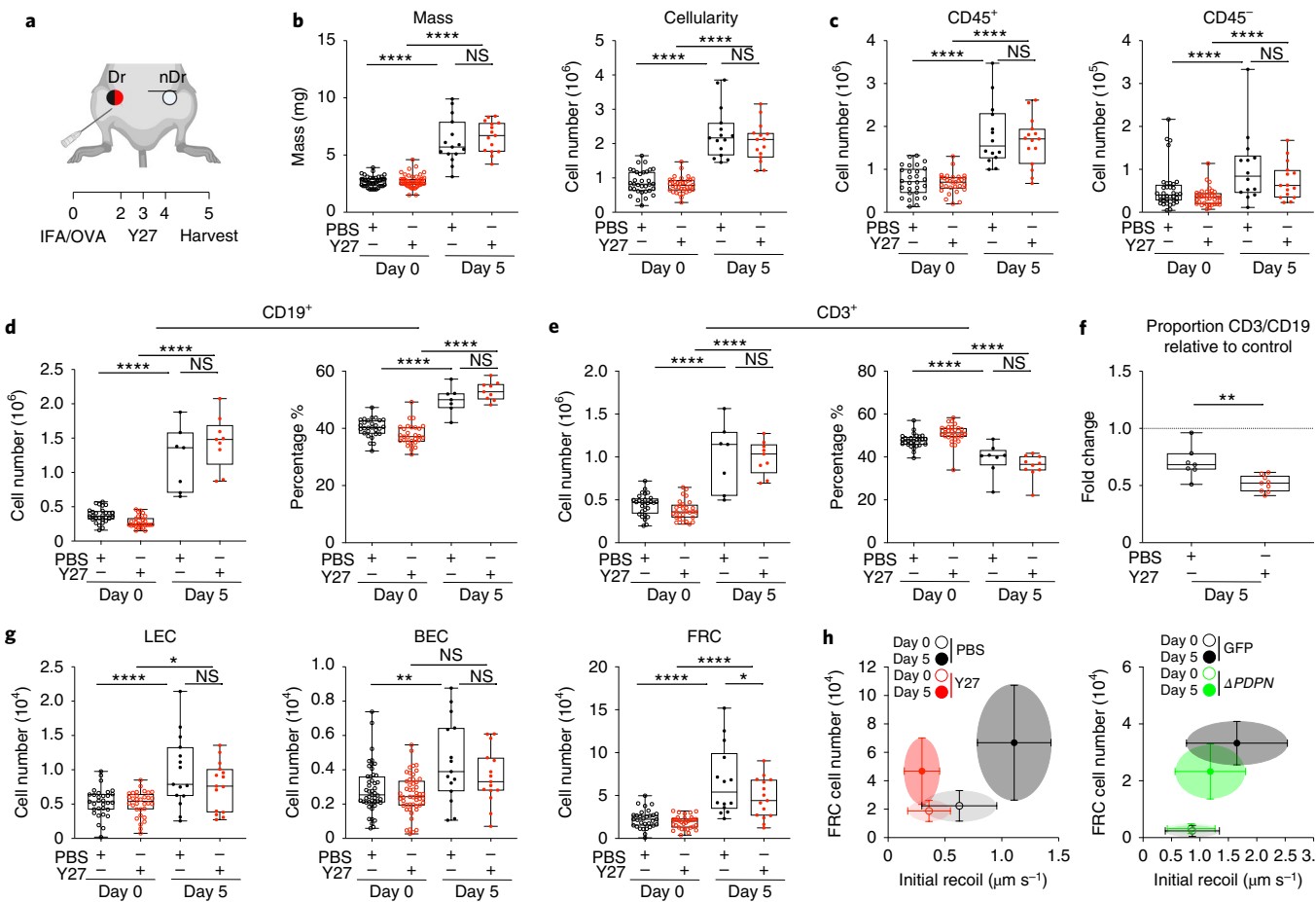

**Fig. 8 | Mechanical tension gates FRC proliferation in vivo. a**, Schematic of immunization and Y27 treatment timeline. Draining LNs (Dr) and non-draining LNs (nDr). **b**, LN mass and cellularity. Box plots indicate median, interquartile range, and minimum/maximum. One-way ANOVA with Tukey's multiple comparisons, ****$P = 1.00 \times 10^{-6}$. $n$ indicates LNs on day 0 ($n \geq 28$), day $0^{Y27}$ ($n \geq 28$), day 5 ($n \geq 7$), and day $5^{Y27}$($n \geq 9$) over two independent experiments. **c**, CD45+ and CD45− cell numbers. Box plots indicate median, interquartile range, and minimum/maximum. One-way ANOVA with Tukey's multiple comparisons, ****$P = 1.00 \times 10^{-6}$. **d,e**, CD19+ and CD3+ cell numbers and percentages of live. One-way ANOVA with Tukey's multiple comparisons, ****$P = 1.00 \times 10^{-6}$. **f**, Ratio of the percentage of CD19+ and CD3+ compared to day 0 steady state. Box plots indicate median, interquartile range, and minimum/maximum. Dotted line marks control. Mann–Whitney test (two tailed), **$P = 0.0033$. **g**, Total number of LECs (left), BECs (middle), and FRCs (right) after IFA/OVA immunization, ± ROCK inhibition (Y27632). Box plot indicates median, interquartile range, and minimum/maximum. One-way ANOVA with Tukey's multiple comparisons, ****$P = 1.00 \times 10^{-6}$, **$P = 0.0016$, *$P < 0.05$. **h**, Scatter plots of FRC numbers against initial recoil velocity. Each dot represents the mean ± s.d. for both $Y27$ (Figs. 8g and 3f) and $\Delta PDPN$ (Figs. 7m and 6h) cell number and initial recoil velocity.

Through this study, we addressed the relative contributions of inert extracellular matrix versus cell structures to tissue mechanics. Because the extracellular matrix of the LN conduits become disrupted through acute LN expansion[9], leaving only the cellular network intact, we are able to conclude that the cellular structures are sufficient to resist and remodel in response to the forces of LN expansion. It is a common assumption that the matrix structures of tissues provide physical guidance for cellular organization[2]. Here, we question that assumption and show that the cytoskeleton and cell–cell connections are sufficient to maintain and remodel LNs during tissue expansion. Cell–matrix adhesion in other tissue contexts is reinforced via forces through the cytoskeleton[45]. We conclude that reduced FRC-matrix adhesion would occur during the acute phase of LN expansion as a consequence of reduced FRC contractility. Indeed, we have previously reported that inhibition of actomyosin contractility in FRCs reduces focal complexes and impacts the tethering of microtubules to sites of cell–matrix adhesion[9]. In future studies, we should now consider the nature of the cell–cell junctions in the FRC network, as these may also play a key role in maintaining FRC network integrity and contribute to tissue mechanics.

Our study provides evidence that FRC proliferation in vivo is mechanically sensitive. A mechanical cue resolves the conflicting reported kinetics[3–5] of FRC proliferation observed with different adjuvants, as an immune challenge causing rapid increases in lymphocyte numbers would increase FRC network tension sooner and induce FRC proliferation earlier. A mechanical trigger for proliferation is also consistent with our observations that there is no specific niche of proliferative FRCs around blood vessels or under the capsule; instead, proliferative FRCs were seen sporadically throughout the FRC network. A mechanical mechanism is an ideal measurement system to ensure that the ratio of FRCs to lymphocytes is maintained independently of LN size. What remains unresolved is how neighboring FRCs maintain connections and how forces might be transmitted through these unstudied cell junctions.

We have directly compared two different mechanical perturbations. We find that both ROCK inhibition and PDPN deletion alter FRC proliferation in vitro, and both pathways are required for FRCs to sense the rigidity of their environment. In tissues, inhibition of actomyosin reduced FRC network tension and therefore prevented the increase in tissue tension required to initiate FRC

proliferation. Deletion of PDPN on the other hand did not alter steady-state tissue tension. Instead, we find that PDPN expression by FRCs is required to adapt FRC cell-intrinsic mechanics, leading us to conclude that PDPN is a key mechanical sensor. Targeting FRC network mechanics through either actomyosin contractility or PDPN expression attenuated FRC proliferation in vivo. However, when we compared the impact of these mechanical perturbations on T cell activation, we found that only PDPN deletion impacted LN expansion and T cell activation and constrained T cell proliferation. We suggest that by inhibiting actomyosin contractility directly, we are able to permit sufficient stretching of the FRC network to allow space for lymphocyte proliferation. In contrast, PDPN is required for FRCs to adapt and stretch, and failure to do so constrains LN expansion. Because it is known that PDPN has additional functions in FRCs (as a ligand for dendritic cell trafficking[43], roles in maintaining high endothelial venule integrity through crosstalk with platelets[16], and as a binding partner for the key chemokine CCL21[46]), our in vivo studies here are intentionally short-term to specifically test the function of PDPN in tissue mechanics. In future studies, it will be interesting to test the role of PDPN in longer-term assays and ask what other tissue functions require this key signaling molecule. It will also be important to examine the role of PDPN in lymphoid tissue function in disease. There are several missense mutations reported in PDPN in patients with diffuse large B cell lymphoma (https://www.cbioportal.org), and it is possible that PDPN expression levels and function are relevant to other pathological states.

In summary, this study further highlights the essential role for fibroblastic stroma in lymphoid tissue homeostasis[41]. Beyond their known roles in lymphocyte trafficking and production of growth and survival factors for lymphocyte activation, we now show that FRCs are also key cellular mechanical sensors. How external mechanical forces cooperate with cell intrinsic physical properties to control cell and tissue function in vivo is a relatively new research field[47]. We address this paradigm to show that FRCs can change their cell-intrinsic mechanical properties in response to changing external forces, allowing the tissue to maintain structure and function through rapid expansion in response to immune challenge. This brings together molecular and cell biology with biophysics approaches to provide mechanistic insights into lymphoid tissue remodeling and integrates tissue mechanics into our understanding of immune function.

## Online content

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

## Methods

**Mice.** Experiments were performed in accordance with national and institutional guidelines for animal care and approved for the Laboratory for Molecular and Cell Biology by the institutional animal ethics committee review board, European research council, and the UK Home office. Breeding of animal lines were maintained off site at Charles River Laboratory. Wild-type C57BL/6J mice were purchased from Charles River Laboratories. Generation of novel mouse model of *Pdgfra*-mGFP-CreERT2 was designed as follows: EGFP with a membrane tag (N-terminal 0–20 amino acids of neuromodulin GAP-43) was inserted into the *Pdgfra* (*CD140α*) gene locus in combination with a CreERT2 cassette (linked to mGFP with a P2A self-cleavage peptide[48]) using Cyagen CRISPR-Cas9 technology. *Pdgfra*-mGFP-CreERT2 mice were crossed with R26R-Confetti[25] or *Pdpn*fl/fl animals[40] to generate *Pdgfra*-mGFP-CreERT2 x R26R-Confetti (*Pdgfra*iR26R-Confetti) and *Pdgfra*-mGFP-CreERT2 x *Pdpn*fl/fl (*Pdgfra*mGFPΔPDPN) mouse models, respectively. Activation of the Cre recombinase was achieved through the administration of tamoxifen (22 mg ml⁻¹) resuspended in corn oil. Tamoxifen was dosed (82 µg g⁻¹) intraperitoneally on 3 consecutive days a week before immunization. Females and males aged between 8 and 15 weeks were used for experiments, unless stated otherwise. Animals were assigned experimental groups at random.

**Immunization model.** Mice were immunized via subcutaneous injection into the right flank, proximal to the hip, with 100 µl of an emulsion of OVA with IFA or complete Freunds adjuvant (CFA) where stated (100 µg OVA per mouse) (Hooke Laboratories). Where stated, mice were treated with Y27632 (Tocris, 1254). Next, 10 µl 1 mg ml⁻¹ Y27632 dissolved in PBS was injected subcutaneously into the right flank, proximal to the hip; 10 µl sterile PBS was used as injection control. Y27632/ PBS injections were given on 3 consecutive days 24 h after IFA/OVA injection. After 5 or 3 days, mice were culled, and inguinal LNs from both flanks (naive and inflamed) were extracted for paired histological studies, flow cytometry analysis, or ex vivo laser ablation.

**Ex vivo cultures.** Preparation for ex vivo live LN laser ablation was optimized following previously established methods[14,49–51]. UltraPure low-melting-point agarose (Thermo Fisher Scientific) was prepared in PBS at 3% w/v and maintained at 37 °C. LNs were embedded into agarose and left for 5 min to set on ice. LN blocks were secured by superglue to cutting stage and placed into ice-cold PBS cutting chamber. Leica Vibratome (VT1200S) 200-µm-thick sections were cut at a rate of 0.3 mm s⁻¹ and 1.5 mm amplitude until the LN was completely sectioned. Collected sections were placed into RPMI 1640 containing 10% fetal bovine serum, 1% penicillin and streptomycin (P/S) (Thermo Fisher Scientific, 15140122), and 1% insulin-transferrin-selenium (ITS) (Thermo Fisher Scientific, 41400045) at 37 °C, 10% CO₂ and imaged within 24 h. Live sections recovered for 1 h before being used for live imaging.

**Laser ablation.** LN sections were transferred to glass-bottom 35-mm MatTek dishes (P35G-1.5-20-C) with a small volume of RPMI media containing 1% P/S and 1% ITS. A glass coverslip was placed on top to secure the section. Sections imaged on a Zeiss LSM 880 inverted multiphoton microscope with the imaging chamber maintained at 37 °C, 10% CO₂. Where stated, sections were treated with 100 µM Y27632 (Tocris, 1254) diluted in ex vivo culture media for at least 1 h before imaging and ablation. Sections were imaged with a Plan-Apochromat ×40 oil objective (NA 1.3), 1,024 × 1,024 resolution, and ×4 digital zoom for ablation ROIs. Laser ablation of the FRC network was achieved by using a pulsed Chameleon Vision II TiSa with laser power 75–80% (coherent) tuned to 760 nm. Ablation was performed on small, manually defined linear ROIs between FRC connections away from cell bodies. Ablation was performed in a single z-plane at the center of the FRC connection in a vibratome slice. Time-lapse videos were recorded over 25 s on a single channel photomultiplier tube (PMT) detector 512 × 512 pixels, with 521-ms scan speed per frame to capture recoil of the network. Recoil of the network was calculated by manually measuring displacement of the FRC network between two points located away from the ablation site, with initial recoil velocities calculated from the displacement one frame after the cut, as in other studies[22,52].

**Immunostaining of tissue sections.** LNs that were used for sectioning and immunofluorescence were fixed in Antigen fix (DiaPath) for 2 h at 4 °C with gentle agitation. LNs were washed for 30 min in PBS before being applied to 30% w/v sucrose 0.05% sodium azide solution at 4 °C overnight. LNs were dipped into Tissue-Tek optimum cutting temperature compound before being embedded into molds containing optimum cutting temperature compound. A maximum of six LNs were embedded into a single block for comparative analysis. LNs were sectioned on the Leica cryostat at a thickness of 20 µm.

For immunofluorescent staining, sections were permeabilized and blocked using 10% normal goat serum (NGS), 0.3% Triton X-100 in PBS for 2 h at 20–25 °C. Primary antibodies were diluted according to Supplementary Table 1 in 10% NGS, 0.01% Triton X-100 in PBS, and the mix was centrifuged at 15,000 g for 5 min at 4 °C. Sections were incubated with primary antibodies overnight at 4 °C. Sections were then brought to 20–25 °C and washed three times for 15 min

each in 0.05% PBS-Tween 20. Sections were then blocked using 10% NGS, 0.3% Triton X-100 in PBS for 2 h at 20–25 °C. Secondary antibodies were then prepared as primary antibodies with dilutions given in Supplementary Table 1 and were applied to the sections for 2 h at 20–25 °C. This was followed by two 15-min washes of 0.05% PBS-Tween 20 and a final wash of 15 min in PBS. Sections were then mounted using mowiol mounting media.

For staining of 200-µm-thick vibratome sections, slices were first fixed in Antigen fix (DiaPath) for 2 h at 4 °C with gentle agitation. Sections were placed into 0.1 M Tris-HCl, pH 7.4, on ice for 30 min. Sections were permeabilized using IHC buffer containing 0.5% BSA, 2% Triton X-100 (Sigma-Aldrich) in 0.1 M Tris-HCl, pH 7.4, for 20 min at 4 °C with gentle agitation. Primary antibodies were diluted into IHC buffer, according to Supplementary Table 1, centrifuged at 15,000 g for 5 min at 4 °C, and applied to the sections for 1 h at 4 °C with gentle agitation. Sections were then washed twice with 0.1 M Tris-HCL for 15 min. Sections were then incubated for 1 h at 4 °C with secondary antibodies prepared as primary antibodies with dilutions mentioned in Supplementary Table 1. Two final washes of 0.1 M Tris-HCl were performed before mounting the sections with mowiol mounting media and a glass coverslip. Sections were then imaged on the Zeiss LSM 880 inverted multiphoton microscope using a Plan-Apochromat ×40 oil objective (NA1.3).

Unless otherwise stated, confocal images were acquired using Leica TCS SP8 STED 3X or the Leica TCS SP5 using HCX Plan-Apochromat ×40 (NA 1.25) and ×63 (NA 1.4) oil lenses. Images were captured at 1,024 × 1,024 pixels, three-line average onto hybrid pixel (HyD) or photomultiplier tube (PMT) detectors. Fluorophore excitation and acquisition was performed in a sequential and bidirectional manner. Imaging regions were manually defined, and z-stacks (15–40 µm) with regular z-intervals ranging from 0.5 µm to 1 µm (depending on the sample) were acquired using a motorized stage. Tile scans were automatically stitched (numerical) using Leica imaging software. To quantify T cell packing, ECM fiber thickness and pMLC-positive and perlecan positive F-actin fibers, Fiji (ImageJ) was used on acquired z-stacks and maximum projections. For T cell packing (×40 lens), an in-house Fiji (ImageJ) macrocleared nuclei inside PDPN⁺ stain. Then, a single z-plane of DAPI (nuclei) had despeckle and gaussian blur (sigma = 2) applied. Nuclei were then detected and counted using thresholding and watershed segmentation. For F-actin fiber analysis (×63 lens), hand-drawn ROIs of F-actin fibers were applied to PDPN maximum projections to check and count the number of F-actin fibers that were within the PDPN⁺ FRC network. Then, F-actin fiber ROIs were applied to pMLC and perlecan channels to count the number and percentage of pMLC⁺ fibers or perlecan-aligned fibers. The FRC gap analysis used an in-house MATLAB script[3]. Briefly, PDPN fluorescence maximum projections (×40 lens) were converted into a binary mask before a circle-fitting algorithm consecutively fit the largest circle possible within the gaps in the network that did not overlap with other fitted circles. Each circle was given a radius, and the distribution of circles with radius >12 µm were plotted. The MATLAB script is available at https://doi.org/10.5522/04/8798597.v1.

**Flow cytometry of LNs.** LNs were carefully dissected and weighed and placed into RPMI 1640 media on ice. LNs were then processed as previously described[3,53]. Briefly, LNs were placed into a digestion buffer containing collagenase D (250 µg ml⁻¹) (Millipore Sigma), dispase II (800 µg ml⁻¹) (Thermo Fisher Scientific), and DNase I (100 µg ml⁻¹) (Sigma-Aldrich). LNs were gently digested in a water bath at 37 °C, removing and replacing the cell suspension every 10 min until completely digested. Cell suspensions were then centrifuged at 350 g for 5 min. The cells were resuspended in PBS, consisting of 1% BSA (Sigma-Aldrich) and 5 mM EDTA (Sigma-Aldrich), and were then filtered, counted, and resuspended at 10 × 10⁶ cells ml⁻¹. Then, 2.5 × 10⁶ cells were seeded and stained for surface and intracellular markers for a stromal cell or T cell panel (Supplementary Table 1). Cells were blocked with CD16/CD32 Mouse Fc block (BD Biosciences) and then stained with primary antibodies for 20 min at 4 °C. For intracellular staining of Ki67 cells, cells were fixed and permeabilized using FOXP3 fix/perm buffer as specified by the manufacturer (BioLegend). Samples were run on the Fortessa X20 flow cytometer (BD Biosciences) at the University College London Cancer Institute. Data were analyzed using FlowJo software.

**Cell lines and primary cell culture.** Immortalized FRCs (control FRC) were generated as described by Acton et al.[3,32]. PDPN was stably knocked down (PDPN KD FRC) in the parental cell line by transfection of a PDPN shRNA lentivirus. PDPN was completely depleted from the parental cell line (PDPN KO FRC) using CRISPR-Cas9 genetic deletion. In all experiment where exogenous PDPN mutant cell lines (PDPN WT, PDPN T34A, and PDPN S167A-S171A) were used, protein production was induced by the addition of 1 µg ml⁻¹ doxycycline for 48 h. Cell lines were maintained at 10% CO₂, 37 °C in Dulbecco's modified Eagle medium (Thermo Fisher Scientific) supplemented with 10% fetal bovine serum, 1% P/S, and 1% ITS unless otherwise stated. In vitro experimental groups were defined by the genotype of the cell lines. Cells were treated with recombinant CLEC-2-Fc or Control-Fc supernatant[3,32] for approximately 2 h, where indicated, to assay the effect of CLEC-2 signaling through PDPN. Primary cell isolation from murine LNs was performed as previously described[53] and cultured as with cell lines. Primary cells were then treated with 20 µM Y27632 (Tocris, 1254) for indicated timepoints.

Culture media was changed every 3 days, upon which fresh ROCK inhibitor was applied. Cells were collected and analyzed by flow cytometry.

**Tether pulling and trap force measurements.** Trap force measurements were performed using a home-built optical tweezer and a 4-W 1,064-nm laser quantum Ventus within a ×100 oil immersion objective (Nikon, NA 1.30, CFI Plan Fluor DLL) on an inverted microscope (Nikon, Eclipse TE2000-U) equipped with a motorized stage (PRIOR Proscan). The optical tweezer was calibrated following previous studies[30,31]. The trap force calibration was performed in every experiment with typical calibration trap stiffness of k∼0.114 pN nm⁻¹. Measurements were performed using concanavalin A-coated (50 µg ml⁻¹) carboxyl latex beads, (1.9 µm diameter, Thermo Fisher Scientific). Beads were incubated on a shaker with concanavalin A for 2 h before the experiment. Beads were applied to the culture media, manipulated by the optical trap, and brought into contact with the cellular membrane and typically held for 2–5 s to allow binding to membrane. Bead position was recorded every 90 ms in brightfield before and during tether formation. Cells were maintained in the trap at 37 °C and had $CO_2$ flowing into the chamber. CLEC-2 or CTRL supernatant was added 2–4 h before measurement of trap force. Trap force ($F_t - pN/\mu m$) was then calculated based on the trap calibration ($k$), bead position ($\Delta x$), using a homemade Fiji macro[31] and the equation $F_T = k\Delta x$.

**Osmotic swelling assay.** Osmotic shock was performed in accordance with previously reported protocols[54]. By altering the osmolarity of a solution, cells swell or shrink. Osmolarity was estimated using osmolarity calculations. Isotonic solution was prepared with 137 mM NaCl, 5.4 mM KCl, 1.8 mM $CaCl_2$, 0.8 mM $MgCl_2$, 20 mM HEPES, 20 mM D-glucose and pH to 7.4 with NaOH. Hypotonic solutions were prepared by diluting the isotonic preparation in milliq water that is 50 mOsm is a 1/6 dilution of isotonic solution. To control for the dilution of ions in solution and account for the effect this may have on swelling, the 50 mOsm solution was restored to 330 mOsm using D-mannitol at 280 mM, acting as the true isotonic control (ISO). Cells were dissociated from cell culture with sterile Dulbecco's phosphate-buffered saline (Thermo Fisher Scientific) + 2 mM EDTA and placed onto individual 35-mm MatTek dishes and allowed to settle for 30 min. After 30 min, the cells remain rounded on the coverslip. Cells were then treated with either ISO, HYPO 50 mOsm, or Extreme HYPO 0 mOsm for 1 h. Phase contrast images of cell swelling were captured every 30 s using a ×20 air objective on a Nikon Ti inverted microscope with a motorized stage controlled by NIS-elements software. Diameter of individual swelling cells were calculated using manual circular ROIs. The ratio of swelling was then calculated by dividing all diameters (d) by the initial diameter (d0). Area under the curves were calculated for the first 20 min of the swelling response.

**Polyacrylamide gels and cell division quantification.** Polyacrylamide gels were generated as previously described[38]. Polyacrylamide gels were polymerized on 12-mm glass coverslips. Coverslips were first functionalized by incubation for 1 h with a solution of 0.3 % Bind-Silane (Sigma-Aldrich, M6514)/5% acetic acid in ethanol. Coverslips were then rinsed with ethanol and dried with compressed air. Stiff to soft polyacrylamide gel solutions were made up as 0.5 ml solutions in PBS as follows: Stiff gels (∼30 kPa): 306.25 µl PBS, 150 µl acrylamide 40% (Sigma-Aldrich), 37.5 µl bisacrylamide 2% (Thermo Fisher Scientific), 2.5 µl ammonium persulfate (APS) (10% in water, Sigma-Aldrich), 0.25 µl tetramethyl-ethylenediamine (TEMED) (Sigma-Aldrich); (∼12 kPa): 359.35 µl PBS, 94.4 µl acrylamide 40% (Sigma-Aldrich), 40 µl bisacrylamide 2% (Fisher Scientific), 2.5 µl APS (10% in water, Sigma-Aldrich), and 0.25 µl TEMED (Sigma-Aldrich). Soft gels (∼2 kPa): 407.65 µl PBS, 68.6 µl acrylamide 40% (Sigma-Aldrich), 17.5 µl bisacrylamide 2% (Thermo Fisher Scientific), 2.5 µl APS (10% in water, Sigma-Aldrich), and 0.25 µl TEMED (Sigma-Aldrich). Following TEMED addition, 200 µl gel solution was immediately pipetted onto a flat Perspex plate, and a functionalized coverslip was placed on top. Following polymerization, gels were removed from the Perspex using a square-edged scalpel and hydrated by incubation in PBS for 1–2 h. Gels were then functionalized with Sulfo-SANPAH (1 ug ml⁻¹; Thermo Fisher Scientific, 22589) and UV exposure (365 nm wavelength) for 5 min. Gels were then washed with PBS before incubating with collagen (rat tail collagen I, Sigma-Aldrich, C3867-1VL, 0.1 mg ml⁻¹) overnight at 4 °C. Gels were then incubated with cell culture media before cells were imaged on an inverted Nikon Ti microscope using ×20 widefield objective and a CoolLED pE-300 with an incubation chamber at 10% $CO_2$ and 37 °C. Control and PDPN KO FRCs were transfected with nuclear localized GFP (Addgene, 11680 (ref. [55])) and tracked over 72 h using ImageJ on polyacrylamide gels of different stiffnesses. The number of divisions over the whole imaging time were counted and normalized to the starting number of nuclei to calculate the average number of divisions per cell.

**Immunoblot.** Equal cell numbers were grown to confluency, and protein was isolated using 300 µl 4× Laemmli lysis buffer (Bio-Rad) and cell lifters (Thermo Fisher Scientific). Lysates were then sonicated for 20 s followed by 10 min at 95 °C. Then, 1% β-mercaptoethanol (143 mM stock, Sigma-Aldrich) was added to samples to reduce oligomerized protein structures. Electrophoresis

gels (10%) were loaded with the same quantity of lysates and run for 60 min at 110 V. Transfer to polyvinylidene fluoride (PDVF) membranes was carried out at 65 V for 2 h at 4 °C. Membranes were blocked for 2 h at 20–25 °C with 5% skim milk powder (Sigma-Aldrich), 1% BSA in TBS and stained with primary antibodies (Supplementary Table 1) overnight at 4 °C in 1:5 diluted blocking buffer. Membranes were then washed thoroughly with TBS 0.05% Tween 20 and incubated with horseradish peroxidase-conjugated secondary antibodies (Supplementary Table 1) for 1 h at 20–25 °C in 1:5 blocking buffer. After washing with TBS 0.05% Tween 20, membranes were visualized using ECL-horseradish peroxidase reaction and imaged using Image Quant 5000 (GE Life Sciences).

**Linear unmixing and Imaris rendering.** Imaging of $Pdgfra^{iR26R-Confetti}$ LN sections was carried out using lambda mode and chameleon laser at 900 nm to acquire multi-channel images. Widefield images and z-stack intervals of 0.5 nm were obtained, for an approximate thickness of 30–60 µM. The emission wavelengths for each fluorophore were set on Zeiss Zen black software spectral unmixing function by selecting labeled cells within the confetti imaging. The second harmonic of the two-photon laser detected the ECM conduit structure in the LN. Rendering of PDPN, CFP, and YFP in Figs. 1, 3, and 4 and supplementary movies was achieved using Imaris surface tools.

**Quantification and statistical analysis.** Prism7 Software (GraphPad) was used to perform multiple statistical analyses, including appropriate tests that were performed as indicated in figure legends. Data collection and analysis were not performed blind to the conditions of the experiments. Data distribution was assumed to be normal, but this was not formally tested. In general, comparison of multiple groups was performed using one- or two-way ANOVA with Tukey's multiple comparisons depending on the dataset. Comparisons of two data sets were mostly performed using two-tailed Mann–Whitney tests.

**Reporting summary.** Further information on research design is available in the Nature Research Reporting Summary linked to this article.

## Data availability
There are no restrictions on data availability. Data, code, or reagents are available upon request. Numerical source data files for all figures are provided in Excel supplementary data files and listed in the inventory. Image source data files for all figures are supplied in TIFF format in the supplementary data files and listed in the inventory. Source data are provided with this paper.

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

## Acknowledgements
This work was supported by a European Research Council Starting Grant (LNEXPANDS; to S.E.A.), a Cancer Research UK Careers Development Fellowship (CRUK-A19763; to S.E.A.), the Medical Research Council (MC-U12266B), MRC awards MR/L009056/1 and MR/T031646/1 (Y.M.), a Lister Institute Research Prize and the EMBO Young Investigator Programme (Y.M.), the European Union's Horizon 2020 research and innovation program under the Marie Sklodowska-Curie grant agreement 641639 (ITN Biopol, E.K.P.), and the Medical Research Council UK (MRC program award MC_UU_12018/5 (E.K.P.). We thank E. Sahai, G. Charras, and V. G. Martinez for critical reading of the manuscript. We also thank H. Clevers for supplying R26R-confetti mice and S. Watson and C. Buckley for providing the PDPN^fl/fl mouse model.

## Author contributions
H.L.H. and S.E.A. designed the study and wrote the manuscript. H.L.H. performed and analyzed the majority of the experiments. A.C.B. and S.E.A. generated the

PDGFRα-mGFP-CreERT2 mouse model. R.J.T. assisted with laser ablation studies. H.d.B. assisted with optical trap measurements. H.d.B.'s contribution was carried out under the support of E.K.P. at the MRC Laboratory for Molecular and Cell Biology at University College London (current affiliation is the Department of Biochemistry and Biophysics, University of California, San Francisco). S.M. assisted with flow cytometry of LNs. L.J.M. conducted *Pdgfra*[iR26R-Confetti] imaging experiments of FRCs in vivo. L.A.H. conducted in vitro mechanical proliferation assays. C.M.d.W. generated CD44 KO cell lines and conducted in vitro experiments. Y.M. and E.K.P. contributed to study design and editing of the manuscript. All authors contributed to editing the manuscript.

## Competing interests

The authors declare no competing interests.

## Additional information

**Extended data** is available for this paper at https://doi.org/10.1038/s41590-022-01272-5.

**Correspondence and requests for materials** should be addressed to Sophie E. Acton.

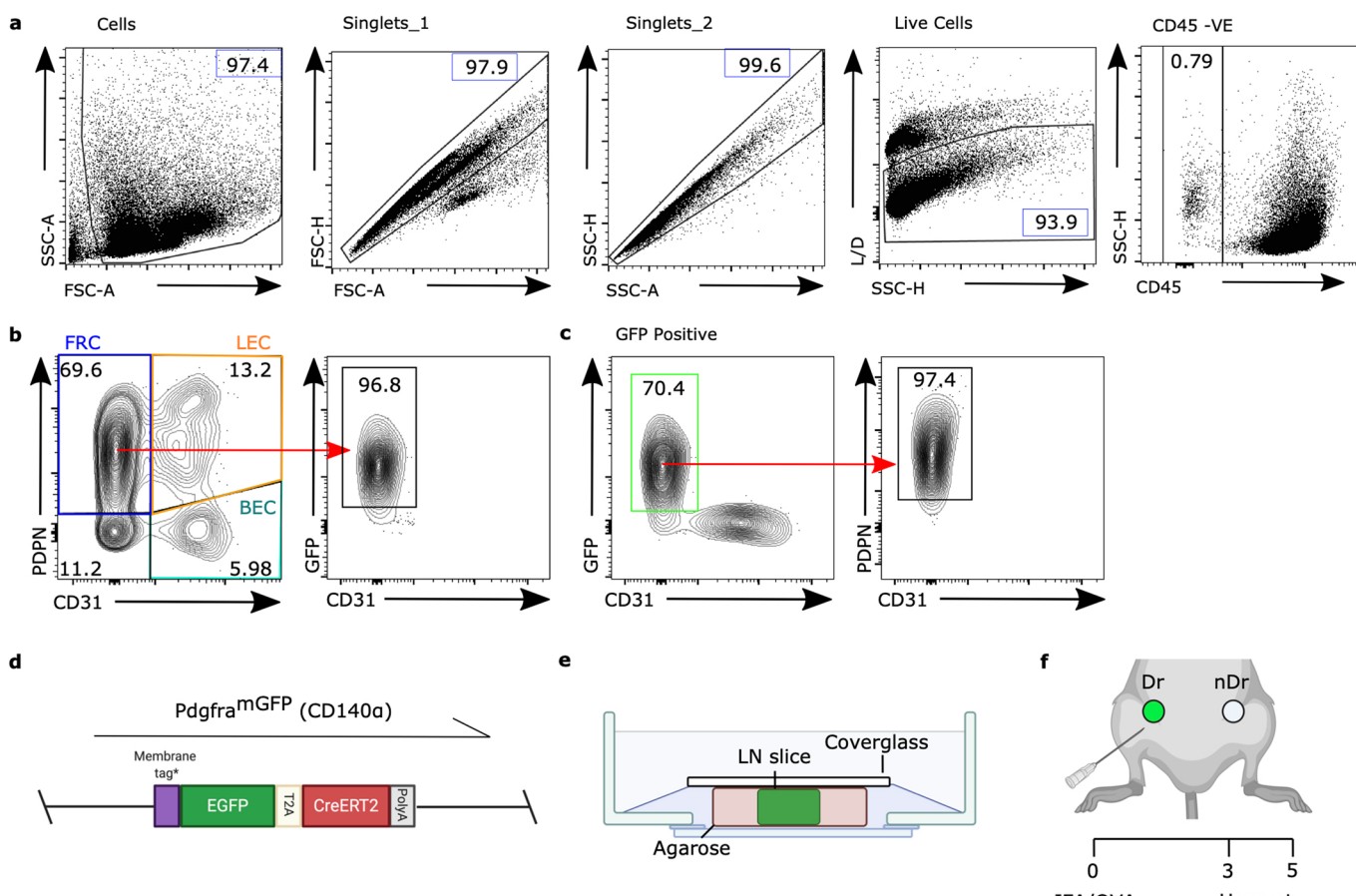

**Extended Data Fig. 1 | Flow cytometry of PDGFRα$^{mGFP}$ fibroblasts, ex vivo laser ablation and immunization. a-b)** Flow cytometric gating strategy and representative dot plots and percentages of **a)** CD45⁻ stroma cells and **b)** subpopulations. Fibroblastic reticular cells (FRC, PDPN⁺CD31⁻), blood endothelial cells (BEC, PDPN⁻CD31⁺), lymphatic endothelial cells (LEC, PDPN⁺CD31⁺). Red arrow shows GFP⁺ FRCs **c)** Representative gating of GFP⁺ Fibroblasts. **d)** A membrane-targeted GFP molecule is driven under the PDGFRα promoter. **e)** LN nodes were embedded in low melt agarose and sliced at 200 µm thickness before being secured by cover glass for imaging. **f)** IFA/OVA is used as model immunization with inguinal draining (dr) and non-draining (nDr) lymph nodes (LNs) harvested day 3 or day 5 post immunization (right). This figure supports Figs. 1–2.

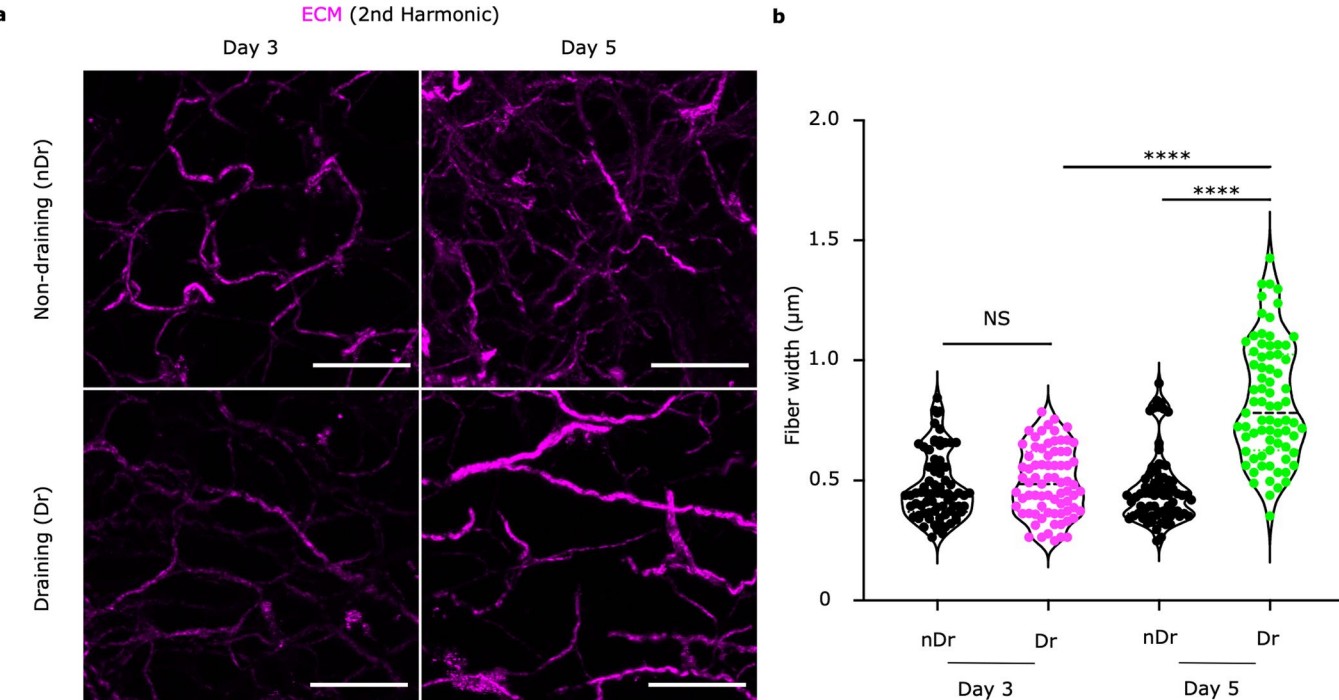

**Extended Data Fig. 2 | Analysis of ECM structures at Day 3 and Day 5 post-immunization. a)** 2nd Harmonic (magenta) signals visualize the ECM network in vibratome sections of lymph nodes. Scale bar, 15 μm. **b)** Quantification of ECM fiber thickness between day 3 and day 5 samples. Violin indicates plot indicates median and minimum/maximum. One-way ANOVA with Tukey's multiple comparisons, ****$p = 1.00E^{-6}$. $n = 75$ individual fiber ROI for all groups from 3 independent LNs. This figure supports Fig. 3.

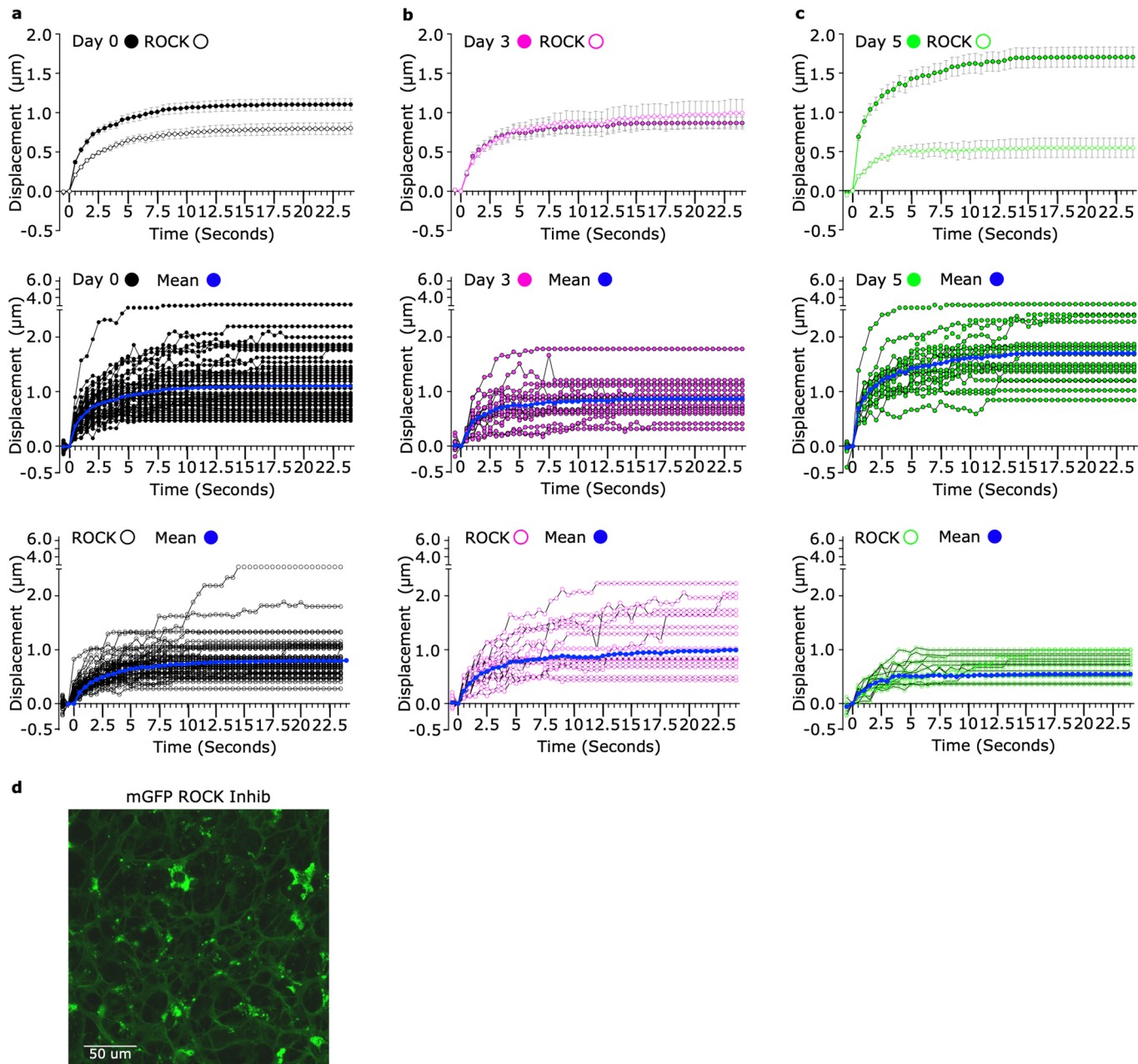

**Extended Data Fig. 3 | Y27 treatment of lymph node slices reduce tension in the FRC network. a-c)** Recoil curves show displacement over time (mean ± SEM) (top panel). Individual recoil curves of control (middle) and Y27 treated LNs (bottom). n = individual ablation where Day 0 (n = 54), Day 0 + Y27 (n = 43), Day 3 (n = 18), Day 3 + Y27 (n = 20), Day 5 (n = 19), Day 5 + Y27 (n = 15) over 3 independent experiments. **d)** Membrane GFP visualizes the FRC network in the ex vivo lymph node slice. Y27 treatment has no effect on the FRC network connectivity. Scale bar, 50 μm. This figure supports Fig. 3.

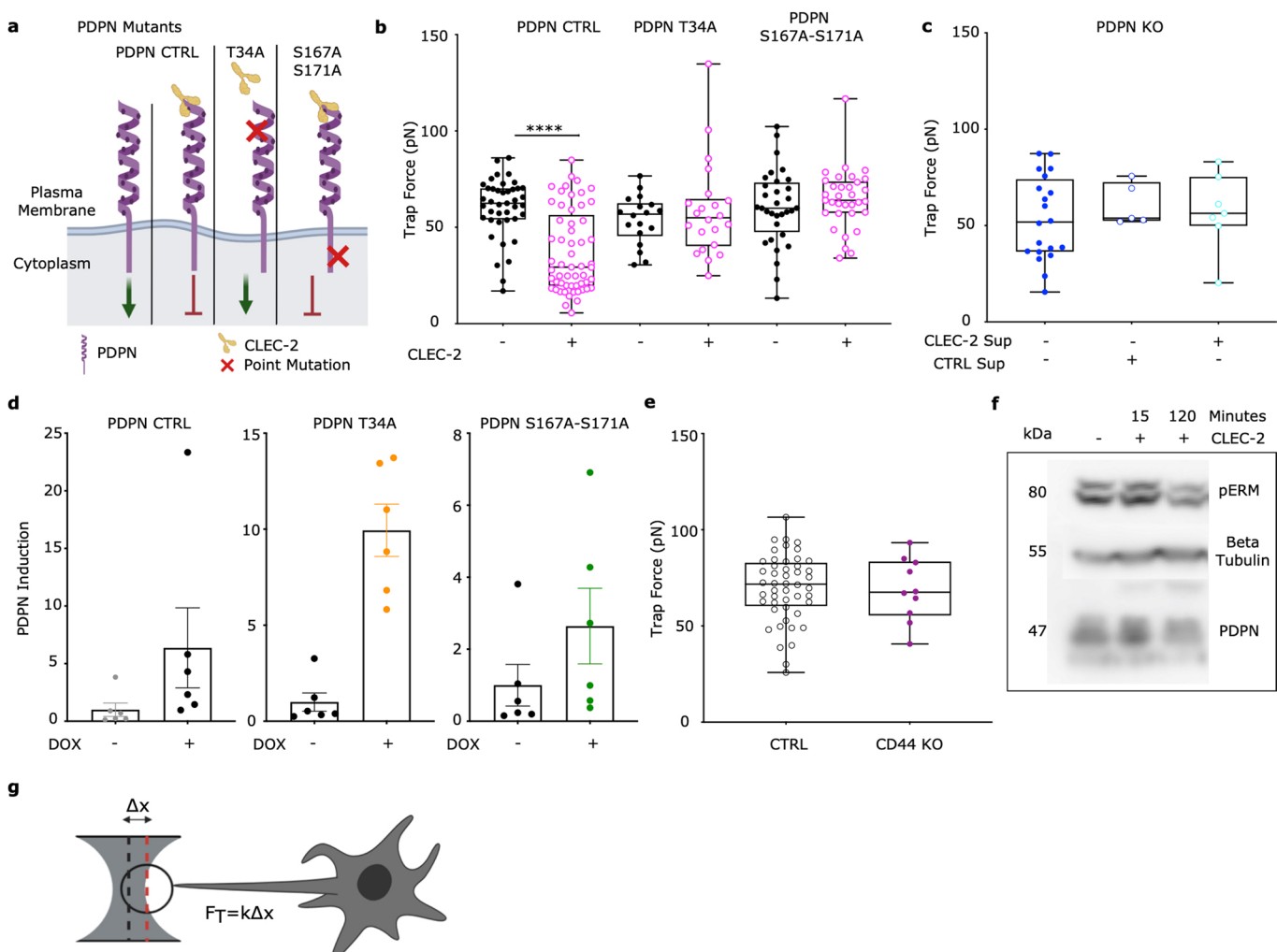

**Extended Data Fig. 4 | Induction of PDPN mutants in PDPN KO fibroblastic reticular cell line. a)** Schema of exogenous PDPN mutants and interaction with CLEC-2. Green arrow denotes active signaling by PDPN leading to actomyosin contractility, red arrow indicates inhibition of PDPN signaling and reduction in actomyosin contractility**. b)** Trap force measurements of FRCs expressing PDPN mutants after pre-treatment of CLEC-2. Box plot indicates median, interquartile range and minimum/maximum. One-way ANOVA with Tukey's multiple comparisons, ****p=1.00E$^{-6}$. n = individual cells where CLEC-2$^-$ PDPN CTRL (n=41), CLEC-2$^+$ PDPN CTRL (n=57), CLEC-2$^-$ T34A (n=18), CLEC-2$^+$ T34A (n=22), CLEC-2$^-$ S167A-S171A (n=32), CLEC-2$^+$ S167A-S171A (n=33) analyzed over 5 independent experiments. **c)** Trap force measurements of PDPN CRISPR KO FRCs treated with CTRL or CLEC-2 supernatant. Box plot indicates median, interquartile range and minimum/maximum. One-way ANOVA with Tukey's multiple comparisons. n = individual cells where PDPN KO (n=20), CTRL SUP$^+$ PDPN KO (n=5), CLEC-2$^+$ SUP PDPN KO (n=7) over 2 independent experiments. **d)** Fold induction of exogenous PDPN, based on PDPN staining geometric mean, for each PDPN mutant cell line treated with or without doxycycline. PDPN CTRL (left), PDPN T34A (middle) and PDPN DSS (right). Mean ±SD error bars, N=6. **e)** Trap force measurements of CTRL and CD44 (purple) KO FRCs. Box plot indicates median, interquartile range and minimum/maximum. Mann–Whitney test (two-tailed). n = individual cells where CTRL (n=48) and CD44 KO (n=10). **f)** Representative immunoblot of pERM in CTRL FRCs after treatment with CLEC-2. n=2 independent experiments. **g)** Schema and equation to calculate trap force (F$_t$) from displacement. This figure supports Fig. 4.

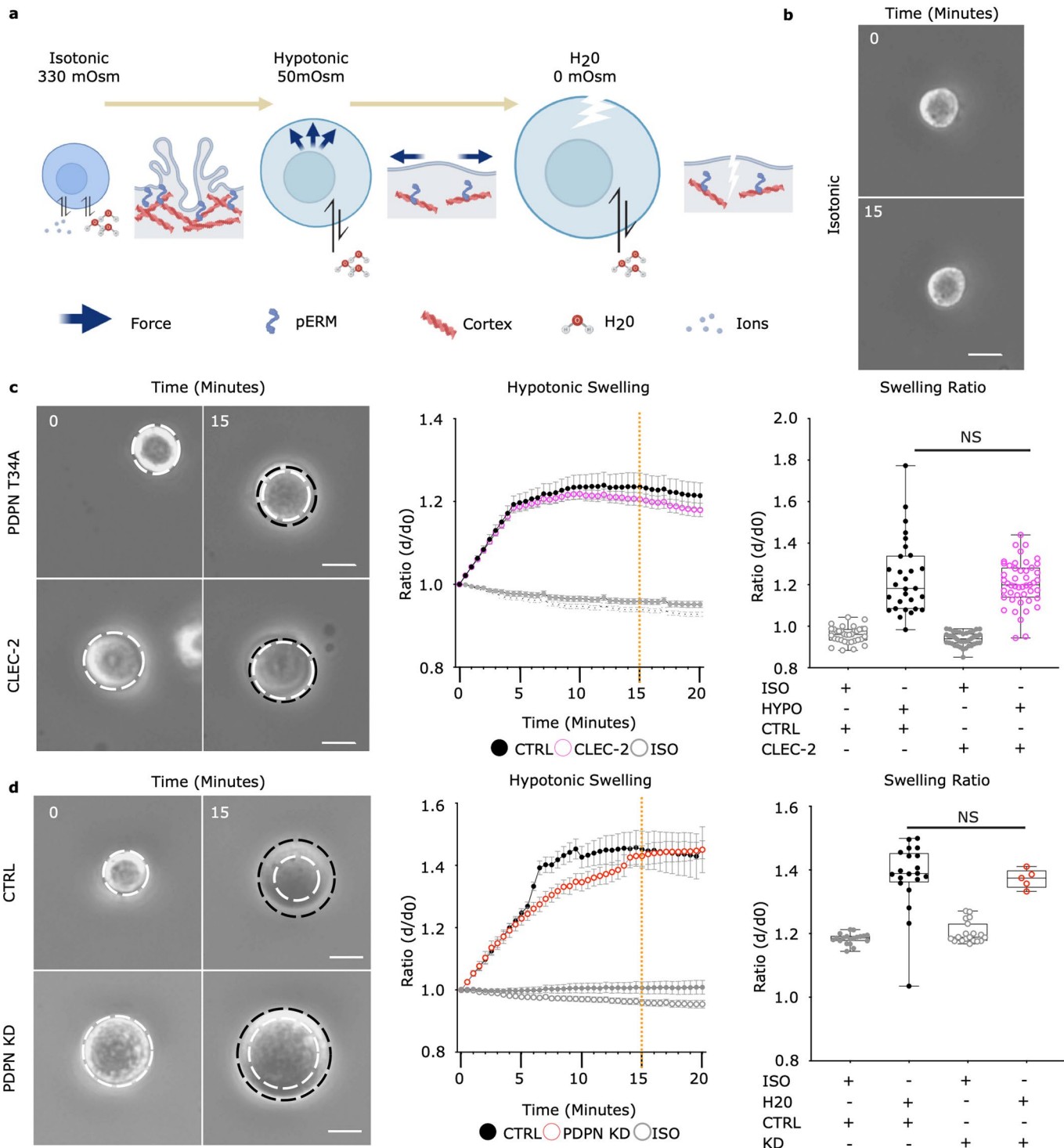

**Extended Data Fig. 5 | Membrane access of fibroblastic reticular cells is increased through the CLEC-2/PDPN signaling axis. a)** Schematic of osmotic shock experiment, inducing swelling by altering osmolarity. **b)** Representative stills of FRC in isotonic control media. Scale bar, 25 μm. **c-d)** Time course of **c)** PDPN T34A FRCs with or without CLEC-2 treatment in hypotonic conditions, and **d)** time course of PDPN CTRL vs PDPN shRNA KD FRCs in extreme hypotonic conditions ($H_2O$). Stills (left) show swelling of cells in hypotonic media. White dotted circle marks initial size before swelling and is compared to swelling at t=20 (black dotted circle). Scale bar, 25 μm. Change in diameter ratio over time (middle, mean ± SEM). Swelling ratio comparisons between control and **c)** CLEC-2 treatment or **d)** PDPN shRNA KD FRCs at 15 min post swelling (orange dotted line, right). Box plot indicates median, interquartile range and minimum/maximum. One-way ANOVA with Tukey's multiple comparisons. **c)** n = individual cells where CLEC-2⁻ ISO (n=42), CLEC-2⁻ HYPO (n=47), CLEC-2⁺ ISO (n=34), CLEC-2⁺ HYPO (n=29) analyzed over 5 independent experiments. **d)** n = individual cells where Control ISO (n=19), Control H20 (n=20), PDPN KD ISO (n=19), PDPN KD H20 (n=5) analyzed over 5 independent experiments. This figure supports Fig. 4.

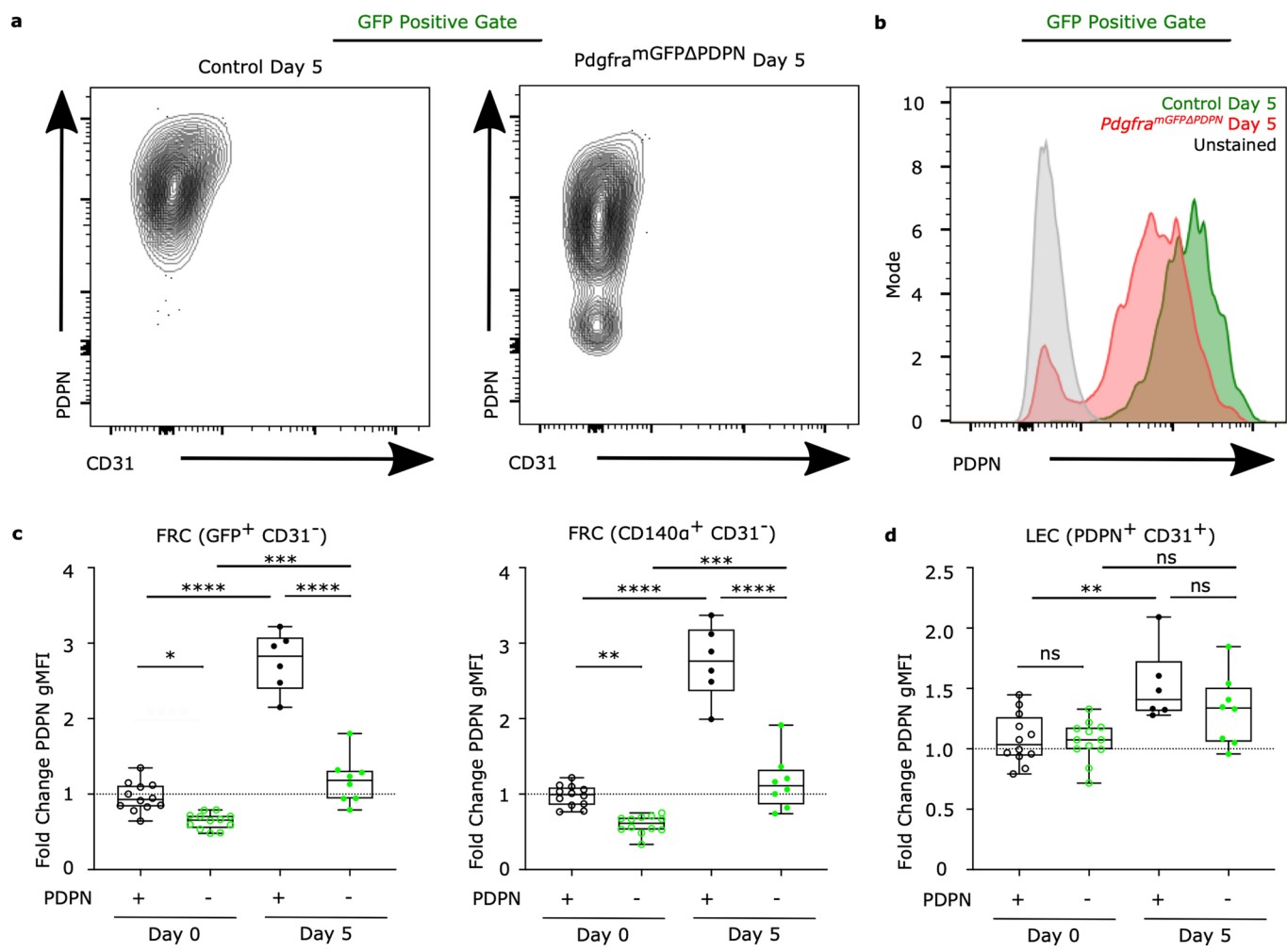

**Extended Data Fig. 6 | Flow cytometry of *Pdgfra^mGFPΔPDPN* fibroblasts. a)** Example PDPN contour plots for GFP⁺ FRCs from control and *Pdgfra^mGFPΔPDPN* LNs. **b)** Representative histograms of surface protein expression for PDPN in control and *Pdgfra^mGFPΔPDPN* LNs 5 Days post CFA/OVA. **c)** Fold change in PDPN expression (gMFI) in FRCs (GFP⁺ CD31⁻, CD140α⁺ CD31⁻). Box plot indicates median, interquartile range and minimum/maximum. One-way ANOVA with Tukey's multiple comparisons. ****p = 1.00E⁻⁶, ***p = 0.000143, 0.0004, **p = 0.0084, *p = 0.0145. **d)** Fold change in PDPN expression (gMFI) for LECs (PDPN⁺ CD31⁺). Box plot indicates median, interquartile range and minimum/maximum. One-way ANOVA with Tukey's multiple comparisons. **p = 0.0033. **c, d)** n = individual LNs where (n=12), Day 0^ΔPDPN (n=12), Day 5 (n=6), Day 5^ΔPDPN (n=8) over 2 independent experiments. This figure supports Figs. 6, 7.

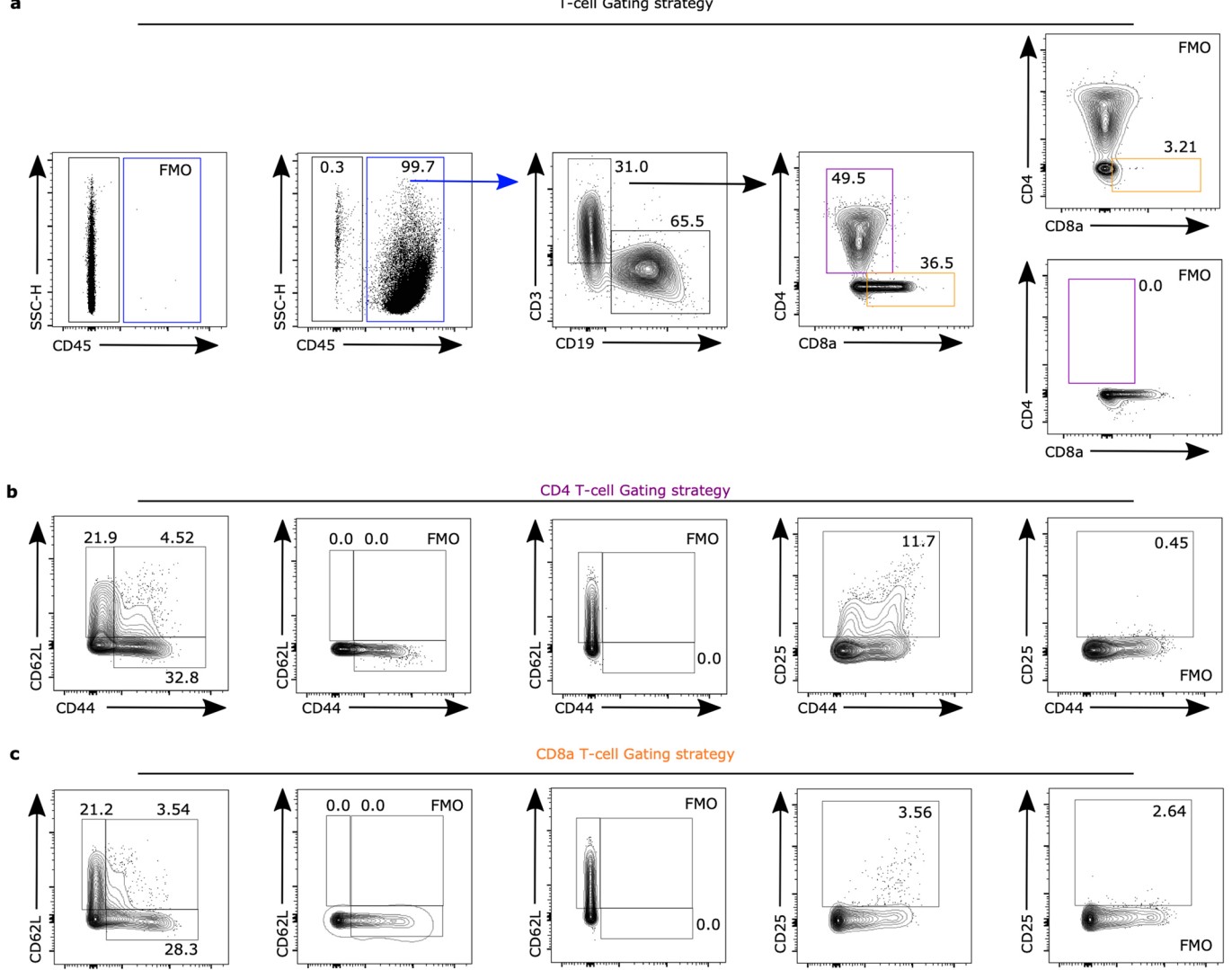

**Extended Data Fig. 7 | Flow cytometry T-cell gating strategy. a)** Example dot plots and percentages of CD45⁺, CD45⁻, CD19⁺, CD3⁺ populations and FMOs. **b)** Example dot plots and percentages of CD3⁺ CD4⁺ subpopulations and FMOs. **c)** Example dot plots and percentages of CD3⁺ CD8a⁺ subpopulations and FMOs. This figure supports Fig. 7.

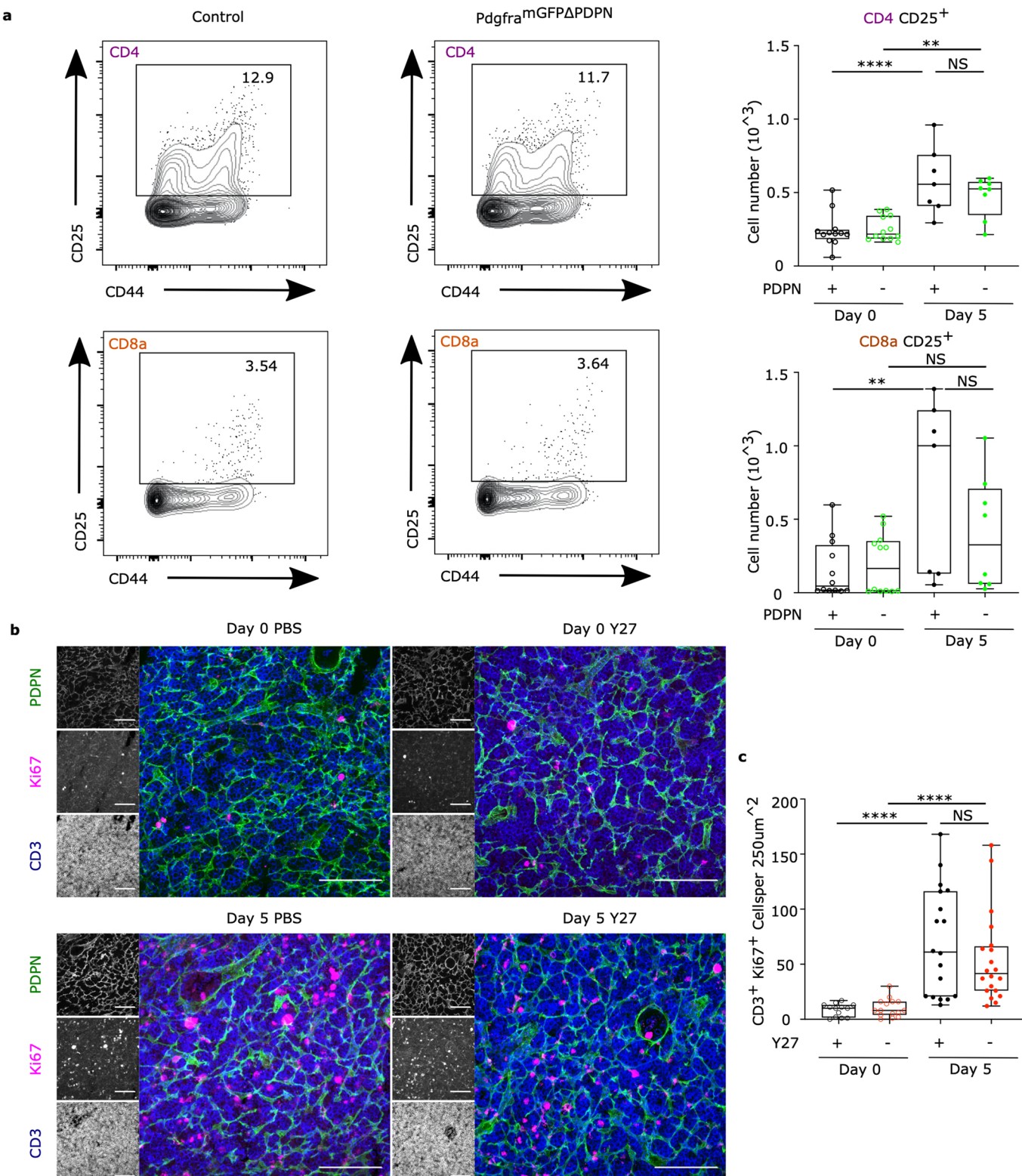

**Extended Data Fig. 8 | T-cell activation and proliferation is not altered by PDPN deletion or Y27 treatment. a)** Representative flow cytometric gating and cell number comparing control and *Pdgfra*[mGFPΔPDPN] CD3+ CD4+ CD25+, CD3+ CD8+ CD25+ population. Box plot indicates median, interquartile range and minimum/maximum. One-way ANOVA with Tukey's multiple comparisons, ****p = 1.00E[−6], **p = 0.0046, 0.0063. n = individual LNs where Day 0 (n = 12), Day 0[ΔPDPN] (n = 12), Day 5 (n = 7), Day 5[ΔPDPN](n = 8) over 2 independent experiments. **b)** Representative images of Ki67+ CD3+ staining of PBS or Y27 treated lymph nodes. Scale bar, 100 μm. **c)** Quantification of Ki67+ CD3+ T cells per area. Box plot indicates median, interquartile range and minimum/maximum. One-way ANOVA with Tukey's multiple comparisons, ****p = 1.00E[−6]. n = individual image ROIs where Day 0 (n = 13), Day 0 + Y27 (n = 15), Day 5 (n = 18), Day 5 + Y27 (n = 20) from 3 independent LNs. This figure supports Figs. 7 and 8.

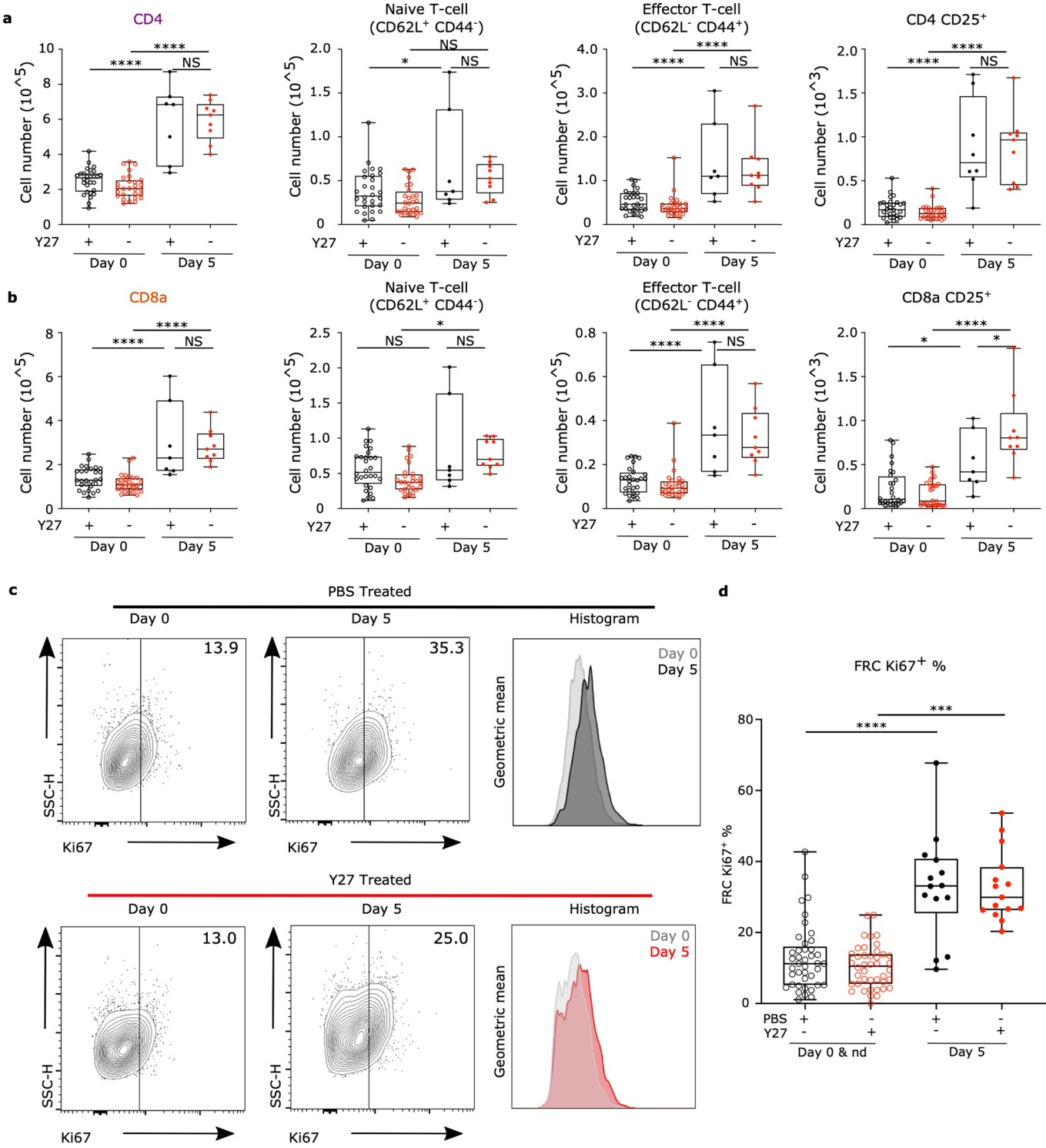

**Extended Data Fig. 9 | Flow cytometry gating strategy and Ki67 measurements. a-b)** CD3+ CD4+ and CD3+ CD8+ subpopulation cell numbers +/-ROCK inhibition (Y27632). Box plot indicates median, interquartile range and minimum/maximum. One-way ANOVA with Tukey's multiple comparisons, ****p=1.00E$^{-6}$, *p=0.0387, 0.0117, 0.0201, 0.0226. n = individual LNs where Day 0 (n=28), Day 0$^{Y27}$ (n=28), Day 5 (n=7), Day 5$^{Y27}$(n=9) over 2 independent experiments. **c)** Example of Ki67+ FRCs cell plots and histogram of Ki67 geometric mean, comparing immunization and Y27 treatments. **d)** Quantification of Ki67+ FRCs. Box plot indicates median, interquartile range and minimum/maximum. One-way ANOVA with Tukey's multiple comparisons. ****p=1.00E$^{-6}$, ***p=1.00E$^{-5}$. n = individual LNs where Day 0 (n=28), Day 0$^{Y27}$ (n=29), Day 5 (n=14), Day 5$^{Y27}$(n=15) over 5 independent experiments. This figure supports Fig. 8.

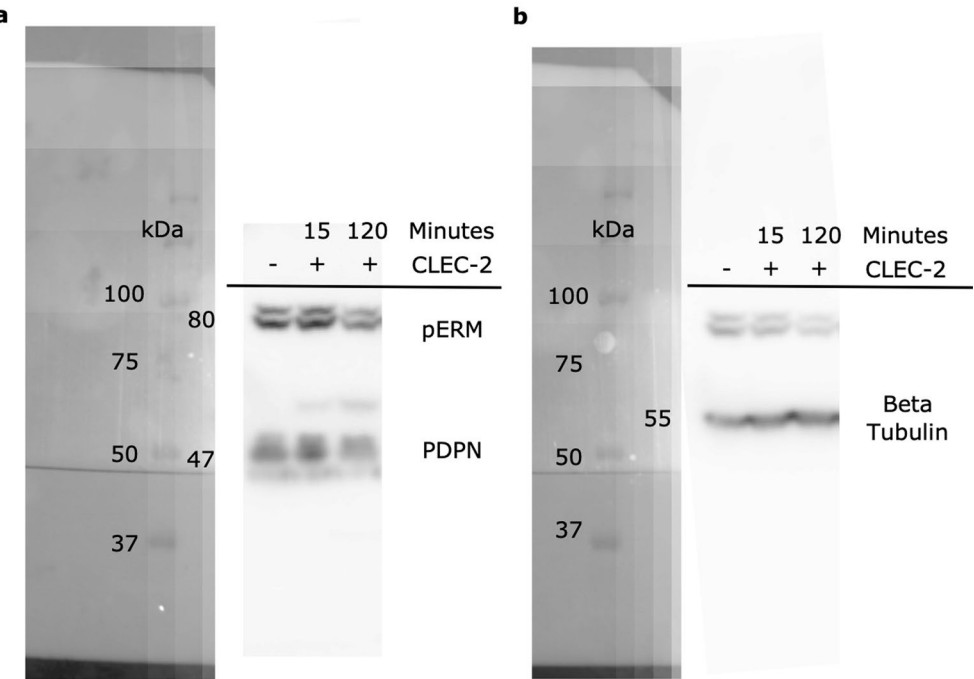

**Extended Data Fig. 10 | Immunoblot of pERM in CTRL FRCs after treatment with CLEC2. a)** Immunoblot and ladder (kDa) for pERM and PDPN.
**b)** Immunoblot and ladder (kDa) for beta-Tubulin.

# Reporting Summary

Nature Research wishes to improve the reproducibility of the work that we publish. This form provides structure for consistency and transparency in reporting. For further information on Nature Research policies, see Authors & Referees and the Editorial Policy Checklist.

## Statistics

For all statistical analyses, confirm that the following items are present in the figure legend, table legend, main text, or Methods section.

| n/a | Confirmed | |
|---|---|---|
| ☐ | ☒ | The exact sample size (*n*) for each experimental group/condition, given as a discrete number and unit of measurement |
| ☐ | ☒ | A statement on whether measurements were taken from distinct samples or whether the same sample was measured repeatedly |
| ☐ | ☒ | The statistical test(s) used AND whether they are one- or two-sided<br>*Only common tests should be described solely by name; describe more complex techniques in the Methods section.* |
| ☐ | ☒ | A description of all covariates tested |
| ☐ | ☒ | A description of any assumptions or corrections, such as tests of normality and adjustment for multiple comparisons |
| ☐ | ☒ | A full description of the statistical parameters including central tendency (e.g. means) or other basic estimates (e.g. regression coefficient) AND variation (e.g. standard deviation) or associated estimates of uncertainty (e.g. confidence intervals) |
| ☐ | ☒ | For null hypothesis testing, the test statistic (e.g. *F*, *t*, *r*) with confidence intervals, effect sizes, degrees of freedom and *P* value noted<br>*Give P values as exact values whenever suitable.* |
| ☒ | ☐ | For Bayesian analysis, information on the choice of priors and Markov chain Monte Carlo settings |
| ☒ | ☐ | For hierarchical and complex designs, identification of the appropriate level for tests and full reporting of outcomes |
| ☒ | ☐ | Estimates of effect sizes (e.g. Cohen's *d*, Pearson's *r*), indicating how they were calculated |

*Our web collection on statistics for biologists contains articles on many of the points above.*

## Software and code

Policy information about availability of computer code

| Data collection | No software was used. |
|---|---|
| Data analysis | Standard plugins available in Fiji/ImageJ (Version 2.3.0/1.53f) were used in imaging analysis.<br>Flow cytometry data was analysed using FlowJo (Version 10.7.1).<br>3D rendering was achieved using Imaris (Version 9.9)<br>Gap Analysis performed in MATLAB (MATLABR_2019b) (DOI:https://doi.org/10.5522/04/8798597.v1)<br>Statistical tests were performed using Graphpad Prism (Version 8.4.3) |

For manuscripts utilizing custom algorithms or software that are central to the research but not yet described in published literature, software must be made available to editors/reviewers. We strongly encourage code deposition in a community repository (e.g. GitHub). See the Nature Research guidelines for submitting code & software for further information.

## Data

Policy information about availability of data

All manuscripts must include a data availability statement. This statement should provide the following information, where applicable:

- Accession codes, unique identifiers, or web links for publicly available datasets
- A list of figures that have associated raw data
- A description of any restrictions on data availability

No restrictions on data availability
Data, code or reagents are available upon request.
Numerical source data files for all figures are provided in excel in supplementary data files and listed in the inventory
Image source data files for all figures are supplied in tiff format in supplementary data files and listed in the inventory

# Field-specific reporting

Please select the one below that is the best fit for your research. If you are not sure, read the appropriate sections before making your selection.

☒ Life sciences  ☐ Behavioural & social sciences  ☐ Ecological, evolutionary & environmental sciences

For a reference copy of the document with all sections, see nature.com/documents/nr-reporting-summary-flat.pdf

# Life sciences study design

All studies must disclose on these points even when the disclosure is negative.

| Sample size | No statistical methods were used to pre-determine sample sizes but our sample sizes are similar to those reported in previous publications (Acton et al, Nature, 2014, Astarita, Nature Immunology, 2015). In vivo experiments were repeated independently at least 2 times with N>=6 mice total per group. |
|---|---|
| Data exclusions | All data was included |
| Replication | Experiments were replicated independently at least 2 times, and data pooled for presentation. |
| Randomization | Animals were assigned experimental groups at random. In vitro experiment groups were defined by the genotype of the cell lines. |
| Blinding | Data collection and analysis were not performed blind to the conditions of the experiments. |

# Reporting for specific materials, systems and methods

We require information from authors about some types of materials, experimental systems and methods used in many studies. Here, indicate whether each material, system or method listed is relevant to your study. If you are not sure if a list item applies to your research, read the appropriate section before selecting a response.

## Materials & experimental systems

| n/a | Involved in the study |
|---|---|
| ☐ | ☒ Antibodies |
| ☐ | ☒ Eukaryotic cell lines |
| ☒ | ☐ Palaeontology |
| ☐ | ☒ Animals and other organisms |
| ☒ | ☐ Human research participants |
| ☒ | ☐ Clinical data |

## Methods

| n/a | Involved in the study |
|---|---|
| ☒ | ☐ ChIP-seq |
| ☐ | ☒ Flow cytometry |
| ☒ | ☐ MRI-based neuroimaging |

## Antibodies

| Antibodies used | Details included in Methods section of the manuscript. (Supplementary Table 1.) |
|---|---|
| Validation | All antibodies are commercially available and validated by the manufacturer and/or knockout cell lines. Details of Validation methods are found in Supplementary Table 1. |

## Eukaryotic cell lines

Policy information about cell lines

| Cell line source(s) | Details included in Methods section of the manuscript. Immortalised FRCs were generated as described in Acton et al 2014. Parental immortalised fibroblastic reticular cell line (Control FRC). Podoplanin (PDPN) was stably knocked down (PDPN KD FRC) in the parental cell line by transfection of a PDPN shRNA lentivirus. PDPN was completely depleted from the parental cell line (PDPN KO FRC) using CRISPR cas9 genetic deletion |
|---|---|
| Authentication | Original FRC cell lines is published in Acton et al Nature 2014, and authenticated (Karyotyping) by The Francis Crick Institute, London, UK. |
| Mycoplasma contamination | All cell lines are subject to mycoplasm testing, and found negative. |
| Commonly misidentified lines (See ICLAC register) | No commonly misidentified cell lines were used in this study |

# Animals and other organisms

Policy information about studies involving animals; ARRIVE guidelines recommended for reporting animal research

| | |
|---|---|
| Laboratory animals | Details included in Methods section of the manuscript. All C57BL/6J mice were purchased from Charles River Laboratories<br>Mouse, females, C57BL/6J, 8-12 weeks<br>Mouse, males & females, Pdgfra-mGFP-CreERT2 , 8-15 weeks<br>Mouse, males & females, PdgframGFPdelPDPN , 8-12 weeks<br>Mouse, males & females, PdgfraiR26R-Confetti, 8-12 weeks<br>Animal room conditions - ambient temperature 20-24 degrees centigrade, humidity 55%+/- 10%, 12hr dark light cycle with 30 minute dusk dawn ( at 7am and 7pm).<br>Animal cage conditions - Tecniplast blue line IVC (Individually ventilated cages), Lignocel select fine aspen sawdust with Arbocel comfort natural nesting, small aspen chew blocks and cardboard tubes, Fed ad lib Teklad global 18% Protein rodent diet (2018), UV filtered fresh water (autoclaved) |
| Wild animals | The study did not involve wild animals. |
| Field-collected samples | The study did not involve samples collected from the field. |
| Ethics oversight | All animal experiments were reviewed and approved by the Animal and Ethical Review Board (AWERB) within University College London on behalf of the Laboratory for Molecular and Cell Biology (LMCB) and approved by the UK Home Office in accordance with the Animals (Scientific Procedures) Act 1986 and the ARRIVE guidelines. |

Note that full information on the approval of the study protocol must also be provided in the manuscript.

# Flow Cytometry

## Plots

Confirm that:

☒ The axis labels state the marker and fluorochrome used (e.g. CD4-FITC).

☒ The axis scales are clearly visible. Include numbers along axes only for bottom left plot of group (a 'group' is an analysis of identical markers).

☒ All plots are contour plots with outliers or pseudocolor plots.

☒ A numerical value for number of cells or percentage (with statistics) is provided.

## Methodology

| | |
|---|---|
| Sample preparation | LNs were carefully dissected, weighed, and placed into RPMI 1640 media on ice. LNs were then processed as previously described 3,53. Briefly, LNs were placed into a digestion buffer containing collagenase D (250µg/ml) (Millipore Sigma), dispase II (800µg/ml) (Thermo Fisher Scientific) and DNase I (100µg/ml) (Sigma Aldrich). LNs were gently digested in a water bath at 37 ℃, removing and replacing the cell suspension every 10 minutes until completely digested. Cell suspensions were then centrifuged at 350g for 5 minutes. The cells were resuspended PBS, consisting of 1% BSA (Sigma Aldrich) and 5mM EDTA (Sigma Aldrich), and were filtered, counted, and resuspended at 10×10^6 cells/ml. 2.5×10^6 cells were seeded and stained for surface and intracellular markers for a stromal cell or T-cell panel (Antibody Table 1). Cells were blocked with CD16/CD32 Mouse Fc block (BD) and then stained with primary antibodies for 20 minutes at 4℃. For intracellular staining of Ki67 cells were fixed and permeabilised using FOXP3 fix/perm buffer as specified by the manufacturer (BioLegend). |
| Instrument | Samples were run on the Fortessa X20 flow Cytometer (BD Biosciences) at the UCL Cancer Institute. |
| Software | Flow cytometry data was collected using FACSDiva (Version DIVA 9)<br>Flow cytometry data was analysed using FlowJo (Version 10.7.1). |
| Cell population abundance | Abundance of relevant cell populations in post-sort fractions directly after flow sort was 95-100%. |
| Gating strategy | For all flow cytometry experiments, FSC/SSC gating was set to exclude cell debris (FCS/SSC-low population), followed by double gating on single cells. For flow cytometry analysis of in vivo immunization, immune cells were excluded by use of CD45, and the FRC fraction was determined as CD45-CD31-PDPN+. For details and full gating strategy, see Supplementary Figure 6. |

☒ Tick this box to confirm that a figure exemplifying the gating strategy is provided in the Supplementary Information.

