## [Peer Review File · Nature Immunology]

Peer Review Information

Journal: Nature Immunology

Manuscript Title: Lymph node tissue homeostasis and adaptation to immune challenge resolved by fibroblast network mechanics

Corresponding author name(s): Sophie Acton

Editorial Notes:

Redactions – unpublished data Parts of this Peer Review File have been redacted as indicated to maintain the confidentiality of unpublished data.

Reviewer Comments & Decisions:

Decision Letter, initial version:

Subject: Decision on Nature Immunology submission NI-LE32588

Message: 15th Oct 2021

Dear Sophie,

Thank you for providing your point-by-point response to your Letter entitled, "Lymph node tissue homeostasis and adaptation to immune challenge resolved by fibroblast network mechanics". As noted in my previous message, while the referees find your work of considerable potential interest, they have raised a number of substantial concerns that must be addressed. In light of these comments, we cannot accept the current manuscript for publication, but would be very interested in considering a revised version (in the expanded Article format) that addresses these concerns.

We invite you to submit a substantially revised manuscript, however please bear in mind that we will be reluctant to approach the referees again in the absence of major revisions.

Specifically, the revision should include new experiments to address:

(1) measurement of other LN parameters, such as the degree of T cell packing in the tissue, to validate findings from the quantitative measurements of LN tension.

(2) experiments using the inducible conditional KO mouse model (Pdgfra-mGFP-CreERT2 x PDPN^{f/f}) to validate the pharmacologic inhibitor studies; use this mouse model to compare network tension, kinetics of FRC proliferation & F-actin organization in FRCs, and the relationship to the underlying conduit in control vs PDPN-deleted LNs.

(3) examine actomyosin structures following changes to tissue tension by staining F-actin structures in FRCs immediately following ablation and upon inhibition with the ROCK inhibitor.

(4) quantify how proliferation/quantity of FRC correlates with tissue tension at multiple time points after immune challenge.

(5) quantify the numbers and phenotypes of T cells and how these are affected by changing FRC network tension, and therefore FRC proliferation.

(6) examine FRC proliferation in vitro in a range of different mechanical states: soft vs. rigid substrates, and determining whether treatment with ROCK inhibitor treatment alters proliferation of FRCs.

Please include the additional textual clarifications as indicated in your response letter.

When you revise your manuscript, please take into account all reviewer and editor comments, please highlight all changes in the manuscript text file in Microsoft Word format.

* If you have not done so already please begin to revise your manuscript so that it conforms to our Letter format instructions at <http://www.nature.com/ni/authors/index.html>. Refer also to any guidelines provided in this letter.

The Reporting Summary can be found here:
<https://www.nature.com/documents/nr-reporting-summary.pdf>

- that unprocessed scans are clearly labelled and match the gels and western blots presented in figures.
- that control panels for gels and western blots are appropriately described as loading on

sample processing controls

-- all images in the paper are checked for duplication of panels and for splicing of gel lanes.

[REDACTED]

If you wish to submit a suitably revised manuscript we would hope to receive it within 6 months. If you cannot send it within this time, please let us know. We will be happy to consider your revision so long as nothing similar has been accepted for publication at Nature Immunology or published elsewhere.

Nature Immunology is committed to improving transparency in authorship. As part of our efforts in this direction, we are now requesting that all authors identified as 'corresponding author' on published papers create and link their Open Researcher and Contributor Identifier (ORCID) with their account on the Manuscript Tracking System (MTS), prior to acceptance. ORCID helps the scientific community achieve unambiguous attribution of all scholarly contributions. You can create and link your ORCID from the home page of the MTS by clicking on 'Modify my Springer Nature account'. For more information please visit www.springernature.com/orcid.

Thank you for the opportunity to review your work.

Kind regards,

Laurie

Laurie A. Dempsey, Ph.D.
Senior Editor
Nature Immunology
l.dempsey@us.nature.com
ORCID: 0000-0002-3304-796X

Referee expertise:

Referee #1: tertiary lymphoid structures

Referee #2: tertiary lymphoid structures

Referee #3: tertiary lymphoid structures

Reviewers' Comments:

Reviewer #2:

Remarks to the Author:

Nature Immunology Manuscript

Horsnell et al.

Lymph node homeostasis and adaptation to immune challenge resolved by fibroblast network mechanics

The Horsnell et al., manuscript aims to better understand the mechanics and mechanism of lymph node swelling during the course of an immune response. First, the authors assess the mechanical properties of ex vivo mouse lymph nodes through a precise laser ablation system. Next, to decouple extracellular matrix and cellular-based tension in lymph node size, the authors use an immunization strategy with a pharmacological inhibitor of cytoskeletal contractility in combination with laser ablation. The authors then investigate the mechanism by which fibroblastic reticular cells (FRCs) regulate cell surface mechanics, as they posited that FRC are essential to lymph node swelling mechanics. To do so, they used an in vitro membrane tension system. Here, they show that podoplanin (PDPN)-CLEC2 signaling is essential to regulate FRC membrane tension and contend that this increase in plasma membrane surface area may be important to increase FRC-FRC cell contacts in vivo during lymph node expansion. Lastly, the authors delve into whether tissue tension may trigger FRC proliferation during lymph node expansion and find that pharmacological inhibition of ROCK, a key molecule for cytoskeletal contractility, does indeed limit FRC proliferation in vivo.

Overall, the authors nicely pair in vitro techniques with in vivo validation of their questions. The experimental results are properly controlled and well-conceived. Some major points would strengthen the current version of the manuscript:

1) The reader would be well-served if the authors expanded the figure and text length. Currently, the text is very economical but this makes aspects challenging to understand for an immunological audience that may not be versed in the cell-biology heavy techniques that are a primary focus of the paper.

2) The authors use targeted laser ablation and recoil measurements extensively. Are there orthogonal techniques that can validate this approach?

3) The authors rely heavily on pharmacologic inhibition to validate in vitro findings in vivo. This is insufficient as effects of ROCK inhibition may extend beyond FRC, particularly as other LN stromal cells express PDPN. The authors have made a new genetic tool (Pdgfra-mGFP-CreERT2) and this, or existing transgenic mice, could be used to nail their hypotheses. For example, does a Pdgfra-mGFP-CreERT2;PDPN^{flox} validate the findings from figure 4?

4) The findings would be more accessible and exciting to the reader if they were tied to the outcome of an immune response. For instance, the authors show that ROCK inhibition results in diminished FRC proliferation at day 5 upon immunization. Do the authors speculate that this diminished proliferation, which did not impair CD45⁺ cell numbers, would impact the outcome of an immune response? The paper would be strengthened if this were tested.

Minor points:

- 1) The authors need to specify the route and duration of tamoxifen exposure used for Pdgfra-mGFP-CreERT2 recombination.
- 2) As the authors exclusively assay LN fibroblasts in the manuscript, the authors may consider using these cells, rather than ear fibroblasts in Figure S1.
- 3) The authors may want to consider BrdU/EdU as a proliferative measure for FRCs upon immunization as the Ki67 staining does not provide a clear positive population (though there is signal above FMO).

Reviewer #3:

Remarks to the Author:

Scope: In this manuscript the authors describe molecular signals controlling cellular physical properties determining the tissues mechanics of lymph nodes. The team proposes that cytoskeletal mechanics of the cellular mesh work of fibroblastic reticular cells determines tissue tension independently of the ECM scaffold. Specifically the CLEC-2/podoplanin signaling regulates the fibroblasts that are responsible mediating tissue expansion.

Significance: The significance of this work describes how fibroblastic reticular cell structure of the lymph node is the active mechanical component during tissue expansion. It suggests cellular structures vs underlying ECM tissue mechanics plays a role in expansion in this highly cellularized system.

Major Findings:

The major findings of this study include determining the fibroblastic reticular cell structure of the lymph node is the active mechanical component during tissue expansion. This is mediated through CLEC-2/PDPN signaling priming the cell surface of FRCs. FRCs divide to restore the architecture as expansion increases tissue tension. Overall, this project investigates an interesting and unsolved problem that has implications in disease and therapy. For this reason it is interesting. However many of the conclusions made in the manuscript are only lightly supported by the evidence provided and the many complex hypotheses and corresponding experiments used to test them are not systematic enough to rigorously conclude what is stated in the manuscript. More work would need to be done to fully conclude each of the major points made. It is suggested that more evidence of each mechanism be presented in multiple experimental systems in order to adequately support the stated conclusions.

Major Critiques:

1. Need to stain for other ECM components (collagen, laminin, entactin etc.), not just perlecan at steady state and over 5 day period to validate that tension occurs independently of extracellular matrix integrity. Need to validate multiple components of the ECM is fragmented and not responsible for increased tissue tension in conjunction with FRC meshwork maintenance over the same period of time.
2. ECM integrity should be quantified through mechanical testing to determine fragmentation and interaction with FRC
3. Need to include CD3 stain and quantification compared to steady state over 5-day period demonstrating increased lymphocyte packing in FRC network upon immunogenic challenging. Not sufficient to suggest mechanical forces are generated by increased packing of lymphocytes in the FRC meshwork with no quantification of lymphocyte present in the FRC meshwork.

4. Should quantify CD3+ cells after ROCK inhibition to validate lymphocyte proliferation is unaffected after 5 days. CD45 is a broad hematopoietic lineage marker, and the region of interest is the paracortex which is dominated by t cells, so CD3 should be quantified to determine if there is a reduction in the cell population most directly correlated with the region of interest.
5. FRC mesh size should be quantified at steady state, day 3 and day 5 timepoint to determine if there are changes in average mesh size over the course of immune challenging that are affecting the mechanical tissue properties. How does the mesh size change in response to the CLEC-2/podoplanin signaling
6. Need to quantify how proliferation/quantity of FRC correlates with tissue tension at multiple time points after immune challenging

Minor Critiques:

1. Should define nDr and Dr in text or figure captions (figure 2)
2. Should be consistent with abbreviations nDr or nd in figure caption (figure 3 C says nDr and G-I say nd)

Reviewer #4:

Remarks to the Author:

Summary

This is an interesting study highlighting the mechanical forces involved in lymph node expansion, particularly through the FRC network during remodeling. The study significantly advances our understanding of lymph node mechanics and the underlying mechanisms involved, including F-actin fibers around FRCs, matrix integrity, and CLEC2-DC mediated regulation of FRC membrane mechanics that contribute to overall FRC fibril tension. The work shows beautifully executed ablation studies that demonstrate changes in FRC tension throughout the remodeling process and clear involvement of CLEC2 binding on membrane tension of FRCs that appears to be mediated by CD44-PDPN interactions. The authors also demonstrate dependence of FRC swelling and contraction on CLEC2, and that mechanical tension (removed through a ROCK inhibitor) enhances FRC proliferation. The work is novel and methodology, including statistics, is appropriate. Conclusions are mostly robust, though several experiments (see below) should be performed to strengthen the key points. The manuscript is well written, though particularly Figure 2 needs to be improved in clarity (images + explanation). References are appropriate.

Suggested improvements:

- How the F-actin cables and pMLC2 are disrupted during laser ablation needs to be included to better understand the mechanical dynamics and changes with treatment; this could be done by including post ablation staining similar to 1F and needs to be added in the text.
- There needs to be discussion of how the LN slice culture model could potentially impact the architecture/network and FRC integrity; how long can these be cultured before network integrity is altered and how quickly are the experiments performed compared to changes. If there are changes to the architectural integrity, this could significantly affect the experimental readouts
- A key point missing is an explanation for why the recoil speed is reduced at day 3, which as the author note, is surprising/unexpected.
- Figure 2E:

- o Overlay images in Figure 2E are very difficult to decipher with the description in the text as to what the reader should be looking for specifically; since this should be written for a very broad audience the description needs to be more clear
- o How was the absence of T cell facing F-actin bundles assessed (Fig 2E)?
- o It looks like at day 3, the F-actin also overlaps with some of the PDPN indicating contractile cables span the FRC bodies here as well. To show that there is a difference between day 3 and day 5 more than qualitative single images need to be provided; the number of fibers overlapping with PDPN+ FRCs that include matrix vs not at day 3 vs 5 needs to be quantified
- Considering that ablations were performed specifically around FRC bundles, I'm not sure generalizations about overall tissue tension can be made from these single ablation studies. The authors should carefully reword their conclusions in the paragraph on Fig 2 to reflect their data showing FRC network related tension
- If lymphocyte packing is thought to be the main reason for increased mechanical forces, the authors should show an increase in lymphocytes between FRC network with immunization
- Fig 3C/SF2: ROCK inhibition does not appear to change FRC network tension at day 3 – the curves are almost overlapping; this data needs to be explained and the differences between day 3 and 5 need to be elucidated
- Fig 3G: in vivo increase in FRC cell-cell contact should be quantified
- The conclusion that additional cell-cell contact means that “additional plasma membrane is used for morphological adaptations and incorporated into cell extensions” is not most effectively supported by the data. How does the in vitro data integrate with the in vivo findings of more cell-cell contact? This needs to be discussed more effectively in the text
- Figure 4 does not include lymphocyte staining or markers of proliferation on lymphocytes; authors should include T + B cell staining + Ki67 with and without rock inhibitors at day 0 and day 5
- Figure 4: to truly demonstrate that tension is required to reduce the proliferative potential of FRCs, Ki67 staining in FRCs should also be performed with and without Y27 at days 0 and 5 compared to untreated control. This would strengthen the point.
- To make the conclusion that mechanical stimulus is responsible for FRC expansion, additional experiments directly demonstrating this in FRCs in vitro without surrounding inflammatory response would be helpful in solidifying this point. Experiments could be done to change tension experienced by FRCs by placing them on a substrate and modulating tension on substrate and subsequently assessing Ki67 expression and overall proliferation
- Conclusions about only the FRC is the mechanical component during tissue expansion are a little strong considering studies largely focused on ex vivo recoil experiments to demonstrate tissue tension and did not correlate this with whole tissue changes, etc. The reviewer realizes that the correlation with whole tissue is not a feasible experiment, but conclusions should be adjusted accordingly not to overstate the findings
- Indication that the laser ablation experiments in Figure 1 are done on live slice cultures needs to be included in the text and figure description.

Author Rebuttal to Initial comments

Point-by-point response - NI-LE32588,

Lymph node tissue homeostasis and adaptation to immune challenge resolved by fibroblast network mechanics

We thank the reviewers for their comments and suggestions which we found constructive and beneficial for our study. We have fully addressed all of the points as detailed below.

In summary, we have now included quantification of additional mechanical parameters – T cell packing, gap analysis – to validate and support our measurements of LN tissue tension. We have extended our examination of actin structures in FRCs in LNs and quantified their contractile status and relationship to the underlying conduits. We have examined FRC proliferation in a range of mechanical states and shown that both PDPN signaling and actomyosin contractility are important for the intrinsic mechanical sensitivity of FRCs. The most significant change to the manuscript is the addition of experiments using conditional deletion of PDPN in vivo. Using this model we have measured network tension, FRC proliferation and F-actin organization. These experiments significantly strengthen our original findings and provide an interesting and important comparison to the pharmacological studies. We have been able to use both systems side by side to quantify how FRC proliferation correlates with tissue tension through immune challenge. All changes to the text are highlighted to make clear how the additional data have been incorporated into this revised manuscript.

Reviewer #2

Overall, the authors nicely pair in vitro techniques with in vivo validation of their questions. The experimental results are properly controlled and well-conceived. Some major points would strengthen the current version of the manuscript:

1) The reader would be well-served if the authors expanded the figure and text length. Currently, the text is very economical but this makes aspects challenging to understand for an immunological audience that may not be versed in the cell-biology heavy techniques that are a primary focus of the paper.

Through this review process we have extended the manuscript and now submit in a longer format as an article. The revision we now submit includes 4 additional figures addressing the reviewers comments.

2) The authors use targeted laser ablation and recoil measurements extensively. Are there orthogonal techniques that can validate this approach?

Laser ablation is the gold-standard for taking quantitative measurements of tension within tissues, and is widely used by biophysicists for this purpose. We have edited the text to make clear that these measurements provide mechanical data specific to the FRC network, which spans the whole tissue. These recoil measurements do not include direct measurement of other whole tissue parameters such as the extracellular matrix of the bounding

capsule or the degree of lymphocyte packing in the tissue. Additional data relating to the degree of T cell compaction, as also suggested by reviewer 3 is now included in figure 2 to address this point and strengthen the findings.

3) The authors rely heavily on pharmacologic inhibition to validate *in vitro* findings *in vivo*. This is insufficient as effects of ROCK inhibition may extend beyond FRC, particularly as other LN stromal cells express PDPN. The authors have made a new genetic tool (Pdgfra-mGFP-CreERT2) and this, or existing transgenic mice, could be used to nail their hypotheses. For example, does a Pdgfra-mGFP-CreERT2;PDPN^{flox} validate the findings from figure 4?

This is an excellent suggestion, and we have added this to the revised manuscript. We have crossed PDPN-floxed mice (kindly shared by Prof Steve Watson and Prof Chris Buckley) to a new (unpublished) Pdgfra-mGFP-CreERT2 model which our group has generated. This provides the ideal conditional system to directly test the role for PDPN signalling in FRC network mechanics. We have added the following experiments to the manuscript:

- Comparison network tension of control vs. PDPN-deleted LNs at steady state, and through early lymph node expansion – **new Figure 6**
- Comparison of the FRC population expansion in control vs PDPN-deleted LNs – **new Figure 7**
- Comparison of F-actin organisation in in control vs PDPN-deleted FRCs, and the relationship between FRCs to the underlying conduit in control vs PDPN-deleted LNs – **new Figure 6**

These new figures also include relevant controls to show the efficacy of podoplanin deletion at the relevant time points both by flow cytometry and immunofluorescence tissue staining. We find We find that both ROCK inhibition, targeting actomyosin contractility, and PDPN deletion alter FRC proliferation *in vitro* and both pathways are required for FRCs to sense the rigidity of their environment. We also note several difference between these two perturbations and have added a paragraph to discuss these observations (**lines 383-398**).

We thank the reviewer for this suggestion. These experiments were ambitious within the timeframe but have significantly strengthened our manuscript.

4) The findings would be more accessible and exciting to the reader if they were tied to the outcome of an immune response. For instance, the authors show that ROCK inhibition results in diminished FRC proliferation at day 5 upon immunization. Do the authors speculate that this diminished proliferation, which did not impair CD45⁺ cell numbers, would impact the outcome of an immune response? The paper would be strengthened if this were tested.

In the original version we presented only simplified data of immune cells versus stromal cells. We agree that adding data quantifying the numbers and phenotypes of T cells and how these are affected by changing FRC network

tension, and therefore FRC proliferation, is an important addition to the manuscript. We have added the following experiments to the manuscript:

- We have repeated the *in vivo* experiments using Y27632 ROCK inhibitor adding details of T cell:B cell ratio (CD3, CD19), along with analysis of T cell subset and activation status e.g. CD4, CD8, CD44, CD62L to **new Figure 8**.
- In addition, we have included data on the impact of FRC-specific deletion of PDPN expression on the outcome of immune responses – **new Figure 7**.

Minor points:

1) The authors need to specify the route and duration of tamoxifen exposure used for *Pdgfra*-mGFP-CreERT2 recombination.

Apologies for the omission, the confetti cassette was induced following 3 x I.P. doses of tamoxifen, prior to immunisation. The mGFP is expressed without induction. We have included details of how the tamoxifen exposure was used to conditionally delete podoplanin expression in a revised version (**lines 418-420**).

2) As the authors exclusively assay LN fibroblasts in the manuscript, the authors may consider using these cells, rather than ear fibroblasts in Figure S1.

We now also show LN examples for this validation, and we have validated this model in multiple tissues. We had wanted to make clear that the mGFP driven by *PDGFR α* is widely expressed in all tissue resident fibroblast populations. We have edited **Figure S1** to show GFP expression in FRCs to be more relevant and specific to the study.

3) The authors may want to consider BrdU/EdU as a proliferative measure for FRCs upon immunization as the Ki67 staining does not provide a clear positive population (though there is signal above FMO).

We chose Ki67 over BrdU/EdU since it will broadly label all cells in G1, S G2 and mitosis, i.e. any stage of cell cycle. Since in steady state, the majority of FRCs exist in a quiescent state, Ki67 has the benefit of showing the % of actively cycling FRCs rather than just those cells undergoing DNA replication. Our analysis always uses FMO controls to accurately gate the proliferative populations. We have extend the text at the beginning of new Figure 5 to add clarity to this point (**lines 211-217**)

Reviewer #3

Overall, this project investigates an interesting and unsolved problem that has implications in disease and therapy. For this reason it is interesting. However many of the conclusions made in the manuscript are only lightly supported by the evidence provided and the many complex hypotheses and corresponding experiments used to test them are not systematic enough to rigorously conclude what is stated in the manuscript.

We are pleased that the reviewer find this study interesting and agrees that we address an unsolved problem and is therefore novel. The reviewer is correct that this manuscript does not address the details of ECM regulation in the conduit network, which is an essential starting point and context for this study. However, we have undertaken extensive analysis of the conduit components and fragmentation through LN expansion in another recent study. We have made these findings clearer throughout the text to properly contextualise the work we present in this manuscript (**lines 124-130**)

Major Critiques:

1. Need to stain for other ECM components (collagen, laminin, entactin etc.), not just perlecan at steady state and over 5 day period to validate that tension occurs independently of extracellular matrix integrity. Need to validate multiple components of the ECM is fragmented and not responsible for increased tissue tension in conjunction with FRC meshwork maintenance over the same period of time.

This is an important point, and highly relevant to the mechanical data we present. However the points raised here are fully addressed in our recent published study – Martinez *et al.* Cell Reports 2019

[https://www.cell.com/cell-reports/pdf/S2211-1247\(19\)31436-6.pdf](https://www.cell.com/cell-reports/pdf/S2211-1247(19)31436-6.pdf)

In this published work we show that multiple components of the conduit network are fragmented and reduced in quantity over the same time course. Further, we show that the fragmentation of the conduit ECM components leads to a partial disruption of conduit flow of small molecules through the tissue. We were able to elucidate a general mechanism whereby FRCs reduced adhesion with the ECM, and through regulation of microtubule networks, reduced the deposition of matrix throughout the early phases of tissue expansion.

Figures 5E and 5F from Martinez et al. 2019 – showing conduit fragmentation and reduced association between FRCs and underlying matrix following immune challenge (day5)

Figure 6A from Martinez et al. 2019 – Showing partial disruption of conduit flow following immune challenge, while FRC network remains intact.

Since this work is a key to LN mechanics we have edited the text to make the known and published work regarding the fragmentation of the conduits much clearer (lines 124-130).

Importantly, we have also added data to the revised manuscript showing how individual FRCs remain in their network positions despite fragmentation of the underlying ECM conduit scaffolds – **New Figure 3D**. Additionally we have quantified the relationship between F-actin cables in the FRC network and the underlying conduit in control and Y27632 treated lymph nodes – **New Figure 3A, 3I**.

2. ECM integrity should be quantified through mechanical testing to determine fragmentation and interaction with FRC

As described above, this has been addressed by making clearer the relationship of this manuscript to our previous published study.

3. Need to include CD3 stain and quantification compared to steady state over 5-day period demonstrating increased lymphocyte packing in FRC network upon immunogenic challenging. Not sufficient to suggest mechanical forces are generated by increased packing of lymphocytes in the FRC meshwork with no quantification of lymphocyte present in the FRC meshwork.

We agree that this is an important point and adding this data strengthens our findings.

We have shown and quantified the density and ‘packing’ of T cells within the FRC network over the time course (days 0, 3, 5) as suggested using immunofluorescence staining (CD3, PDPN, nuclear markers) – **New Figure 2B,C**. We have presented this data alongside quantification of FRC network mesh size – **new Figure 2D,E** which when combined with T cell packing data give an important overall measurement of forces in the tissue.

4. Should quantify CD3+ cells after ROCK inhibition to validate lymphocyte proliferation is unaffected after 5 days. CD45 is a broad hematopoietic lineage marker, and the region of interest is the paracortex which is dominated by t cells, so CD3 should be quantified to determine if there is a reduction in the cell population most directly correlated with the region of interest.

This experiment is also suggested by reviewer 2, and we agree it is an important addition.

- We have repeated the *in vivo* experiments using Y27632 ROCK inhibitor adding details of T cell:B cell ratio (CD3, CD19), along with analysis of T cell subset and activation status e.g. CD4, CD8, CD44, CD62L to **new Figure 8**.
- In addition, we have included data on the impact of FRC-specific deletion of PDPN expression on the outcome of immune responses – **new Figure 7**.

5. FRC mesh size should be quantified at steady state, day 3 and day 5 timepoint to determine if there are changes in average mesh size over the course of immune challenging that are affecting the mechanical tissue properties.

This data has been included alongside the quantification of lymphocyte packing (point 3). In 2014 we developed an automated mathematical method, we termed ‘gap analysis’ to quantify mesh size in the FRC network. We used this method in our study which first showed the critical interaction between CLEC-2⁺ DC and FRCs controlling LN expansion. (Acton *et al.* Nature 2014)

<https://www.nature.com/articles/nature13814.pdf>

Figure 3f from Acton *et al.* Nature 2014 – Showing increased ‘mesh size’ following immune challenge (day5)

In our revised manuscript we present and quantify the mesh size at steady state, day 3 and day 5 timepoints – **New Figure 2D,E**. These data are consistent with the observations first published in 2014, but the addition of the day 3 data along with the T cell packing data have been useful additions to help us to interpret the relationship between external forces exerted by T cells and the cell intrinsic mechanics of the FRCs.

How does the mesh size change in response to the CLEC-2/podoplanin signalling

This is an interesting question, and relevant to the mechanics of the LN tissue. This data is also already published in the publication above (Acton *et al.* Nature 2014). We showed that mesh size increased 5 days post immunisation in control LNs. CD11c^{ΔCLEC-2} mice failed to expand their LNs tissue to the same degree as control mice, which corresponded with a reduced FRC network mesh size (Figure 4c from Acton *et al.* Nature 2014).

Figure 4c-e from Acton et al. Nature 2014 – Showing impact of CLEC-2 expression in dendritic cells in FRC mesh size, LN deformability and LN expansion.

This key data, already published, is now made clearer in the introduction (lines 35-40), to better put our mechanical questions in context. We also draw comparison of LN remodelling between FRC-specific deletion of PDPN and dendritic cell deletion of CLEC-2 in the main text (lines 257-259).

6. Need to quantify how proliferation/quantity of FRC correlates with tissue tension at multiple time points after immune challenging

This correlation makes the overall message much clearer and is a good additional representation of the data. We have added, as suggested, xy plots relating tissue tension to FRC number – **New Figure 8**. This is shown for steady state and following immunisation, and includes changes to both network tension and FRC proliferation following treatment with ROCK inhibitor Y27632 and PDPN deletion. We have focussed on early timepoints, since our data identify increased network tension as a trigger for cell cycle entry, rather than a continuous signal to sustain proliferation of the FRC population. This distinction is now also better explained in the main text (lines 334-338) the discussion (383-398) since we have extended the revised manuscript be a full article.

Minor Critiques:

1. Should define nDr and Dr in text or figure captions (figure 2)

2. Should be consistent with abbreviations nDr or nd in figure caption (figure 3 C says nDr and G-I say nd)

These have been edited, we apologies for the inconsistency.

Reviewer #4

The study significantly advances our understanding of lymph node mechanics and the underlying mechanisms involved, including F-actin fibers around FRCs, matrix integrity, and CLEC2-DC mediated regulation of FRC membrane mechanics that contribute to overall FRC fibril tension. The work shows beautifully executed ablation studies that demonstrate changes in FRC tension throughout the remodeling process and clear involvement of CLEC2 binding on membrane tension of FRCs that appears to be mediated by CD44-PDPN interactions. The authors also demonstrate dependence of FRC swelling and contraction on CLEC2, and that mechanical tension (removed through a ROCK inhibitor) enhances FRC proliferation. The work is novel and methodology, including statistics, is appropriate. Conclusions are mostly robust, though several experiments (see below) should be performed to strengthen the key points. The manuscript is well written, though particularly Figure 2 needs to be improved in clarity (images + explanation). References are appropriate.

Suggested improvements:

- How the F-actin cables and pMLC2 are disrupted during laser ablation needs to be included to better understand the mechanical dynamics and changes with treatment; this could be done by including post ablation staining similar to 1F and needs to be added in the text.

We agree that looking at actomyosin structures following changes to tissue tension would nicely link the manuscript back to the initial observations in figure 1. We have attempted to stain F-actin structures in FRCs immediately following ablation to address this point but have unfortunately found this to be technically impossible since the laser ablation requires fast scanning using a different imaging objective that we would need to gain the resolution of cytoskeletal structures.

What we have included is F-actin and pMLC staining of LN treated with ROCK inhibitor Y27632 which directly links the reduced tension measurements from the laser ablation studies to the cytoskeletal structures – **New Figure 3H-J**. Beyond this, we have repeated the imaging and quantification of F-actin and pMLC staining in the FRC network in PDPN-deleted lymph nodes - **New Figure 6**, and made comparisons between Y27632 and PDPN deletion throughout the manuscript.

- There needs to be discussion of how the LN slice culture model could potentially impact the architecture/network and FRC integrity; how long can these be cultured before network integrity is altered and how quickly are the experiments performed compared to changes. If there are changes to the architectural integrity, this could significantly affect the experimental readouts

The methods section has been edited to make clear these are live ex-vivo tissue sections that we use in the manuscript (**Lines 435-443**). In figure 1B we show that the structure of the vibrotome sections is intact and maintains lymphocyte packing.

Briefly, ablation experiments are conducted within 2-4 hours after sectioning. The tissue sections are sufficiently thick that we can perform the ablations >50 microns from the cut surface. The main benefit of this technique is that we can ensure we are ablating FRCs deep within the paracortex, well away from the LN capsule. Attempting to perform laser ablation on intact lymph nodes limits the depth of tissue we were able to penetrate and measurements using this method would be much closer to the lymph node capsule meaning that measurements are reading out tension in either capsular fibroblasts or lymphatic structures which was not the aim of our study.

- A key point missing is an explanation for why the recoil speed is reduced at day 3, which as the author note, is surprising/unexpected.

We would explain the reduced recoil at day 3 through the changing cell surface mechanics of FRCs, which is investigated in detail in the original figure 3 – now **new Figure 4**. In the extended main text, we have explained this better to link better the in vivo and in vitro experiments (**lines 168-170**). The addition of experiments in LNs where PDPN is conditionally deleted (as suggested by reviewer 2 above) have also addressed this query.

- Figure 2E:
 - o Overlay images in Figure 2E are very difficult to decipher with the description in the text as to what the reader should be looking for specifically; since this should be written for a very broad audience the description needs to be more clear

We have added additional description and quantification to improve clarity to this important point. We also feel that the supplementary movies that accompany this figure aid clarity since the rotation in these movies makes the spatial orientation of these complex structures much easier to understand.

- o How was the absence of T cell facing F-actin bundles assessed (Fig 2E)?

It looks like at day 3, the F-actin also overlaps with some of the PDPN indicating contractile cables span the FRC bodies here as well. To show that there is a difference between day 3 and day 5 more than qualitative single images need to be provided; the number of fibers overlapping with PDPN+ FRCs that include matrix vs not at day 3 vs 5 needs to be quantified

This is a good point and important data to add to the manuscript. We have included quantification as suggested of F-actin fibres in FRCs, comparing the proportion of F-actin fibres that are associated with the underlying ECM (perlecan) of the conduit – **New Figure 3**. We agree that adding quantification to this figure has increased the impact of this data.

- Considering that ablations were performed specifically around FRC bundles, I'm not sure generalizations about overall tissue tension can be made from these single ablation studies. The authors should carefully reword their conclusions in the paragraph on Fig 2 to reflect their data showing FRC network related tension

We have reworded our conclusions and to make clear that we are measuring network tension specifically.

- If lymphocyte packing is thought to be the main reason for increased mechanical forces, the authors should show an increase in lymphocytes between FRC network with immunization

This is a good suggestion, which reviewer 3 also recommends. As we detail above, we have added quantification of T cell 'packing' and FRC mesh size to the new **Figure 2**.

- Fig 3C/SF2: ROCK inhibition does not appear to change FRC network tension at day 3 – the curves are almost overlapping; this data needs to be explained and the differences between day 3 and 5 need to be elucidated

We would interpret these data to mean that at day 3, actomyosin does not contribute the network tension at this time point. Network tension is very low at this time point, but other mechanisms and cellular structures must contribute to residual network tension at this point. We explain the reduction in network tension with our experiments examining the regulation of FRC cell surface mechanics in the original figure 3 – now **new Figure 4**. Adding analysis of T cell packing over the time course, and examining the phenotype of PDPN-deleted FRC networks has also helped to highlight the differences between day 3 and day 5.

- Fig 3G: in vivo increase in FRC cell-cell contact should be quantified

We agree that as shown the increase in FRC protrusion and interdigitation with neighbouring cells lacks quantification. Very little is known about the cell-cell junctions between FRCs and our data show the unexpected complexity and high number of FRC protrusions that interact between neighbouring cells during lymph node remodelling. While we have the resolution to identify these fine protrusions for the first time we don't think we can accurately quantify the area of FRC cell-cell contact using this assay. Therefore we have adjusted our description of this data accordingly.

- The conclusion that additional cell-cell contact means that “additional plasma membrane is used for morphological adaptations and incorporated into cell extensions” is not most effectively supported by the data. How does the in vitro data integrate with the in vivo findings of more cell-cell contact? This needs to be discussed more effectively in the text

We have done as suggested and edited the text (**lines 203-2-6**). The statement quoted is indeed a hypothesis since we cannot directly link the in vitro data with the changes in FRC protrusions in the tissue.

- Figure 4 does not include lymphocyte staining or markers of proliferation on lymphocytes; authors should include T + B cell staining + Ki67 with and without rock inhibitors at day 0 and day 5

We agree that this experiment would make a very good addition to a revised manuscript and similar experiments are also suggested by the other reviewers.

What we have now included:

- We have repeated the *in vivo* experiments using Y27632 ROCK inhibitor adding details of T cell:B cell ratio (CD3, CD19), along with analysis of T cell subset and activation status e.g. CD4, CD8, CD44, CD62L to **new Figure 8**.
 - In addition, we have included data on the impact of FRC-specific deletion of PDPN expression on the outcome of immune responses – **new Figure 7**.
- Figure 4: to truly demonstrate that tension is required to reduce the proliferative potential of FRCs, Ki67 staining in FRCs should also be performed with and without Y27 at days 0 and 5 compared to untreated control. This would strengthen the point.

We have added (as also suggested by Reviewer 3), xy plots relating tissue tension to FRC number – **New Figure 8**. This is shown for steady state and following immunisation, and includes changes to both network tension and FRC proliferation following treatment with ROCK inhibitor Y27632 and PDPN deletion. We have focussed on early timepoints, since our data identify increased network tension as a trigger for cell cycle entry, rather than a continuous signal to sustain proliferation of the FRC population.

- To make the conclusion that mechanical stimulus is responsible for FRC expansion, additional experiments directly demonstrating this in FRCs in vitro without surrounding inflammatory response would be helpful in solidifying this point. Experiments could be done to change tension experienced by FRCs by placing them on a substrate and modulating tension on substrate and subsequently assessing Ki67 expression and overall proliferation

To address this point directly we would need to be able to model the quiescent state of steady state FRCs *in vitro*. The FRC cell lines we used are immortalised (using E6) and are therefore proliferative and remain in the cell cycle. Freshly isolated primary FRCs also quickly enter the cell cycle when isolated from tissues and plated onto rigid substrates. This phenomenon is interesting however, and perhaps indeed shows that FRC proliferation is sensitive to mechanical forces (through sensing substrate rigidity). In other contexts and various cell types, *in vitro* experiments have shown that stretching forces and substrate rigidity are well established to impact proliferation.

We have addressed this query by examining FRC proliferation *in vitro* in a range of different mechanical states: soft vs. rigid substrates, and determining whether treatment with ROCK inhibitor or PDPN expression alters proliferation of FRCs. These data are presented in **new Figure 5**. We find that FRC proliferation is significantly affected by the mechanical properties of the substrate. Further that the mechanical sensitivity of FRC proliferation is attenuated by both Y27632 treatment and PDPN KO, the two mechanical pathways we have examined *in vivo*. Beyond addressing the original reviewers comment we also present data showing that surface expression of PDPN protein is also sensitive to cell mechanics and is significantly reduced when cultured in the presence of Y27632 ROCK inhibitor. We think these data add additional breadth to the study and are therefore also included in **new Figure 5**.

- Conclusions about only the FRC is the mechanical component during tissue expansion are a little strong considering studies largely focused on *ex vivo* recoil experiments to demonstrate tissue tension and did not correlate this with whole tissue changes, etc. The reviewer realizes that the correlation with whole tissue is not a feasible experiment, but conclusions should be adjusted accordingly not to overstate the findings

Yes we are in agreement and we have edited the text to refer to network tension where appropriate.

- Indication that the laser ablation experiments in Figure 1 are done on live slice cultures needs to be included in the text and figure description.

This is clarified in the revised manuscript (**lines 435-443**).

Decision Letter, first revision:

Subject: Decision on Nature Immunology submission NI-LE32588A

Message: 6th May 2022

Dear Sophie,

Thank you for providing a rebuttal/response to the referees' comments on your revised manuscript entitled, "Lymph node tissue homeostasis and adaptation to immune challenge resolved by fibroblast network mechanics". We are very interested in the possibility of publishing your study in Nature Immunology, and would like you to revise the manuscript

as indicated in your response.

We therefore invite you to revise your manuscript taking into account all reviewer and editor comments. Please highlight all changes in the manuscript text file in Microsoft Word format.

Specifically, the revision should include new experiments to address:

- (1) Add the flow cytometry data showing the CD44+CD62L- effector T cell populations as indicated.
- (2) Add the day 3 ECM comparison to the day 5 ECM analysis as indicated.
- (3) Add the images/quantification of the CD3+ Ki67+ cells as indicated.
- (4) Add the data on recruitment of the CD44+CD62L+ central memory T cells.

I suggest to include in the discussion that PDPN missense mutations/SNPS have been noted in the cbioportal.org database, [REDACTED].

Also, fine to expand discussion regarding the effects of adjuvants versus cellular infiltration & relevant time scales for responses, as well as discussion of changes to focal adhesions, etc as noted in your response.

Please include the additional textual clarifications as indicated in your response letter.

When you revise your manuscript, please take into account all reviewer and editor comments, please highlight all changes in the manuscript text file in Microsoft Word format.

* If you have not done so already please begin to revise your manuscript so that it conforms to our Letter format instructions at <http://www.nature.com/ni/authors/index.html>. Refer also to any guidelines provided in this letter.

* Please include a revised version of any required reporting checklist. It will be available to referees to aid in their evaluation of the manuscript goes back for peer review. They are available here:

Reporting summary:

When submitting the revised version of your manuscript, please pay close attention to our

<https://www.nature.com/nature-portfolio/editorial-policies/image-integrity>>Digital Image Integrity Guidelines. and to the following points below:

Please use the link below to submit your revised manuscript and related files:
[REDACTED]

We hope to receive your revised manuscript within four weeks. If you cannot send it within this time, please let us know. We will be happy to consider your revision so long as nothing similar has been accepted for publication at Nature Immunology or published elsewhere.

Nature Immunology is committed to improving transparency in authorship. As part of our efforts in this direction, we are now requesting that all authors identified as 'corresponding author' on published papers create and link their Open Researcher and Contributor Identifier (ORCID) with their account on the Manuscript Tracking System (MTS), prior to acceptance. ORCID helps the scientific community achieve unambiguous attribution of all scholarly contributions. You can create and link your ORCID from the home page of the MTS by clicking on 'Modify my Springer Nature account'. For more information please visit <http://www.springernature.com/orcid>.

Kind regards,

Laurie

Laurie A. Dempsey, Ph.D.
Senior Editor
Nature Immunology

l.dempsey@us.nature.com
ORCID: 0000-0002-3304-796X

Reviewers' Comments:

Reviewer #2:

Remarks to the Author:

Nature Immunology Manuscript

Horsnell et al.

Lymph node homeostasis and adaptation to immune challenge resolved by fibroblast network mechanics - Resubmission

The resubmission of the Horsnell et al., results in an improved manuscript – the authors have done a lot of work and should be applauded. As this is a resubmission, I am omitting incorporation elements that were in the original critique. Indeed, they have gone far to describe the cell biology of fibroblastic reticular cells, which through direct and indirect mechanisms are key to immune responses. Yet, while the cell biology is very nice, the immunology seems to be second to this. As above, I concede that the authors are studying a cell type key to immunity but I am not sure the study in its current form sufficiently falls within the breadth of interest of the intended audience.

To make this more relevant to the average immunologist, we raised this point during the initial submission:

The findings would be more accessible and exciting to the reader if they were tied to the outcome of an immune response. For instance, the authors show that ROCK inhibition results in diminished FRC proliferation at day 5 upon immunization. Do the authors speculate that this diminished proliferation, which did not impair CD45+ cell numbers, would impact the outcome of an immune response? The paper would be strengthened if this were tested.

In the resubmission, the authors used CFA/OVA in their fabulous genetic model (Pdgfrac^{cre};PDPN^{flox}) and upon ROCK inhibition to evidence that their findings are immunologically-relevant. Indeed, they see alterations in T and B cells and T cell activation, but no disease/infection/autoimmunity relevance is provided. Indeed, are ROCK-inhibited, or Pdgfrac^{cre};PDPN^{fl} mice, more susceptible to viral infection or disease indications or are these findings relevant for an autoimmune context? If no mouse studies will be pursued to this point, are single nucleotide polymorphisms in PDPN associated in humans with disease?

Additionally, the data as presented leave the reader wondering about the mechanism underpinning their IFA/OVA experiments - the authors state on line 53 that alterations to FRCs “directly inhibit T cell activation and proliferation.” Yet, their data do not describe how this direct interaction occurs. Based upon the authors previous works, it stands to reason that this would be indirect and any T cell phenotypes would be due to an inability of DCs to adequately prime T cells, due to a loss of CLEC2-PDPN interactions in vivo. It seems salient to investigate this possibility - examination of CD25 is unconvincing on this point, particularly as the data the authors reference is not statistically significant – or to provide other readouts to show the mechanism by which lack of PDPN on FRC or ROCK inhibition can mechanistically alter T or B cell responses.

Reviewer #3:

Remarks to the Author:

The authors have substantially revised the manuscript, and it is much more clearly written, and the methods used and how they are appropriate to probe certain questions is dramatically clearer, as are how the data is then interpreted. They have also added more data that support the central claims, much strengthening the report. Overall, the revision is much stronger and provides a compelling case for the interpretation presented. While the current revisions have addressed some of the major concerns and improved upon the previous work, there are still some claims that are not rigorously supported by the current set of studies.

Major Critiques

1. Authors state that ECM fragmentation occurs but do not quantify the degree of fragmentation on d3 vs d5. While the ECM is remodeling, is there increased ECM protein expression on d5 than on d3, or a difference in the size of the fragments. Could this explain the reduction in recoil speed observed on d3 vs d5? Even with reduced ECM production from FRCs are there other cell types that can deposit and remodel the ECM
2. Another control group to add could be transiently depleting T cells and adding the adjuvant to determine if the increase in mesh size and expansion is in response to the influx of T cells or if the stimuli from the adjuvant is affecting the mechanics and response of the FRC network. This would help determine if T cells are responsible for the increased tissue tension/packing or if other lymphocyte populations are.
3. Figure 2C authors quantify nuclei per 100um³ but don't specify if this is CD3 only or also taking into account FRCs. Authors show FRC proliferating on d5. Separate quantification of T cells should be performed to determine if there are more T cells infiltrating the lymph node.
4. Authors should measure focal adhesions to determine degree of interaction of FRCs with ECM
5. Should quantify DC and other APCs in lymph node to determine if they also have reduced presence in the lymph node and if that is also causing a reduction in T cell retention/presence in lymph node
6. How does the high endothelial venule structure change with immune challenge and PDPN deletion? Are these structures that allow T cell trafficking altered (diameter, geometry, flow) due to the mechanical changes in FRC network?

Minor Critiques

7. ndr and dr appear to not be defined in the figure captions (Fig 6, Fig 8).

Reviewer #4:

Remarks to the Author:

The authors did a great job revising this manuscript and have included any additional valuable data that improve the quality of analysis shown. The key missing piece is the discussion of relevant literature – right now the discussion reads like a summary of the authors' findings rather than putting the findings into context. This needs to be revised to strengthen the impact of the manuscript and provide context for the readers who may not be as familiar with FRCs, tissue mechanics, ROCK signaling, etc.

A few additional minor comments remain:

- Some of the text in the methods, particularly new parts, needs revising for English (some sentences seem to be unfinished)
- Line 181: "Indeed dendritic cell" should be plural
- There are numerous occasions where the authors use present tense to describe their findings (though most of the manuscript is written in past tense for these descriptions), for examples lines 186-187 and line 229. The authors should carefully read through the manuscript to make sure all text describing the data is written in past tense
- Figure 7 lays out CD44+ and CD62L+ T cells; in the text and in the figure, it is a bit confusing to only say that cells are CD62L+ without also adding if they are CD44+/- . Each of these subsets represents a different cell type – CD44+CD62L- = effector/effector memory T cells, CD44+CD62L+ = central memory T cells, CD62L+CD44- = naïve T cells. CD44+ cells could be both central and effector/effector memory cells, so authors need to make sure this is clarified in the text and figure.
- The reduction in LECs in the KO mice is interesting, especially since the authors data suggests that there are no differences in naïve T cell recruitment. LECs are known to express CCL21, which recruits naïve T cells to the lymph nodes via CCR7+ signaling. CCL21 also recruits central memory T cells, so it would be interesting to see if this population (CD62L+CD44+) is changed at all. The authors have the data readily available so this would be easy to add to the figure for completeness. (This is a point that could for instance be added to the discussion as well)

Author Rebuttal, first revision:

NI-LE32588R1 point by point

Reviewer #2

The resubmission of the Horsnell et al., results in an improved manuscript – the authors have done a lot of work and should be applauded.... Indeed, they have gone far to describe the cell biology of fibroblastic reticular cells, which through direct and indirect mechanisms are key to immune responses.

We thank the Reviewer for their positive comments. We have extended the paper significantly based on the comments we received and feel these additions have significantly strengthened the manuscript.

Yet, while the cell biology is very nice, the immunology seems to be second to this. As above, I concede that the authors are studying a cell type key to immunity but I am not sure the study in its current form sufficiently falls within the breadth of interest of the intended audience.

The field of stromal biology is now well accepted as critical to our understanding of crucial processes that drive the immunological response. Our manuscript brings together molecular and cell biology with biophysics approaches to provide mechanistic insights into the phenomenon of lymphoid tissue remodelling. This is key to the understanding of how the lymph node tissue regulates immune responses at a whole tissue level. Our manuscript is the first mechanical study of lymph node expansion and we anticipate that our study will inspire many additional studies on the topic of mechanical control of immune function.

To make this more relevant to the average immunologist, we raised this point during the initial submission: *“The findings would be more accessible and exciting to the reader if they were tied to the outcome of an immune response. For instance, the authors show that ROCK inhibition results in diminished FRC proliferation at day 5 upon immunization. Do the authors speculate that this diminished proliferation, which did not impair CD45+ cell numbers, would impact the outcome of an immune response? The paper would be strengthened if this were tested.”*

We have shown in this resubmission that mechanical perturbation has a clear impact on the immune response. Namely, that the number of effector T cells is constrained when FRC proliferation is attenuated. Clarification of this point is also requested by reviewer 4.

- *We have extended our description of these data in the text and added additional panels to figure 7 and Supplementary figure 8. Lines 304-307*
- *We now show representative dot plots (CD62L vs. CD44 expression) for CD4+ and CD8+ T cell subsets in Figure 7H.*
- *We also show summary data of the numbers of all subsets in Figure 7I-J. CD25 expression is also included in Supplementary Figure 8A.*

Altering FRC network mechanics does not alter the number of naïve T cells in steady state or following immunisation (Figure 7H-J) suggesting that recruitment of T cells to the tissue is unaffected.

In the resubmission, the authors used CFA/OVA in their fabulous genetic model (Pdfracre;PDPNflox) and upon ROCK inhibition to evidence that their findings are immunologically-relevant. Indeed, they see alterations in T and B cells and T cell activation, but no disease/infection/autoimmunity relevance is provided. Indeed, are ROCK-inhibited, or Pdfracre;PDPNfl mice, more susceptible to viral infection or disease indications or are these findings relevant for an autoimmune context?

If no mouse studies will be pursued to this point, are single nucleotide polymorphisms in PDPN associated in humans with disease?

We are in agreement with the Reviewer that at this point we would not undertake further mouse studies but this point is very interesting and has been mentioned in the discussion as a future direction for additional studies (*Lines 428-430*).

To answer this Reviewer’s point, PDPN has clear relevance in human disease. Specifically, we have found several examples of missense mutations in podoplanin in patients with diffuse large B cell lymphoma (cbiportal.org). After discussion with the editor, this finding is now in the discussion (*Lines 428-430*).

We also work [REDACTED].

We are therefore confident that changes to PDPN expression are relevant to disease contexts.

Additionally, the data as presented leave the reader wondering about the mechanism underpinning their IFA/OVA experiments - the authors state on line 53 that alterations to FRCs “directly inhibit T cell activation and proliferation.” Yet, their data do not describe how this direct interaction occurs.

We agree that the mechanism would be better described as indirect, we have changed the text (*line 59*)

Based upon the authors previous works, it stands to reason that this would be indirect and any T cell phenotypes would be due to an inability of DCs to adequately prime T cells, due to a loss of CLEC2-PDPN interactions *in vivo*. It seems salient to investigate this possibility - examination of CD25 is unconvincing on this point, particularly as the data the authors reference is not statistically significant – or to provide other readouts to show the mechanism by which lack of PDPN on FRC or ROCK inhibition can mechanistically alter T or B cell responses.

Yes, since podoplanin is required as a ligand for dendritic cell migration this is an important point. We have already made reference to this in the revised text but we now include data to make this point clearer. The defect in DC trafficking is very clear in CLEC-2 deleted mouse models, and in these studies DCs use podoplanin expression on both afferent lymphatic vessels and within the FRCs network. Therefore CLEC-2 deletion significantly delays DC arrival in draining LNs. In our inducible PDPN deletion model there remains a small percentage of PDPN+ FRCs and the lymphatic endothelium is not targeted.

- We have included images and quantification of CD3+ Ki67+ cells in the tissue context, comparing steady state and day 5 post immune challenge (Supplementary Figure 8B-C) when tissue mechanics are perturbed through ROCK inhibition. Confirming that ROCK inhibition does not impair antigen-presentation or T cell activation *in vivo*.

Reviewer #3

(Remarks to the Author)

The authors have substantially revised the manuscript, and it is much more clearly written, and the methods used and how they are appropriate to probe certain questions is dramatically clearer, as are how the data is then interpreted. They have also added more data that support the central claims, much strengthening the report. Overall, the revision is much stronger and provides a compelling case for the interpretation presented.

We thank reviewer 3 for their very positive review of our revised manuscript.

While the current revisions have addressed some of the major concerns and improved upon the previous work, there are still some claims that are not rigorously supported by the current set of studies.

Major Critiques

1. Authors state that ECM fragmentation occurs but do not quantify the degree of fragmentation on d3 vs d5. While the ECM is remodeling, is there increased ECM protein expression on d5 than on d3, or a difference in the size of the fragments?

Could this explain the reduction in recoil speed observed on d3 vs d5?

Even with reduced ECM production from FRCs are there other cell types that can deposit and remodel the ECM

The Reviewer raises an important point since the revised manuscript did not include specific analysis of the ECM at day 3.

- In the revised manuscript we now make direct comparison between the ECM structure at day 3 and day 5 post immune challenge (Supplementary Figure 2)

We find that the disruption in the ECM occurs at day 5 which correlated with the increased network tension we measure by laser ablation – this data is already presented in the revised manuscript. We find no significant changes to the ECM at day 3, but at day 5 we quantify some areas with thicker fibrils. When combined with our data showing the breakages and disruption in the ECM network at day 5, these remaining thicker fibrils may be caused by recoil of broken fibres. We also observe that fibres appear straighter in draining LNs at day 5, consistent with the logic that the fibres would be pulled taught before breaking at day 5.

- A description of these data is included in the text (*lines 135-141*).

2. Another control group to add could be transiently depleting T cells and adding the adjuvant to determine if the increase in mesh size and expansion is in response to the influx of T cells or if the stimuli from the adjuvant is affecting the mechanics and response of the FRC network. This would help determine if T cells are responsible for the increased tissue tension/packing or if other lymphocyte populations are.

There are several studies using different adjuvants showing that T cell numbers drive FRC network remodelling (CFA, IFA, LPS, Montanide). Consistently these studies show an initial lag phase where the FRC network elongates, followed by a second phase of FRC proliferation. The timing of these phases varies with the immune challenge used but correlates with the number of T cells in the tissue. Additionally we have previously published that challenging lymph nodes with CLEC-2 recombinant protein alone did not cause LN expansion therefore leading us to conclude that increased number of lymphocytes drive tissue remodelling. Since we find that LN remodelling is mechanically driven, increased cell numbers of any immune cell population could contribute to the increased external forces exerted on the stromal architecture.

- We have expanded our description of these studies in the introduction (*Lines 26-32*)

3. Figure 2C authors quantify nuclei per 100um³ but don't specify if this is CD3 only or also taking into account FRCs.

This data is CD3+ cells and we have made this clear in the figure legend and the text (*line 94-95*).

Authors show FRC proliferating on d5. Separate quantification of T cells should be performed to determine if there are more T cells infiltrating the lymph node.

- As shown above, we have added this quantification of T cell proliferation in the tissue in Supplementary figure 8 B-C and clarify this point regarding naïve T cell recruitment and initiation of T cell proliferation in the text (*lines 338-339*).

4. Authors should measure focal adhesions to determine degree of interaction of FRCs with ECM

We do have evidence that focal complex formation is altered by both the CLEC-2/PDPN signalling axis and by actomyosin contractility. In a previous manuscript we also quantified changes to focal complexes induced by contact with dendritic cells as would be the case in the tissue context (*Figure 5, Martinez et al. 2019*)

Since cell-matrix adhesion in other tissue contexts is reinforced through force transmission through the cytoskeleton we concluded that reduced FRC-matrix adhesion would occur during the acute phase of LN expansion as an indirect consequence of reduced contractility through the FRC network.

- We have included this point in the discussion (lines 387-394).

5. Should quantify DC and other APCs in lymph node to determine if they also have reduced presence in the lymph node and if that is also causing a reduction in T cell retention/presence in lymph node

The *in vivo* data in PDPN deleted LN and Y27632 lymph nodes are deliberately short term assays to test tissue mechanics in the acute phase of LN expansion. Since it is known that PDPN has complex additional roles in FRC function (DC trafficking, HEV integrity, CCL21 availability) then we expect to find other interesting phenotypes in this model over a longer time frame.

- We have extended the discussion to better cover this point raised (lines 422-428)

6. How does the high endothelial venule structure change with immune challenge and PDPN deletion? Are these structures that allow T cell trafficking altered (diameter, geometry, flow) due to the mechanical changes in FRC network?

Since PDPN plays a vital role in HEV integrity, through production of S1P following contact with CLEC-2+ platelets (*Herzog et al. Nature 2013*) changes to HEV function is a likely phenotype and one we will plan to investigate. However, we do not observe alteration in LN cellularity in steady state, or changes to lymphocyte recruitment in the short time frame of our current mechanically focussed study. Any loss of HEV integrity might develop over a longer time frame and could be the basis for a follow up study on the broad range of PDPN-dependent lymphoid tissue phenotypes.

Minor Critiques

7. ndr and dr appear to not be defined in the figure captions (Fig 6, Fig 8).

We have corrected these in both captions.

Reviewer #4

(Remarks to the Author)

The authors did a great job revising this manuscript and have included any additional valuable data that improve the quality of analysis shown. The key missing piece is the discussion of relevant literature – right now the discussion reads like a summary of the authors’ findings rather than putting the findings into context. This needs to be revised to strengthen the impact of the manuscript and provide context for the readers who may not be as familiar with FRCs, tissue mechanics, ROCK signaling, etc.

We thank the reviewer for their positive review. Given the interdisciplinary nature of our study, the reviewer raises an important point. We agree that a broader, wider discussion will further strengthen the manuscript.

- We have added discussion on the interplay between cell contractility, cell-matrix adhesion and cell-cell adhesion (*lines 387-394*)
- We have added discussion of the additional functions of podoplanin (*lines 422-428*)
- We have also added comment on the relevance of podoplanin in disease contexts (*lines 428-430*)
- We have raised the question of how we should as immunologists consider the role of cell and tissue mechanics in immune function (*line 438-440*)

A few additional minor comments remain:

- Some of the text in the methods, particularly new parts, needs revising for English (some sentences seem to be unfinished)

- Line 181: “Indeed dendritic cell” should be plural
- There are numerous occasions where the authors use present tense to describe their findings (though most of the manuscript is written in past tense for these descriptions), for examples lines 186-187 and line 229. The authors should carefully read through the manuscript to make sure all text describing the data is written in past tense

We have corrected these minor edits in a revised manuscript and will work with the editor to ensure the manuscript complies with the style of the journal if accepted.

- Figure 7 lays out CD44⁺ and CD62L⁺ T cells; in the text and in the figure, it is a bit confusing to only say that cells are CD62L⁺ without also adding if they are CD44⁺/. Each of these subsets represents a different cell type – CD44⁺CD62L⁻ = effector/effector memory T cells, CD44⁺CD62L⁺ = central memory T cells, CD62L⁺CD44⁻ = naïve T cells. CD44⁺ cells could be both central and effector/effector memory cells, so authors need to make sure this is clarified in the text and figure.

As also requested by Reviewer 2 we have edited Figure 7 to clarify this point

- *We have extended our description of these data in the text and added additional panels to figure 7 and Supplementary figure 8. Lines 304-307*
- *We now show representative dot plots (CD62L vs. CD44 expression) for CD4⁺ and CD8⁺ T cell subsets in Figure 7H.*
- *We also show summary data of the numbers of all subsets in Figure 7I-J. CD25 expression is also included in Supplementary Figure 8A.*

Altering FRC network mechanics does not alter the number of naïve T cells in steady state or following immunisation (Figure 7H-J) suggesting that recruitment of T cells to the tissue is unaffected.

- The reduction in LECs in the KO mice is interesting, especially since the authors data suggests that there are no differences in naïve T cell recruitment. LECs are known to express CCL21, which recruits naïve T cells to the lymph nodes via CCR7⁺ signaling. CCL21 also recruits central memory T cells, so it would be interesting to see if this population (CD62L⁺CD44⁺) is changed at all. The authors have the data readily available so this would be easy to add to the figure for completeness. (This is a point that could for instance be added to the discussion as well)

- *As detailed above we now show central memory T cells are unaffected.*

- We have also added a comment on the interaction between PDPN and CCL21 in the discussion (lines 422-425)

Decision Letter, second revision:

Subject: Your manuscript, NI-LE32588B

Message: Our ref: NI-LE32588B

2nd Jun 2022

Dear Dr. Acton,

Thank you for your patience as we've prepared the guidelines for final submission of your Nature Immunology manuscript, "Lymph node tissue homeostasis and adaptation to immune challenge resolved by fibroblast network mechanics" (NI-LE32588B). Please carefully follow the step-by-step instructions provided in the attached file, and add a response in each row of the table to indicate the changes that you have made. Please also check and comment on any additional marked-up edits we have proposed within the text. Ensuring that each point is addressed will help to ensure that your revised manuscript can be swiftly handed over to our production team.

We would like to start working on your revised paper, with all of the requested files and forms, by June 14th. Please get in contact with us if you anticipate delays.

When you upload your final materials, please include a point-by-point response to any remaining reviewer comments and please make sure to upload your checklist.

If you have not done so already, please alert us to any related manuscripts from your group that are under consideration or in press at other journals, or are being written up for submission to other journals (see: <https://www.nature.com/nature-portfolio/editorial-policies/plagiarism#policy-on-duplicate-publication> for details).

In recognition of the time and expertise our reviewers provide to Nature Immunology's editorial process, we would like to formally acknowledge their contribution to the external peer review of your manuscript entitled "Lymph node tissue homeostasis and adaptation to immune challenge resolved by fibroblast network mechanics". For those reviewers who give their assent, we will be publishing their names alongside the published article.

Nature Immunology offers a Transparent Peer Review option for new original research manuscripts submitted after December 1st, 2019. As part of this initiative, we encourage our authors to support increased transparency into the peer review process by agreeing to have the reviewer comments, author rebuttal letters, and editorial decision letters published as a Supplementary item. When you submit your final files please clearly state in your cover letter whether or not you would like to participate in this initiative. Please note that failure to state your preference will result in delays in accepting your manuscript for publication.

Cover suggestions

As you prepare your final files we encourage you to consider whether you have any images or illustrations that may be appropriate for use on the cover of Nature

Immunology.

Nature Immunology has now transitioned to a unified Rights Collection system which will allow our Author Services team to quickly and easily collect the rights and permissions required to publish your work. Approximately 10 days after your paper is formally accepted, you will receive an email in providing you with a link to complete the grant of rights. If your paper is eligible for Open Access, our Author Services team will also be in touch regarding any additional information that may be required to arrange payment for your article.

Please note that *Nature Immunology* is a Transformative Journal (TJ). Authors may publish their research with us through the traditional subscription access route or make their paper immediately open access through payment of an article-processing charge (APC). Authors will not be required to make a final decision about access to their article until it has been accepted. [Find out more about Transformative Journals](https://www.springernature.com/gp/open-research/transformative-journals).

If you have any questions about costs, Open Access requirements, or our legal forms, please contact ASJournals@springernature.com.

Please use the following link for uploading these materials: [REDACTED]

Best regards,

Elle Morris
Senior Editorial Assistant
Nature Immunology
Phone: 212 726 9207
Fax: 212 696 9752
E-mail: immunology@us.nature.com

On behalf of

Laurie A. Dempsey, Ph.D.
Senior Editor
Nature Immunology
l.dempsey@us.nature.com
ORCID: 0000-0002-3304-796X

Final Decision Letter:

Subject: Decision on Nature Immunology submission NI-A32588C

Message: In reply please quote: NI-A32588C

Dear Sophie,

I am delighted to accept your manuscript entitled "Lymph node tissue homeostasis and adaptation to immune challenge resolved by fibroblast network mechanics" for publication in an upcoming issue of Nature Immunology.

Over the next few weeks, your paper will be copyedited to ensure that it conforms to Nature Immunology style. Once your paper is typeset, you will receive an email with a link to choose the appropriate publishing options for your paper and our Author Services team will be in touch regarding any additional information that may be required.

Due to the importance of these deadlines, we ask that you please let us know now whether you will be difficult to contact over the next month. If this is the case, we ask you

provide us with the contact information (email, phone and fax) of someone who will be able to check the proofs on your behalf, and who will be available to address any last-minute problems.

Please note that *Nature Immunology* is a Transformative Journal (TJ). Authors may publish their research with us through the traditional subscription access route or make their paper immediately open access through payment of an article-processing charge (APC). Authors will not be required to make a final decision about access to their article until it has been accepted. [Find out more about Transformative Journals](https://www.springernature.com/gp/open-research/transformative-journals).

Authors may need to take specific actions to achieve [compliance with funder and institutional open access mandates](https://www.springernature.com/gp/open-research/funding/policy-compliance-faqs). If your research is supported by a funder that requires immediate open access (e.g. according to [Plan S principles](https://www.springernature.com/gp/open-research/plan-s-compliance)) then you should select the gold OA route, and we will direct you to the compliant route where possible. For authors selecting the subscription publication route, the journal's standard licensing terms will need to be accepted, including [self-archiving policies](https://www.springernature.com/gp/open-research/policies/journal-policies). Those licensing terms will supersede any other terms that the author or any third party may assert apply to any version of the manuscript.

Your paper will be published online soon after we receive your corrections and will appear in print in the next available issue. Content is published online weekly on Mondays and Thursdays, and the embargo is set at 16:00 London time (GMT)/11:00 am US Eastern time (EST) on the day of publication. Now is the time to inform your Public Relations or Press Office about your paper, as they might be interested in promoting its publication. This will allow them time to prepare an accurate and satisfactory press release. Include your manuscript tracking number (NI-A32588C) and the name of the journal, which they will need when they contact our office.

About one week before your paper is published online, we shall be distributing a press release to news organizations worldwide, which may very well include details of your work. We are happy for your institution or funding agency to prepare its own press release, but it must mention the embargo date and *Nature Immunology*. Our Press Office will contact you closer to the time of publication, but if you or your Press Office have any enquiries in the meantime, please contact press@nature.com.

Also, if you have any spectacular or outstanding figures or graphics associated with your

manuscript - though not necessarily included with your submission - we'd be delighted to consider them as candidates for our cover. Simply send an electronic version (accompanied by a hard copy) to us with a possible cover caption enclosed.

Please note that we encourage the authors to self-archive their manuscript (the accepted version before copy editing) in their institutional repository, and in their funders' archives, six months after publication. Nature Portfolio recognizes the efforts of funding bodies to increase access of the research they fund, and strongly encourages authors to participate in such efforts. For information about our editorial policy, including license agreement and author copyright, please visit www.nature.com/ni/about/ed_policies/index.html

Kind regards,

Laurie

Laurie A. Dempsey, Ph.D.
Senior Editor
Nature Immunology
l.dempsey@us.nature.com
ORCID: 0000-0002-3304-796X